



# Review article: How does glacier discharge affect marine biogeochemistry and primary production in the Arctic?

Mark J. Hopwood[1], Dustin Carroll[2], Thorben Dunse[3,4], Andy Hodson[3,5], Johnna M. Holding[6], José L. Iriarte[7], Sofia Ribeiro[8], Eric P. Achterberg[1], Carolina Cantoni[9], Daniel F. Carlson[14], Melissa Chierici[5,10], Jennifer S. Clarke[1], Stefano Cozzi[9], Agneta Fransson[11], Thomas Juul-Pedersen[12], Mie H. S. Winding[12], and Lorenz Meire[12,13]

[1]GEOMAR Helmholtz Centre for Ocean Research Kiel, Kiel, Germany
[2]Moss Landing Marine Laboratories, San José State University, Moss Landing, CA
[3]Western Norway University of Applied Sciences, Sogndal, Norway
[4]The University of Oslo, Oslo, Norway
[5]The University Centre in Svalbard, Longyearbyen, Svalbard
[6]Department of Bioscience, Aarhus University, Silkeborg, Denmark
[7]Instituto de Acuicultura and Centro Dinámica de Ecosistemas Marinos de Altas Latitudes – IDEAL,
Universidad Austral de Chile, Puerto Montt, Chile
[8]Geological Survey of Denmark and Greenland, Copenhagen, Denmark
[9]CNR-ISMAR Istituto di Scienze Marine, Trieste, Italy
[10]Institute of Marine Research, Fram Centre, Tromsø, Norway
[11]Norwegian Polar Institute, Fram Centre, Tromsø, Norway
[12]Greenland Climate Research Centre, Greenland Institute of Natural Resources, Nuuk, Greenland
[13]Royal Netherlands Institute for Sea Research, and Utrecht University, Yerseke, the Netherlands
[14]Institute of Coastal Research, Helmholtz-Zentrum Geesthacht, Centre for Materials and Coastal Research,
Geesthacht, Germany

**Correspondence:** Mark J. Hopwood (mhopwood@geomar.de)

**Abstract.** Freshwater discharge from glaciers is increasing across the Arctic in response to anthropogenic climate change, which raises questions about the potential downstream effects in the marine environment. Whilst a combination of long-term monitoring programmes and intensive Arctic field campaigns have improved our knowledge of glacier–ocean interactions in recent years, especially with respect to fjord/ocean circulation, there are extensive knowledge gaps concerning how glaciers affect marine biogeochemistry and productivity. Following two cross-cutting disciplinary International Arctic Science Committee (IASC) workshops addressing the importance of glaciers for the marine ecosystem, here we review the state of the art concerning how freshwater discharge affects the marine environment with a specific focus on marine biogeochemistry and biological productivity. Using a series of Arctic case studies (Nuup Kangerlua/Godthåbsfjord, Kongsfjorden, Kangerluarsuup Sermia/Bowdoin Fjord, Young Sound and Sermilik Fjord), the interconnected effects of freshwater discharge on fjord–shelf exchange, nutrient availability, the carbonate system, the carbon cycle and the microbial food web are investigated. Key findings are that whether the effect of glacier discharge on marine primary production is positive or negative is highly dependent on a combination of factors. These include glacier type (marine- or land-terminating), fjord–glacier geometry and the limiting resource(s) for phytoplankton growth in a specific spatio-temporal region (light, macronutrients or micronutrients). Arctic glacier fjords therefore often exhibit distinct discharge–productivity relationships, and multiple case-studies must be considered in order to understand the net effects of glacier discharge on Arctic marine ecosystems.

## 1 Introduction

Annual freshwater discharge volume from glaciers has increased globally in recent decades (Rignot et al., 2013; Bamber et al., 2018; Mouginot et al., 2019) and will continue to do so across most Arctic regions until at least the middle of this century under a Representative Concentration Pathway (RCP) 4.5 climate scenario (Bliss et al., 2014; Huss and Hock, 2018). This increase in discharge (surface runoff and subsurface discharge into the ocean) raises questions about the downstream effects in marine ecosystems, particularly with respect to ecosystem services such as carbon sequestration and fisheries (Meire et al., 2015, 2017; Milner et al., 2017). In order to understand the effect of glaciers on the present-day marine environment and under future climate scenarios, knowledge of the physical and chemical perturbations occurring in the water column as a result of glacier discharge and the structure, function, and resilience of ecosystems within these regions must be synthesized.

Quantifying the magnitude of environmental perturbations from glacial discharge is complicated by the multiple concurrent, and occasionally counteracting, effects that glacial discharge has in the marine environment. For example, ice-rock abrasion means that glacially fed rivers can carry higher sediment loads than temperate rivers (Chu et al., 2009; Overeem et al., 2017). Extensive sediment plumes where glacier discharge first enters the ocean limit light penetration into the water column (Murray et al., 2015; Halbach et al., 2019), and ingestion of glacial flour particles can be hazardous, or even fatal, to zooplankton, krill and benthic fauna (White and Dagg, 1989; Włodarska-Kowalczuk and Pearson, 2004; Arendt et al., 2011; Fuentes et al., 2016). However, these plumes also provide elevated concentrations of inorganic components such as calcium carbonate, which affects seawater alkalinity (Yde et al., 2014; Fransson et al., 2015), and dissolved silicic acid (hereafter Si) (Brown et al., 2010; Meire et al., 2016a) and iron (Fe) (Statham et al., 2008; Lippiatt et al., 2010), which can potentially increase marine primary production (Gerringa et al., 2012; Meire et al., 2016a).

The impacts of glacier discharge can also depend upon the spatial and temporal scales investigated (van de Poll et al., 2018). In semi-enclosed Arctic coastal regions and fjord systems, summertime discharge typically produces strong, near-surface stratification. This results in a shallow, nutrient-poor layer which reduces primary production and drives phytoplankton biomass deeper in the water column (Rysgaard et al., 1999; Juul-Pedersen et al., 2015; Meire et al., 2017). On broader scales across continental shelves, freshening can similarly reduce vertical nutrient supply throughout summer (Coupel et al., 2015) but may also impede the breakdown of stratification in autumn, thereby extending the phytoplankton growing season (Oliver et al., 2018). Key research questions are how and on what spatial and temporal timescales these different effects interact to enhance, or reduce, marine primary production. Using a synthesis of field studies from glacier catchments with different characteristics (Fig. 1), we provide answers to three questions arising from two interdisciplinary workshops on the importance of Arctic glaciers for the marine ecosystem under the umbrella of the International Arctic Science Committee (IASC).

1. Where and when does glacial freshwater discharge promote or reduce marine primary production?

2. How does spatio-temporal variability in glacial discharge affect marine primary production?

3. How far-reaching are the effects of glacial discharge on marine biogeochemistry?

## 2 Fjords as critical zones for glacier–ocean interactions

In the Arctic and sub-Antarctic, most glacial discharge enters the ocean through fjord systems (Iriarte et al., 2014; Straneo and Cenedese, 2015). The strong lateral gradients and seasonal changes in environmental conditions associated with glacial discharge in these coastal environments differentiate these ecosystems from offshore systems (Arendt et al., 2013; Lydersen et al., 2014; Krawczyk et al., 2018). Fjords can be efficient sinks for organic carbon (Smith et al., 2015) and $CO_2$ (Rysgaard et al., 2012; Fransson et al., 2015), sustain locally important fisheries (Meire et al., 2017) and are critical zones for deep mixing which dictate how glacially modified waters are exchanged with the coastal ocean (Mortensen et al., 2014; Straneo and Cenedese, 2015; Beaird et al., 2018). Fjord-scale processes therefore comprise an integral part of all questions concerning how glacial discharge affects Arctic coastal primary production (Arimitsu et al., 2012; Renner et al., 2012; Meire et al., 2017).

Fjords act as highly stratified estuaries and provide a pathway for the exchange of heat, salt, and nutrients between near-glacier waters and adjacent coastal regions (Mortensen et al., 2014, 2018; Straneo and Cenedese, 2015). In deep fjords, such as those around much of the periphery of Greenland, warm, saline water is typically found at depth (> 200 m), overlaid by cold, fresher water and, during summer, a thin layer (∼ 50 m or less) of relatively warm near-surface water (Straneo et al., 2012). The injection of freshwater into fjords from subglacial discharge (Xu et al., 2012; Carroll et al., 2015) and terminus (Slater et al., 2018) and iceberg melt (Moon et al., 2018) can drive substantial buoyancy-driven flows in the fjord (Carroll et al., 2015, 2017; Jackson et al., 2017), which amplify exchange with the shelf system as well as submarine melting and the calving rates of glacier termini. To date, such modifications to circulation and exchange between glacier fjords and shelf waters have primarily been studied in terms of their effects on ocean physics and melting at glacier termini, yet they also have profound impacts on marine productivity (Meire et al., 2016a; Kanna et al., 2018; Torsvik et al., 2019).

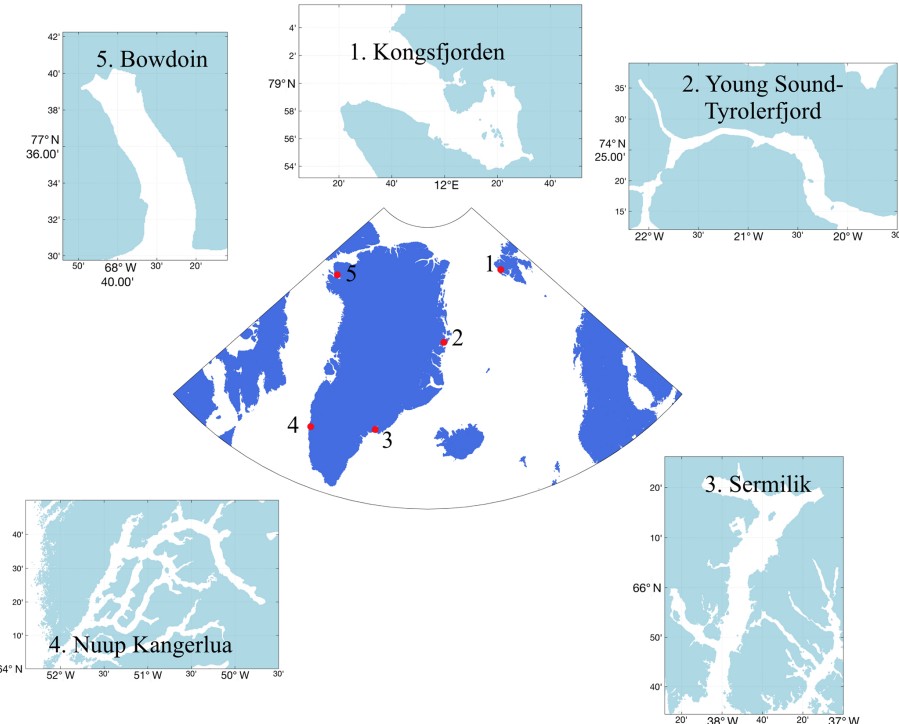

**Figure 1.** Locations of five key Arctic field sites, where extensive work bridging the glacier and marine domains has been conducted, discussed herein in order to advance understanding of glacier–ocean interactions. 1: Kongsfjorden (Svalbard); 2: Young Sound (E Greenland); 3: Sermilik (SE Greenland); 4: Nuup Kangerlua/Godthåbsfjord (SW Greenland); 5: Bowdoin Fjord/Kangerluarsuup Sermia (NW Greenland).

While renewal of fjord waters from buoyancy-driven processes is mainly thought to occur over seasonal to sub-annual timescales (Gladish et al., 2014; Mortensen et al., 2014; Carroll et al., 2017), energetic shelf forcing (i.e. from coastal/katabatic winds and coastally trapped waves) can result in rapid exchange over synoptic timescales (Straneo et al., 2010; Jackson et al., 2014; Moffat, 2014) and similarly also affect marine productivity (Meire et al., 2016b). Katabatic winds are common features of glaciated fjords. Down-fjord wind events facilitate the removal of low-salinity surface waters and ice from glacier fjords, as well as the inflow of warmer, saline waters at depth (Johnson et al., 2011). The frequency, direction and intensity of wind events throughout the year thus adds further complexity to the effect that fjord geometry has on fjord–shelf exchange processes (Cushman-Roisin et al., 1994; Spall et al., 2017). Topographic features such as sills and lateral constrictions can exert a strong control on fjord–shelf exchange (Gladish et al., 2014; Carroll et al., 2017, 2018). Ultimately, circulation can thereby vary considerably depending on fjord geometry and the relative contributions from buoyancy, wind and shelf forcing (Straneo and Cenedese, 2015; Jackson et al., 2018). Some variability in the spatial patterns of primary production is therefore expected between Arctic glacier fjord systems as differences in geometry and forcing affect exchange with the shelf and water column structure.

These changes affect the availability of the resources which constrain local primary production (Meire et al., 2016b; Arimitsu et al., 2016; Calleja et al., 2017).

> **Nuup Kangerlua / Godthåbsfjord (SW Greenland) 64° N, 051° W**
>
> Nuup Kangerlua (also known as Godthåbsfjord) is a large glacier fjord system (~190 km long, 4–8 km wide and up to 625 m deep). The fjord hosts six different glaciers (three land-terminating and three marine-terminating), including the marine-terminating glaciers Kangiata Nunaata Sermia, Akugdlerssup Sermia, and Narsap Sermia. The shallowest sill within the fjord is at ~170 m depth (Mortensen et al., 2011). Nuup Kangerlua is one of few well-studied Greenland fjord systems, due to extensive work conducted by the Greenland Institute of Natural Resources. A data portal is available containing monthly fjord data through the Greenland Ecosystem Monitoring Programme (GEM; http://g-e-m.dk).

Fjord–shelf processes also contribute to the exchange of active cells and microbial species' resting stages, thus preconditioning primary production prior to the onset of the growth season (Krawczyk et al., 2015, 2018). Protists (unicellular eukaryotes) are the main marine primary producers in the Arctic. This highly specialized and diverse group includes species that are ice-associated (sympagic) and/or pelagic. Many protists in fjords and coastal areas of the Arctic maintain diverse seed banks of resting stages, which promotes the resilience and adaptability of species on timescales from seasons to decades (Ellegaard and Ribeiro, 2018). Yet seawater inflow into fjords can still change the dominant species within a single season. In Nuup Kangerlua (Godthåb-

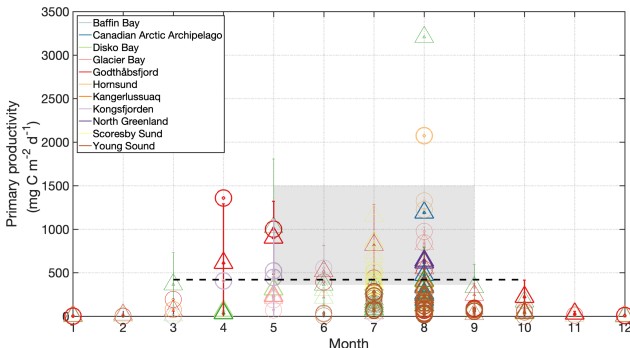

**Figure 2.** Primary production for Arctic glacier fjord systems including Disko Bay (Andersen, 1977; Nielsen and Hansen, 1995; Jensen et al., 1999; Nielsen, 1999; Levinsen and Nielsen, 2002), Godthåbsfjord (Juul-Pedersen et al., 2015; Meire et al., 2017), Kangerlussuaq (Lund-Hansen et al., 2018), Kongsfjorden (Hop et al., 2002; Iversen and Seuthe, 2011; Hodal et al., 2012; van de Poll et al., 2018), Nordvestfjord/Scoresby Sund (Seifert et al., 2019), Hornsund (Smoła et al., 2017), Young Sound (Rysgaard et al., 1999; Meire et al., 2017; Holding et al., 2019), the Canadian Arctic Archipelago (Harrison et al., 1982) and Glacier Bay (Reisdorph and Mathis, 2015). Circles represent glacier fjords, triangles are sites beyond glacier fjords and bold markers are < 80 km from a marine-terminating glacier. Error bars are standard deviations for stations where multiple measurements were made at the same station. Dashed line is the pan-Arctic mean primary production (March–September). Shaded area is the pan-Arctic shelf range of primary production for May–August (Pabi et al., 2008).

sfjord), the spring phytoplankton bloom is typically dominated by *Fragilariopsis* spp. diatoms and *Phaeocystis* spp. haptophytes. Unusually prolonged coastal seawater inflow in spring 2009 led to the mass occurrence of chain-forming
[5] *Thalassiosira* spp. diatoms and the complete absence of the normally abundant *Phaeocystis* spp. (Krawczyk et al., 2015) – a pattern which has been found elsewhere in the Arctic, including Kongsfjorden (Hegseth and Tverberg, 2013).

## 3   Pelagic primary production in Arctic glacier fjords

[10] Key factors controlling rates of primary production across Arctic marine environments are light availability, nutrient availability and grazing (Nielsen, 1999; Taylor et al., 2013; Arrigo and van Dijken, 2015; Tremblay et al., 2015). Seasonal changes in the availability of bioessential resources, the
[15] structure of the water column and the feeding patterns of zooplankton thereby interact to produce distinct bloom periods of high primary production shouldered by periods of low primary production. In glacier fjords, strong lateral and vertical gradients in some, or all, of these factors create a far more dynamic situation for primary producers than in the open ocean
[20] (Etherington and Hooge, 2007; Arendt et al., 2010; Murray et al., 2015).

Large inter- and intra-fjord differences in primary production are demonstrated by field observations around the Arctic which show that glacier fjords range considerably in productivity from very low ($< 40 \, \mathrm{mg \, C \, m^{-2} \, d^{-1}}$) to moderately pro- [25] ductive systems ($> 500 \, \mathrm{mg \, C \, m^{-2} \, d^{-1}}$) during the meltwater season (e.g. Jensen et al., 1999; Rysgaard et al., 1999; Hop et al., 2002; Meire et al., 2017). For comparison, the pan-Arctic basin exhibits a mean production of $420 \pm 26 \, \mathrm{mg \, C \, m^{-2} \, d^{-1}}$ [30] (mean March–September 1998–2006) (Pabi et al., 2008), which has increased across most regions in recent decades due to reduced summertime sea-ice coverage (Arrigo and van Dijken, 2015), and summertime (May–August) Arctic shelf environments exhibit a range of $360–1500 \, \mathrm{mg \, C \, m^{-2} \, d^{-1}}$ [35] (Pabi et al., 2008). So is it possible to generalize how productive Arctic glacier fjords are?

Extensive measurements of primary production throughout the growth season in glacier fjords are only available for Godthåbsfjord (Juul-Pedersen et al., 2015; Meire et al., [40] 2017), Young Sound (Rysgaard et al., 1999; Meire et al., 2017; Holding et al., 2019), Glacier Bay (Alaska, Reisdorph and Mathis, 2015), Hornsund (Svalbard, Smoła et al., 2017) and Kongsfjorden (Iversen and Seuthe, 2011; van de Poll et al., 2018). Observations elsewhere are sparse and typically [45] limited to summertime-only data. Generalizing across multiple Arctic glacier fjord systems therefore becomes challenging due to the paucity of data and the different geographic and seasonal context of individual primary production data points (Fig. 2). Furthermore there are poten- [50] tially some methodological implications when comparing direct measurements of primary production using $^{14}\mathrm{C}$ uptake (e.g. Holding et al., 2019), with estimates derived from changes in water column macronutrient (e.g. Seifert et al., 2019) or dissolved inorganic carbon (e.g. Reisdorph and [55] Mathis, 2015) inventories.

Nevertheless, some quantitative comparison can be made if we confine discussion to months where a meltwater signal may be evident in most glaciated regions (July–September). All available data for Arctic glaciated regions can then be [60] pooled according to whether it refers to primary production within a glacier fjord and whether or not it could plausibly be influenced by the presence of a marine-terminating glacier (see Sect. 5). For the purposes of defining the spatial extent of individual glacier fjords, we consider broad bay areas such [65] as the lower and central parts of Glacier Bay (Etherington and Hooge, 2007; Reisdorph and Mathis, 2015), Scoresby Sund (Scoresby Sound in English; Seifert et al., 2019) and Disko Bay (Jensen et al., 1999; Nielsen, 1999) to be beyond the scale of the associated glacier fjords on the basis of [70] the oceanographic interpretation presented in the respective studies. Defining the potential spatial influence of marine-terminating glaciers is more challenging. Using observations from Godthåbsfjord, where primary production is found to be affected on a scale of 30–80 km down-fjord from the marine- [75] terminating glaciers therein (Meire et al., 2017), we define

a region $< 80\,\text{km}$ downstream of calving fronts as being potentially influenced by marine-terminating glaciers.

Four exclusive categories of primary production data result (Table 1). Primary production for group I is significantly higher than any other group, and group II is also significantly higher than group IV ($p < 0.025$). Primary production is higher in regions designated as having a potential marine-terminating glacier influence. On the contrary, other near-glacier regions (i.e. with land-terminating glaciers) seem to have low summertime primary productivity, irrespective of how mean Arctic primary production is defined (Table 1). What processes could lead to such differences? In the next sections of this review we discuss the biogeochemical features of glacier-affected marine regions that could potentially explain such trends if they do not simply reflect data deficiency.

## 4 Effects of glacial discharge on marine resource availability

One of the most direct mechanisms via which glacial discharge affects downstream marine primary production is by altering the availability of light, macronutrients (such as nitrate, $NO_3$; phosphate, $PO_4$; and silicic acid, Si) and/or micronutrients (such as iron and manganese) in the ocean. The chemical composition of glacial discharge is now relatively well constrained, especially around Greenland (Yde et al., 2014; Meire et al., 2016a; Stevenson et al., 2017), Alaska (Hood and Berner, 2009; Schroth et al., 2011) and Svalbard (Hodson et al., 2004, 2016). Whilst high particle loads (Chu et al., 2012; Overeem et al., 2017) and Si are often associated with glacially modified waters (Fig. 3a) around the Arctic (Brown et al., 2010; Meire et al., 2016a), the concentrations of all macronutrients in glacial discharge (Meire et al., 2016a) are relatively low and similar to those of coastal seawater (Fig. 3a, b and c).

Macronutrient concentrations in Arctic rivers can be higher than in glacier discharge (Holmes et al., 2011) (Fig. 3d, e and f). Nevertheless, river and glacier meltwater alike do not significantly increase the concentration of $PO_4$ in Arctic coastal waters (Fig. 3c and f). River water is, relatively, a much more important source of $NO_3$ (Cauwet and Sidorov, 1996; Emmerton et al., 2008; Hessen et al., 2010), and in river estuaries this nutrient can show a sharp decline with increasing salinity due to both mixing and biological uptake (Fig. 3e). Patterns in Si are more variable (Cauwet and Sidorov, 1996; Emmerton et al., 2008; Hessen et al., 2010). Dissolved Si concentration at low salinity is higher in rivers than in glacier discharge (Fig. 3a and d), yet a variety of estuarine behaviours are observed across the Arctic. Peak dissolved Si occurs at a varying salinity, due to the opposing effects of Si release from particles and dissolved Si uptake by diatoms (Fig. 3d).

---

**Kongsfjorden (W Svalbard) 79° N, 012° E**

Kongsfjorden is a small Arctic fjord on the west coast of Svalbard notable for pronounced sediment plumes originating from multiple proglacial streams and several shallow marine-terminating glaciers. There is no sill at the fjord entrance, and thus warm Atlantic water can be found throughout the fjord in summer (Hop et al., 2002). The major marine-terminating glaciers at the fjord head (Kongsvegen and Kronebreen) have been retreating since before monitoring began (Liestøl, 1988; Svendsen et al., 2002) and are anticipated to transition to land-terminating systems in the coming decades (Torsvik et al., 2019). Research within the fjord is logged in the RIS (Research in Svalbard; https://researchinsvalbard.no) online system.

---

A notable feature of glacial freshwater outflows into the ocean is the high turbidity that occurs in most Arctic glacier fjords. High turbidity in surface waters within glacier fjords arises from the high sediment transport in these drainage systems (Chu et al., 2012), from iceberg melting and also from the resuspension of fine sediments (Azetsu-Scott and Syvitski, 1999; Zajączkowski and Włodarska-Kowalczuk, 2007; Stevens et al., 2016). The generally high sediment load of glacially derived freshwater is evident around Greenland, which is the origin of $\sim 1\,\%$ of annual freshwater discharge into the ocean yet $7\,\%$–$9\,\%$ of the annual fluvial sediment load (Overeem et al., 2017). Sediment load is however spatially and temporally variable, leading to pronounced inter- and intra-catchment differences (Murray et al., 2015). For example, satellite-derived estimates of sediment load for 160 Greenlandic glacier outflows suggest a median sediment load of $992\,\text{mg}\,\text{L}^{-1}$, but some catchments exhibit $> 3000\,\text{mg}\,\text{L}^{-1}$ (Overeem et al., 2017). Furthermore it is suggested that $> 25\,\%$ of the total annual sediment load is released in a single outflow (from the Sermeq glacier) (Overeem et al., 2017).

The extent to which high turbidity in glacier outflows limits light availability in downstream marine environments is therefore highly variable between catchments and with distance from glacier outflows (Murray et al., 2015; Mascarenhas and Zielinski, 2019). The occurrence, and effects, of subsurface turbidity peaks close to glaciers is less well studied. Subsurface turbidity features may be even more spatially and temporally variable than their surface counterparts (Stevens et al., 2016; Kanna et al., 2018; Moskalik et al., 2018). In general, a spatial expansion of near-surface turbid plumes is expected with increasing glacier discharge, but this trend is not always evident at the catchment scale (Chu et al., 2009, 2012; Hudson et al., 2014). Furthermore, with long-term glacier retreat, the sediment load in discharge at the coastline is generally expected to decline as proglacial lakes are efficient sediment traps (Bullard, 2013; Normandeau et al., 2019).

In addition to high turbidity, the low concentration of macronutrients in glacier discharge relative to saline waters is evidenced by the estuarine mixing diagram in Kongsfjorden (Fig. 3) and confirmed by extensive measurements of freshwater nutrient concentrations (e.g. Hodson et al., 2004, 2005). For $PO_4$ (Fig. 3c), there is a slight increase in concentration with salinity (i.e. discharge dilutes the nutrient concentration in the fjord). For $NO_3$, discharge slightly increases

**Table 1.** July–September marine primary production (PP) data from studies conducted in glaciated Arctic regions. PP data points are categorised into four groups according to whether or not they are within 80 km of a marine-terminating glacier and whether or not they are within a glacier fjord. Data sources as per Fig. 2. $n$ is the number of data points; where studies report primary production measurements at the same station for the same month at multiple time points (e.g. Juul-Pedersen et al., 2015) a single mean is used in the data compilation (i.e. $n = 1$ irrespective of the historical extent of the time series).

| Category | Mean PP ($\pm$ standard deviation) mg C m$^{-2}$ d$^{-1}$ | $n$ | Data from |
|---|---|---|---|
| (I) Marine-terminating glacier influence, non-fjord | $847 \pm 852$ | 11 | Disko Bay, Scoresby Sund, Glacier Bay, North Greenland, Canadian Arctic Archipelago |
| (II) Marine-terminating glacier influence, glacier fjord | $480 \pm 403$ | 33 | Godthåbsfjord, Kongsfjorden, Scoresby Sund, Glacier Bay, Hornsund, |
| (III) No marine-terminating glacier influence, non-fjord | $304 \pm 261$ | 42 | Godthåbsfjord, Young Sound, Scoresby Sund, Disko Bay, Canadian Arctic Archipelago |
| (IV) No marine-terminating glacier influence, glacier fjord | $125 \pm 102$ | 35 | Godthåbsfjord, Young Sound, Kangerlussuaq, Disko Bay |

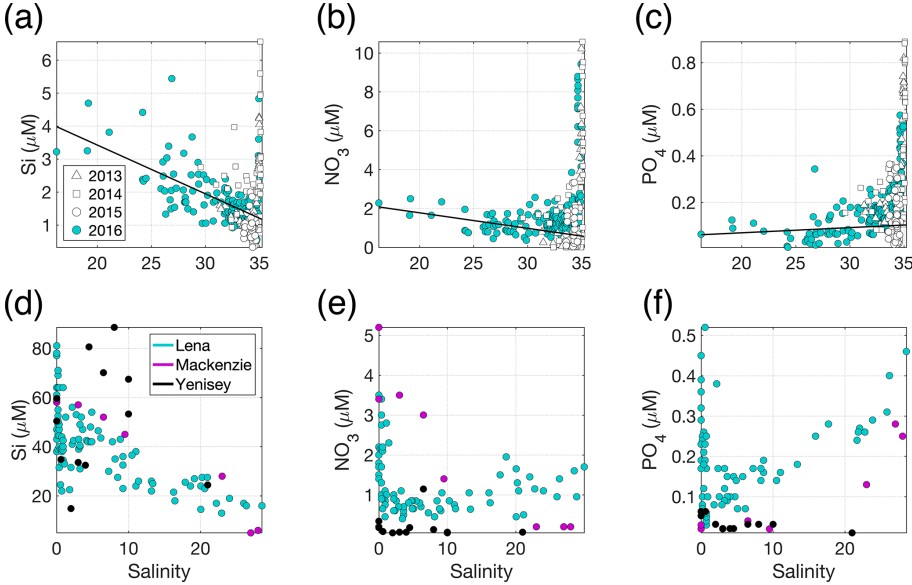

**Figure 3. (a)** Si, **(b)** NO$_3$ and **(c)** PO$_4$ distributions across the measured salinity gradient in Kongsfjorden in summer 2013 (Fransson et al., 2016), 2014 (Fransson et al., 2016), 2015 (van de Poll et al., 2018) and 2016 (Cantoni et al., 2019). Full depth data are shown, with a linear regression (black line) for glacially modified waters ($S < 34.2$) during summer 2016. The position of stations varies between the datasets, with the 2016 data providing the broadest coverage of the inner fjord. Linear regression details are shown in Table S1 in the Supplement. **(d)** Si, **(e)** NO$_3$ and **(f)** PO$_4$ distributions in surface waters of three major Arctic river estuaries: the Lena, Mackenzie and Yenisey (Cauwet and Sidorov, 1996; Emmerton et al., 2008; Hessen et al., 2010). Note the different $y$- and $x$-axis scales.

the concentration in the upper-mixed layer (Fig. 3b). For Si, a steady decline in Si with increasing salinity (Fig. 3a) is consistent with a discharge-associated Si supply (Brown et al., 2010; Arimitsu et al., 2016; Meire et al., 2016a). The spatial distribution of data for summer 2013–2016 is similar and representative of summertime conditions in the fjord (Hop et al., 2002).

Whilst dissolved macronutrient concentrations in glacial discharge are relatively low, a characteristic of glaciated catchments is extremely high particulate Fe concentrations. High Fe concentrations arise both directly from glacier discharge (Bhatia et al., 2013a; Hawkings et al., 2014) and also from resuspension of glacially derived sediments throughout the year (Markussen et al., 2016; Crusius et al., 2017). Total

dissolvable Fe (TdFe) concentrations within Godthåbsfjord are high in all available datasets (May 2014, August 2014 and July 2015) and strongly correlated with turbidity (linear regression: $R^2 = 0.88$, $R^2 = 0.56$ and $R^2 = 0.88$, respectively, Hopwood et al., 2016, 2018). A critical question in oceanography, in both the Arctic and Antarctic, is to what extent this large pool of particulate Fe is transferred into open-ocean environments and thus potentially able to affect marine primary production in Fe-limited offshore regions (Gerringa et al., 2012; Arrigo et al., 2017; Schlosser et al., 2018). The mechanisms that promote transfer of particulate Fe into bioavailable dissolved phases, such as ligand-mediated dissolution (Thuroczy et al., 2012) and biological activity (Schmidt et al., 2011), and the scavenging processes that return dissolved Fe to the particulate phase are both poorly characterized (Tagliabue et al., 2016).

Fe profiles around the Arctic show strong spatial variability in TdFe concentrations, ranging from unusually high concentrations of up to 20 µM found intermittently close to turbid glacial outflows (Zhang et al., 2015; Markussen et al., 2016; Hopwood et al., 2018) to generally low nanomolar concentrations at the interface between shelf and fjord waters (Zhang et al., 2015; Crusius et al., 2017; Cape et al., 2019). An interesting feature of some of these profiles around Greenland is the presence of peak Fe at ∼ 50 m depth, perhaps suggesting that much of the Fe transport away from glaciers may occur in subsurface turbid glacially modified waters (Hopwood et al., 2018; Cape et al., 2019). The spatial extent of Fe enrichment downstream of glaciers around the Arctic is still uncertain, but there is evidence of global variability downstream of glaciers on the scale of 10–100 km (Gerringa et al., 2012; Annett et al., 2017; Crusius et al., 2017).

## 4.1 Non-conservative mixing processes for Fe and Si

A key reason for uncertainty in the fate of glacially derived Fe is the non-conservative behaviour of dissolved Fe in saline waters. In the absence of biological processes (i.e. nutrient assimilation and remineralization), $NO_3$ is expected to exhibit conservative behaviour across estuarine salinity gradients (i.e. the concentration at any salinity is a linear function of mixing between fresh and saline waters). For Fe, however, a classic non-conservative estuarine behaviour occurs due to the removal of dissolved Fe (DFe[1]) as it flocculates and is absorbed onto particle surfaces more readily at higher salinity and pH (Boyle et al., 1977). Dissolved Fe concentrations almost invariably exhibit strong (typically ∼ 90 %) non-conservative removal across estuarine salinity gradients (Boyle et al., 1977; Sholkovitz et al., 1978), and glaciated catchments appear to be no exception to this rule (Lippiatt et al., 2010). Dissolved Fe in Godthåbsfjord exhibits a re-

[1]For consistency, dissolved Fe is defined throughout operationally as < 0.2 µm and is therefore inclusive of ionic, complexed and colloidal species.

moval of > 80 % DFe between salinities of 0–30 (Hopwood et al., 2016), and similar losses of approximately 98 % for Kongsfjorden and 85 % for the Copper river/estuary (Gulf of Alaska) system have been reported (Schroth et al., 2014; Zhang et al., 2015).

Conversely, Si can be released from particulate phases during estuarine mixing, resulting in non-conservative addition to dissolved Si concentrations (Windom et al., 1991), although salinity–Si relationships vary between different estuaries due to different extents of Si release from labile particulates and Si uptake by diatoms (e.g. Fig. 3d). Where evident, this release of dissolved Si typically occurs at low salinities (Cauwet and Sidorov, 1996; Emmerton et al., 2008; Hessen et al., 2010), with the behaviour of Si being more conservative at higher salinities and in estuaries where pronounced drawdown by diatoms is not evident (e.g. Brown et al., 2010). Estimating release of particulate Si from Kongsfjorden data (Fig. 3c) as the additional dissolved Si present above the conservative mixing line for runoff mixing with unmodified saline water that is entering the fjord (via linear regression) suggests a Si enrichment of 13 % ± 2 % (Fig. 3a). This is broadly consistent with the 6 %–53 % range reported for estuarine gradients evident in some temperate estuaries (Windom et al., 1991). Conversely, Hawkings et al. (2017) suggest a far greater dissolution downstream of Leverett Glacier, equivalent to a 70 %–800 % Si enrichment, and thus propose that the role of glaciers in the marine Si cycle has been underestimated. Given that such dissolution is substantially above the range observed in any other Arctic estuary, the apparent cause is worth further consideration.

---

**Bowdoin Fjord (NW Greenland) 78° N, 069° W**

Kangerluarsuup Sermia, also known as Bowdoin Fjord, is one of few glacier-fjord systems where biogeochemical and physical data are available in northern Greenland (Jouvet et al., 2018; Kanna et al., 2018). Bowdoin glacier, a small marine-terminating glacier at the fjord head, and four smaller land-terminating glaciers draining small ice caps isolated from the Greenland Ice Sheet, drain into the fjord, which is typically subject to sea-ice cover until July. The fjord is ~20 km long; the terminus of Bowdoin glacier is ~3 km wide.

---

The general distribution of Si in surface waters for Kongsfjorden (Fransson et al., 2016), Godthåbsfjord (Meire et al., 2016a), Bowdoin Fjord (Kanna et al., 2018), Sermilik (Cape et al., 2019) and along the Gulf of Alaska (Brown et al., 2010) is similar; Si shows pseudo-conservative behaviour declining with increasing salinity in surface waters. The limited reported number of zero-salinity, or very low salinity, endmembers for Godthåbsfjord and Bowdoin are significantly below the linear regression derived from surface nutrient and salinity data (Fig. 4). In addition to some dissolution of particulate Si, another likely reason for this is the limitation of individual zero-salinity measurements in dynamic fjord systems where different discharge outflows have different nutrient concentrations (Kanna et al., 2018), especially given that subglacial discharge is not directly characterized in either location (Meire et al., 2016a; Kanna et al., 2018). As

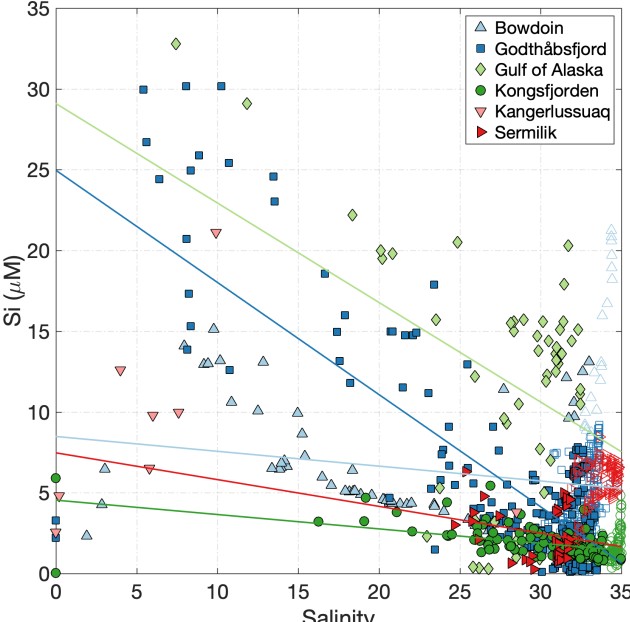

**Figure 4.** Dissolved Si distribution vs. salinity for glaciated Arctic catchments. Data are from Bowdoin Fjord (Kanna et al., 2018), Kongsfjorden (Fransson et al., 2016; van de Poll et al., 2018), Sermilik Fjord (Cape et al., 2019), Kangerlussuaq (Hawkings et al., 2017; Lund-Hansen et al., 2018), Godthåbsfjord (Hopwood et al., 2016; Meire et al., 2016b), and the Gulf of Alaska (Brown et al., 2010). Linear regressions are shown for large surface datasets only. Linear regression details are shown in Table S1. Closed markers indicate surface data (< 20 m depth), and open markers indicate subsurface data.

demonstrated by the two different zero-salinity Si endmembers in Kongsfjorden (iceberg melt of $\sim 0.03\,\mu M$ and surface runoff of $\sim 5.9\,\mu M$), pronounced deviations in nutrient content arise from mixing between various freshwater endmembers (surface runoff, ice melt and subglacial discharge). For example, total freshwater input into Godthåbsfjord is 70 %–80 % liquid, with this component consisting of 64 % ice sheet runoff, 31 % land runoff, and 5 % net precipitation (Langen et al., 2015) and being subject to additional inputs from iceberg melt along the fjord ($\sim 70\,\%$ of calved ice also melts within the inner fjord, Bendtsen et al., 2015).

In a marine context at broad scales, a single freshwater endmember that integrates the net contribution of all freshwater sources can be defined. This endmember includes iceberg melt, groundwater discharge, surface and subsurface glacier discharge, and (depending on location) sea-ice melt, which are challenging to distinguish in coastal waters (Benetti et al., 2019). Close to glaciers, it may be possible to observe distinct freshwater signatures in different water column layers and distinguish chemical signatures in water masses containing subglacial discharge from those containing primarily surface runoff and iceberg melt (e.g. in Godthåbsfjord, Meire et al., 2016a; and Sermilik, Beaird et

al., 2018), but this is often challenging due to mixing and overlap between different sources. Back-calculating the integrated freshwater endmember (e.g. from regression, Fig. 4) can potentially resolve the difficulty in accounting for data-deficient freshwater components and poorly characterized estuarine processes. As often noted in field studies, there is a general bias towards sampling of supraglacial meltwater and runoff in proglacial environments and a complete absence of chemical data for subglacial discharge emerging from large marine-terminating glaciers (e.g. Kanna et al., 2018).

Macronutrient distributions in Bowdoin, Godthåbsfjord and Sermilik unambiguously show that the primary macronutrient supply to surface waters associated with glacier discharge originates from mixing rather than from freshwater addition (Meire et al., 2016a; Kanna et al., 2018; Cape et al., 2019), which emphasizes the need to consider fjord inflow/outflow dynamics in order to interpret nutrient distributions. The apparently anomalous extent of Si dissolution downstream of Leverett Glacier (Hawkings et al., 2017) may therefore largely reflect underestimation of both the saline (assumed to be negligible) and freshwater endmembers rather than unusually prolific particulate Si dissolution. In any case, measured Si concentrations in the Kangerlussuaq region are within the range of other Arctic glacier estuaries (Fig. 4), making it challenging to support the hypothesis that glacial contributions to the Si cycle have been underestimated elsewhere (see also Tables 2 and 3).

## 4.2 Deriving glacier–ocean fluxes

In the discussion of macronutrients herein we have focused on the availability of the bioavailable species (e.g. $PO_4$, $NO_3$ and silicic acid) that control seasonal trends in inter-annual marine primary production (Juul-Pedersen et al., 2015; van de Poll et al., 2018; Holding et al., 2019). It should be noted that the total elemental fluxes (i.e. nitrogen, phosphorus and silicon) associated with lithogenic particles are invariably higher than the associated macronutrients (Wadham et al., 2019), particularly for phosphorus (Hawkings et al., 2016) and silicon (Hawkings et al., 2017). Lithogenic particles are however not bioavailable, although they may to some extent be bioaccessible, depending on the temporal and spatial scale involved. This is especially the case for the poorly quantified fraction of lithogenic particles that escapes sedimentation in inner-fjord environments, either directly or via resuspension of shallow sediments (Markussen et al., 2016; Hendry et al., 2019). It is hypothesized that lithogenic particle inputs from glaciers therefore have a positive influence on Arctic marine primary production (Wadham et al., 2019), yet field data to support this hypothesis are lacking. A pan-Arctic synthesis of all available primary production data for glaciated regions (Fig. 2 and Table 1), spatial patterns in productivity along the west Greenland coastline (Meire et al., 2017), population responses in glacier fjords across multiple taxonomic groups (Cauvy-Fraunié and Dangles, 2019) and sedimentary

records from Kongsfjorden (Kumar et al., 2018) consistently suggest that glaciers, or specifically increasing volumes of glacier discharge, have a net negative, or negligible, effect on marine primary producers – except in the specific case of some marine-terminating glaciers where a different mechanism seems to operate (see Sect. 5).

Two linked hypotheses can be proposed to explain these apparently contradictory arguments. One is that whilst lithogenic particles are potentially a bioaccessible source of Fe, P and Si, they are deficient in bioaccessible N. As $NO_3$ availability is expected to limit primary production across much of the Arctic (Tremblay et al., 2015), this creates a spatial mismatch between nutrient supply and the nutrient demand required to increase Arctic primary production. A related, alternative hypothesis is that the negative effects of discharge on marine primary production (e.g. via stratification and light limitation from high turbidity) more than offset any positive effect that lithogenic particles have via increasing nutrient availability on regional scales prior to extensive sedimentation occurring. A similar conclusion has been reached from analysis of primary production in proglacial streams (Uehlinger et al., 2010). To some extent this reconciliation is also supported by considering the relative magnitudes of different physical and chemical processes acting on different spatial scales with respect to global marine primary production (see Sect. 10).

The generally low concentrations of macronutrients and dissolved organic matter (DOM) in glacier discharge, relative to coastal seawater (Table 2), have an important methodological implication because what constitutes a positive $NO_3$, $PO_4$ or DOM flux into the Arctic Ocean in a glaciological context can actually reduce short-term nutrient availability in the marine environment. It is therefore necessary to consider both the glacier discharge and saline endmembers that mix in fjords, alongside fjord-scale circulation patterns, in order to constrain the change in nutrient availability to marine biota (Meire et al., 2016a; Hopwood et al., 2018; Kanna et al., 2018).

Despite the relatively well constrained nutrient signature of glacial discharge around the Arctic, estimated fluxes of some nutrients from glaciers to the ocean appear to be subject to greater variability, especially for nutrients subject to non-conservative mixing (Table 3). Estimates of the Fe flux from the Greenland Ice Sheet, for example, have an 11-fold difference between the lowest ($> 26\,\mathrm{Mmol\,yr^{-1}}$) and highest ($290\,\mathrm{Mmol\,yr^{-1}}$) values (Hawkings et al., 2014; Stevenson et al., 2017). However, it is debatable if these differences in Fe flux are significant because they largely arise in differences between definitions of the flux gate window and especially how estuarine Fe removal is accounted for. Given that the difference between an estimated removal factor of 90 % and 99 % is a factor of 10 difference in the calculated DFe flux, there is overlap in all of the calculated fluxes for Greenland Ice Sheet discharge into the ocean (Table 3) (Statham et al., 2008; Bhatia et al., 2013a; Hawkings et al., 2014; Stevenson

et al., 2017). Conversely, estimates of DOM export (quantified as DOC) are confined to a slightly narrower range of 7–40 Gmol yr$^{-1}$, with differences arising from changes in measured DOM concentrations (Bhatia et al., 2013b; Lawson et al., 2014b; Hood et al., 2015). The characterization of glacial DOM, with respect to its lability, C : N ratio and implications for bacterial productivity in the marine environment (Hood et al., 2015; Paulsen et al., 2017), is however not readily apparent from a simple flux calculation.

A scaled-up calculation using freshwater concentrations ($C$) and discharge volumes ($Q$) is the simplest way of determining the flux from a glaciated catchment to the ocean. However, discharge nutrient concentrations vary seasonally (Hawkings et al., 2016; Wadham et al., 2016), often resulting in variable $C-Q$ relationships due to changes in mixing ratios between different discharge flow paths; post-mixing reactions; and seasonal changes in microbial behaviour in the snowpack, on glacier surfaces, and in proglacial forefields (Brown et al., 1994; Hodson et al., 2005). Therefore, full seasonal datasets from a range of representative glaciers are required to accurately describe $C-Q$ relationships. Furthermore, as the indirect effects of discharge on nutrient availability to phytoplankton via estuarine circulation and stratification are expected to be a greater influence than the direct nutrient outflow associated with discharge (Rysgaard et al., 2003; Juul-Pedersen et al., 2015; Meire et al., 2016a), freshwater data must be coupled to physical and chemical time series in the coastal environment if the net effect of discharge on nutrient availability in the marine environment is to be understood. Indeed, the recently emphasized hypothesis that nutrient fluxes from glaciers into the ocean have been significantly underestimated (Hawkings et al., 2016, 2017; Wadham et al., 2016) is difficult to reconcile with a synthesis and analysis of available marine nutrient distributions (Sect. 4) in glaciated Arctic catchments, especially for Si (Fig. 4).

---

**Young Sound-Tyrolerfjord (NE Greenland)  74° N, 021° W**

Young Sound-Tyrolerfjord is a catchment fed by rivers from three land-terminating glaciers. Tyrolerfjord is the narrow innermost part of the fjord system in the west, and Young Sound is the wider outer part in the east towards the Atlantic Ocean. The fjord system has a surface area of 390 km$^2$, a length of 90 km and a maximum depth of 360 m. A shallow ~45 m deep sill restricts exchange with the Greenland shelf, and summertime productivity in the fjord is among the lowest measured in the Arctic (as low as <40 mg C m$^{-2}$ d$^{-1}$). In recent years, fjord waters have freshened (Sejr et al., 2017), and freshening of coastal waters has prevented renewal of fjord bottom waters (Boone et al., 2018). A data portal is available reporting work done in the catchment through the Greenland Ecosystem Monitoring Programme (GEM; http://g-e-m.dk).

---

A particularly interesting case study concerning the link between marine primary production, circulation and discharge-derived nutrient fluxes is Young Sound. It was initially stipulated that increasing discharge into the fjord in response to climate change would increase estuarine circulation and therefore macronutrient supply. Combined with a longer sea-ice-free growing season as Arctic temperatures

*Please note the remarks at the end of the manuscript.*

**Table 2.** Measured/computed discharge and saline endmembers for well-studied Arctic fjords (ND, not determined/not reported; BD, below detection).

| Fjord | Dataset | Salinity | $NO_3$ (μM) | $PO_4$ (μM) | Si (μM) | TdFe (μM) |
|---|---|---|---|---|---|---|
| Kongsfjorden (Svalbard) | Summer 2016 (Cantoni et al., 2019) | 0.0 (ice melt) | $0.87 \pm 1.0$ | $0.02 \pm 0.03$ | $0.03 \pm 0.03$ | $33.8 \pm 100$ |
| | | 0.0 (surface discharge) | $0.94 \pm 1.0$ | $0.057 \pm 0.31$ | $5.91 \pm 4.1$ | $74 \pm 76$ |
| | | $34.50 \pm 0.17$ | $1.25 \pm 0.49$ | $0.20 \pm 0.06$ | $1.00 \pm 0.33$ | ND |
| Nuup Kangerlua/ Godthåbsfjord (Greenland) | Summer 2014 (Hopwood et al., 2016; Meire et al., 2016) | 0.0 (ice melt) | $1.96 \pm 1.68$ | $0.04 \pm 0.04$ | ND | $0.31 \pm 0.49$ |
| | | 0.0 (surface discharge) | $1.60 \pm 0.44$ | $0.02 \pm 0.01$ | $12.2 \pm 16.3$ | 13.8 |
| | | $33.57 \pm 0.05$ | $11.5 \pm 1.5$ | $0.79 \pm 0.04$ | $8.0 \pm 1.0$ | ND TS1 |
| Sermilik (Greenland) | Summer 2015 (Cape et al., 2019) | 0.0 (subglacial discharge) | $1.8 \pm 0.5$ | ND | $10 \pm 8$ | ND |
| | | 0.0 (ice melt) | $0.97 \pm 1.5$ | ND | $4 \pm 4$ | ND |
| | | $34.9 \pm 0.1$ | $12.8 \pm 1$ | ND | $6.15 \pm 1$ | ND |
| Bowdoin (Greenland) | Summer 2016 (Kanna et al., 2018) | 0.0 (surface discharge) | $0.22 \pm 0.15$ | $0.30 \pm 0.20$ | BD | ND |
| | | $34.3 \pm 0.1$ | $14.7 \pm 0.9$ | $1.1 \pm 0.1$ | $19.5 \pm 1.5$ | ND |
| Young Sound (Greenland) | Summer 2014 (Paulsen et al., 2017) | (Runoff July–August) | $1.2 \pm 0.74$ | $0.29 \pm 0.2$ | $9.52 \pm 3.8$ | ND |
| | | (Runoff September–October) | $1.0 \pm 0.7$ | $0.35 \pm 0.2$ | $29.57 \pm 10.9$ | ND |
| | | $33.6 \pm 0.1$ (July–August) | $6.4 \pm 1.1$ | $1.18 \pm 0.5$ | $6.66 \pm 0.4$ | ND |
| | | $33.5 \pm 0.04$ (September–October) | $5.6 \pm 0.2$ | $0.62 \pm 0.2$ | $6.5 \pm 0.1$ | ND |

increase, this would be expected to increase primary production within the fjord (Rysgaard et al., 1999; Rysgaard and Glud, 2007). Yet freshwater input also stratifies the fjord throughout summer and ensures low macronutrient availability in surface waters (Bendtsen et al., 2014; Meire et al., 2016a), which results in low summertime productivity in the inner and central fjord ($< 40 \, \mathrm{mg\,C\,m^{-2}\,d^{-1}}$) (Rysgaard et al., 1999, 2003; Rysgaard and Glud, 2007). Whilst annual discharge volumes into the fjord have increased over the past two decades, resulting in a mean annual $0.12 \pm 0.05$ (practical salinity units) freshening of fjord waters (Sejr et al., 2017), shelf waters have also freshened. This has potentially impeded the dense inflow of saline waters into the fjord (Boone et al., 2018) and therefore counteracted the expected increase in productivity.

### 4.3 How do variations in the behaviour and location of higher-trophic-level organisms affect nutrient availability to marine microorganisms?

With the exception of some zooplankton and fish species that struggle to adapt to the strong salinity gradients and/or suspended particle loads in inner-fjord environments (Wçslawski and Legezytńska, 1998; Lydersen et al., 2014), higher-trophic-level organisms (including mammals and birds) are not directly affected by the physical/chemical gradients caused by glacier discharge. However, their food sources, such as zooplankton and some fish species, are directly affected, and therefore there are many examples of higher-level organisms adapting their feeding strategies within glacier fjord environments (Arimitsu et al., 2012; Renner et al., 2012; Laidre et al., 2016). Strong gradients in physical/chemical gradients downstream of glaciers, particularly turbidity, can therefore create localized hotspots of secondary productivity in areas where primary production is low (Lydersen et al., 2014).

It is debatable to what extent shifts in these feeding patterns could have broadscale biogeochemical effects. Whilst some species are widely described as ecosystem engineers, such as *Alle alle* (the little auk) in the Greenland North Water Polynya (González-Bergonzoni et al., 2017), for changes in higher-trophic-level organisms' feeding habits to have significant direct chemical effects on the scale of a glacier fjord system would require relatively large concentrations of such animals. Nevertheless, in some specific hotspot regions this effect is significant enough to be measurable. There is ample evidence that birds intentionally target upwelling plumes in front of glaciers as feeding grounds, possibly due to the stunning effect that turbid, upwelling plumes have upon prey such as zooplankton (Hop et al., 2002; Lydersen et al., 2014). This feeding activity therefore concentrates the effect of avian nutrient recycling within a smaller area than would otherwise be the case, potentially leading to modest nutrient enrichment of these proglacial environments. Yet, with the exception of large, concentrated bird colonies, the effects of such activity are likely modest. In Kongsfjorden, bird populations are well studied, and several species are associated with feeding in proglacial plumes yet still collectively consume only between 0.1 % and 5.3 % of the carbon produced by phytoplankton in the fjord (Hop et al., 2002). The estimated corresponding nutrient flux into the fjord from birds is $2 \, \mathrm{mmol\,m^{-2}\,yr^{-1}}$ nitrogen and $0.3 \, \mathrm{mmol\,m^{-2}\,yr^{-1}}$ phosphorous.

**Table 3.** Flux calculations for dissolved nutrients (Fe, DOC, DON, NO$_3$, PO$_4$ and Si) from Greenland Ice Sheet discharge. Where a flux was not calculated in the original work, an assumed discharge volume of 1000 km$^3$ yr$^{-1}$ is used to derive a flux for comparative purposes (ASi, amorphous silica; LPP, labile particulate phosphorous). For DOM, PO$_4$ and NO$_3$, non-conservative estuarine behaviour is expected to be minor or negligible. Note that whilst we have defined "dissolved" herein as $< 0.2$ μm, the sampling and filtration techniques used, particularly in freshwater studies, are not well standardized, and thus some differences may arise between studies accordingly. Clogging of filters in turbid waters reduces the effective filter pore size; DOP, DON, NH$_4$ and PO$_4$ concentrations often approach analytical detection limits which, alongside field/analytical blanks, are treated differently; low concentrations of NO$_3$, DON, DOP, DOC, NH$_4$ and DFe are easily inadvertently introduced to samples by contamination, and measured Si concentrations can be significantly lower when samples have been frozen.

| Nutrient | Freshwater endmember concentration (μM) | Flux | Estuarine modification | Data |
|---|---|---|---|---|
| Fe | 0.13 | $> 26$ Mmol yr$^{-1}$ | Inclusive, $> 80\%$ loss | Hopwood et al. (2016) |
| | 1.64 | 39 Mmol yr$^{-1}$ | Assumed 90 % loss | Stevenson et al. (2017) |
| | 0.053 | 53 Mmol yr$^{-1}$ | Discussed, not applied | Statham et al. (2008) |
| | 3.70 | 180 Mmol yr$^{-1}$ | Assumed 90 % loss | Bhatia et al. (2013a) |
| | 0.71 | 290 Mmol yr$^{-1}$ | Discussed, not applied | Hawkings et al. (2014) |
| DOC | 16–100 | 6.7 Gmol yr$^{-1}$ | Not discussed | Bhatia et al. (2010, 2013b) |
| | 12–41 | 11–14 Gmol yr$^{-1}$ | Not discussed | Lawson et al. (2014b) |
| | 15–100 | 18 Gmol yr$^{-1}$ | Not discussed | Hood et al. (2015) |
| | 2–290 | 24–38 Gmol yr$^{-1}$ | Not discussed | Csank et al. (2019) |
| | 27–47 | 40 Gmol yr$^{-1}$ | Not discussed | Paulsen et al. (2017) |
| DON | 2.3 | 2.3 Gmol yr$^{-1}$ | Not discussed | Wadham et al. (2016) |
| | 4.7–5.4 | 5 Gmol yr$^{-1}$ | Not discussed | Paulsen et al. (2017) |
| | 1.7 | 0.7–1.1 Gmol yr$^{-1}$ | Not discussed | Wadham et al. (2016) |
| Si | 13–28 | 22 Gmol yr$^{-1}$ | Inclusive | Meire et al. (2016a) |
| | 9.6 | 4 Gmol yr$^{-1}$ | Discussed (+190 Gmol yr$^{-1}$ ASi) | Hawkings et al. (2017) |
| PO$_4$ | 0.23 | 0.10 Gmol yr$^{-1}$ | Discussed (+0.23 Gmol yr$^{-1}$ LPP) | Hawkings et al. (2016) TS2 |
| | 0.26 | 0.26 Gmol yr$^{-1}$ | Not discussed | Meire et al. (2016a) |
| NO$_3$ | 1.4–1.5 | 0.42 Gmol yr$^{-1}$ | Not discussed | Wadham et al. (2016) |
| | 0.5–1.7 | 0.5–1.7 Gmol yr$^{-1}$ | Not discussed | Paulsen et al. (2017) |
| | 1.79 | 1.79 Gmol yr$^{-1}$ | Not discussed | Meire et al. (2016a) |

## 5 Critical differences between surface and subsurface discharge release

**Sermilik Fjord (SE Greenland) 66° N, 038° W**

Sermilik Fjord is home to Helheim Glacier, Greenland's fifth largest in terms of annual discharge volume. The fjord is ~100 km long and ~600–900 m deep, with no sill to restrict fjord–shelf exchange. The circulation of water masses within the fjord, fjord–shelf exchange (Straneo et al., 2011; Beaird et al., 2018), and iceberg dynamics along the fjord have all been characterised. Whilst a large fraction (40–60%) of freshwater from Greenland enters the ocean as solid ice, rather than as meltwater discharge, surprisingly little is known about the fate and effects of this component in the marine environment (Sutherland et al., 2014; Enderlin et al., 2018; Moon et al., 2018).

Critical differences arise between land-terminating and marine-terminating glaciers with respect to their effects on water column structure and associated patterns in primary production (Table 1). Multiple glacier fjord surveys have shown that fjords with large marine-terminating glaciers around the Arctic are normally more productive than their land-terminating glacier fjord counterparts (Meire et al., 2017; Kanna et al., 2018), and, despite large inter-fjord variability (Fig. 2), this observation appears to be significant across all available primary production data for Arctic glacier fjords (Table 1). A particularly critical insight is that fjord-scale summertime productivity along the west Greenland coastline scales approximately with discharge downstream of marine-terminating glaciers but not land-terminating glaciers (Meire et al., 2017). The primary explanation for this phenomenon is the vertical nutrient flux associated with mixing

driven by subglacial discharge plumes, which has been quantified in field studies at Bowdoin glacier (Kanna et al., 2018), Sermilik Fjord (Cape et al., 2019), Kongsfjorden (Halbach et al., 2019) and in Godthåbsfjord (Meire et al., 2016a).

As discharge is released at the glacial grounding line depth, its buoyancy and momentum result in an upwelling plume that entrains and mixes with ambient seawater (Carroll et al., 2015, 2016; Cowton et al., 2015). In Bowdoin, Sermilik and Godthåbsfjord, this nutrient pump provides 99 %, 97 % and 87 %, respectively, of the $NO_3$ associated with glacier inputs to each fjord system (Meire et al., 2016a; Kanna et al., 2018; Cape et al., 2019). Whilst the pan-Arctic magnitude of this nutrient pump is challenging to quantify because of the uniqueness of glacier fjord systems in terms of their geometry, circulation, residence time and glacier grounding line depths (Straneo and Cenedese, 2015; Morlighem et al., 2017), it can be approximated in generic terms because plume theory (Morton et al., 1956) has been used extensively to describe subglacial discharge plumes in the marine environment (Jenkins, 2011; Hewitt, 2020). Computed estimates of subglacial discharge for the 12 Greenland glacier fjord systems where sufficient data are available to simulate plume entrainment (Carroll et al., 2016) suggest that the entrainment effect is at least 2 orders of magnitude more important for macronutrient availability than direct freshwater runoff (Hopwood et al., 2018). This is consistent with limited available field observations (Meire et al., 2016a; Kanna et al., 2018; Cape et al., 2019). As macronutrient fluxes have been estimated independently using different datasets and plume entrainment models in two of these glacier fjord systems (Sermilik and Illulissat), an assessment of the robustness of these fluxes can also be made (Table 4) (Hopwood et al., 2018; Cape et al., 2019). Exactly how these plumes, and any associated fluxes, will change with the combined effects of glacier retreat and increasing glacier discharge remains unclear (De Andrés et al., 2020) but may lead to large changes in fjord biogeochemistry (Torsvik et al., 2019). Despite different definitions of the macronutrient flux (Table 4; "A" refers to the out-of-fjord transport at a defined fjord cross-section window, whereas "B" refers to the vertical transport within the immediate vicinity of the glacier), the fluxes are reasonably comparable and in both cases unambiguously dominate macronutrient glacier-associated input into these fjord systems (Hopwood et al., 2018; Cape et al., 2019).

Whilst large compared to changes in macronutrient availability from discharge without entrainment (Table 3), it should be noted that these nutrient fluxes (Table 4) are still only intermediate contributions to fjord-scale macronutrient supply compared to total annual consumption in these environments. For example, in Godthåbsfjord mean annual primary production is $103.7\,\mathrm{g\,C\,m^{-2}\,yr^{-1}}$, equivalent to biological consumption of $1.1\,\mathrm{mol\,N\,m^{-2}\,yr^{-1}}$. Entrainment from the three marine-terminating glaciers within the fjord is conservatively estimated to supply $0.01$–$0.12\,\mathrm{mol\,N\,m^{-2}\,yr^{-1}}$

(Meire et al., 2017), i.e. 1 %–11 % of the total N supply required for primary production if production were supported exclusively by new $NO_3$ (rather than recycling) and equally distributed across the entire fjord surface. Whilst this is consistent with observations suggesting relative stability in mean annual primary production in Godthåbsfjord from 2005 to 2012 ($103.7\pm17.8\,\mathrm{g\,C\,m^{-2}\,yr^{-1}}$; Juul-Pedersen et al., 2015), despite pronounced increases in total discharge into the fjord, this does not preclude a much stronger influence of entrainment on primary production in the inner-fjord environment. The time series is constructed at the fjord mouth, over 120 km from the nearest glacier, and the estimates of subglacial discharge and entrainment used by Meire et al. (2017) are both unrealistically low. If the same conservative estimate of entrainment is assumed to only affect productivity in the main fjord branch (where the three marine-terminating glaciers are located), for example, the lower bound for the contribution of entrainment becomes 3 %–33 % of total N supply. Similarly, in Kongsfjorden – the surface area of which is considerably smaller compared to Godthåbsfjord ($\sim 230\,\mathrm{km^2}$ compared to $650\,\mathrm{km^2}$) – even the relatively weak entrainment from shallow marine-terminating glaciers (Fig. 5) accounts for approximately 19 %–32 % of N supply. An additional mechanism of N supply evident there, which partially offsets the inefficiency of macronutrient entrainment at shallow grounding line depths, is the entrainment of ammonium from shallow benthic sources (Halbach et al., 2019), which leads to unusually high $NH_4$ concentrations in surface waters. Changes in subglacial discharge, or in the entrainment factor (e.g. from a shift in glacier grounding line depth, Carroll et al., 2016), can therefore potentially change fjord-scale productivity.

A specific deficiency in the literature to date is the absence of measured subglacial discharge rates from marine-terminating glaciers. Variability in such rates on diurnal and seasonal timescales is expected (Schild et al., 2016; Fried et al., 2018), and intermittent periods of extremely high discharge are known to occur, for example from ice-dammed lake drainage in Godthåbsfjord (Kjeldsen et al., 2014). Yet determining the extent to which these events affect fjord-scale mixing and biogeochemistry, as well as how these rates change in response to climate forcing, will require further field observations. Paradoxically, one of the major knowledge gaps concerning low-frequency, high-discharge events is their biological effects; yet these events first became characterized in Godthåbsfjord after observations by a fisherman of a sudden *Sebastes marinus* (Redfish) mortality event in the vicinity of a marine-terminating glacier terminus. These unfortunate fish were propelled rapidly to the surface by ascending freshwater during a high-discharge event (Kjeldsen et al., 2014).

A further deficiency, yet to be specifically addressed in biogeochemical studies, is the decoupling of different mixing processes in glacier fjords. In this section we have primarily considered the effect of subglacial discharge plumes on $NO_3$ supply to near-surface waters downstream of marine-

**Table 4.** A comparison of upwelled $NO_3$ fluxes calculated from fjord-specific observed nutrient distributions (A) (Cape et al., 2019) and using regional nutrient profiles with idealized plume theory (B) (Hopwood et al., 2018). "A" refers to the out-of-fjord transport of nutrients, whereas "B" refers to the vertical transport close to the glacier terminus.

| Location | Field campaign(s) for A | (A) Calculated out-of-fjord $NO_3$ export Gmol yr$^{-1}$ | (B) Idealized $NO_3$ upwelling Gmol yr$^{-1}$ |
| --- | --- | --- | --- |
| Ilulissat Icefjord (Jakobshavn Isbræ) | 2000–2016 | $2.9 \pm 0.9$ | 4.2 |
| Sermilik (Helheim Glacier) | 2015 | 0.88 | 2.0 |
| Sermilik (Helheim Glacier) | 2000–2016 | $1.2 \pm 0.3$ | |

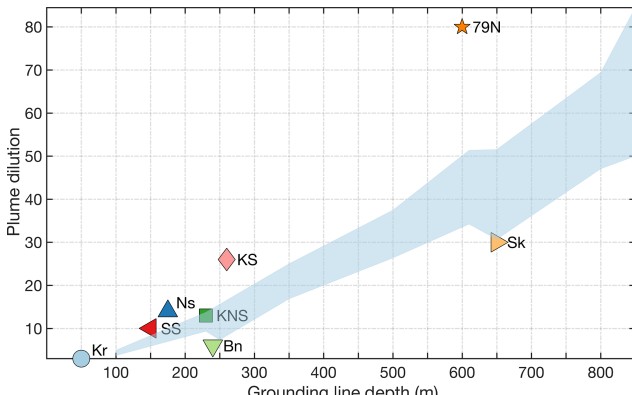

**Figure 5.** The plume dilution (entrainment) factor relationship with glacier grounding line depth as modelled by Carroll et al. (2016) for subglacial freshwater discharge rates of 250–500 m$^3$ s$^{-1}$ and grounding lines of $> 100$ m (shaded area). Also shown are the entrainment factors determined from field observations for Kronebreen (Kongsfjorden, Kr, Halbach et al., 2019), Bowdoin (Bn, Kanna et al., 2018), Saqqarliup Sermia (SS, Mankoff et al., 2016), Narsap Sermia (Ns, Meire et al., 2016a), Kangerlussuup Sermia (KS, Jackson et al., 2017), Kangiata Nunaata Sermia (KNS, Bendtsen et al., 2015), Sermilik (Sk, Beaird et al., 2018) and Nioghalvfjerdsfjorden Glacier (the "79° N Glacier", 79N, Schaffer et al., 2020). Note that the 79° N Glacier is unusual compared to the other Arctic systems displayed as subglacial discharge there enters a large cavity beneath a floating ice tongue and accounts for only 11 % of meltwater entering this cavity, with the rest derived from basal ice melt (Schaffer et al., 2020).

terminating glaciers (Fig. 5). Yet a similar effect can arise from down-fjord katabatic winds which facilitate the out-of-fjord transport of low-salinity surface waters and the inflow of generally macronutrient-rich saline waters at depth (Svendsen et al., 2002; Johnson et al., 2011; Spall et al., 2017). Both subglacial discharge and down-fjord winds therefore contribute to physical changes affecting macronutrient availability on a similar spatial scale, and both processes are expected to be subject to substantial short-term (hours-days), seasonal and inter-fjord variability, which is presently poorly constrained (Spall et al., 2017; Sundfjord et al., 2017).

### 5.1 Is benthic–pelagic coupling enhanced by subglacial discharge?

The attribution of unusually high near-surface $NH_4$ concentrations in surface waters of Kongsfjorden to benthic release in this relatively shallow fjord, followed by upwelling close to the Kronebreen calving front (Halbach et al., 2019), raises questions about where else this phenomenon could be important and which other biogeochemical compounds could be made available to pelagic organisms by such enhanced benthic–pelagic coupling. The summertime discharge-driven upwelling flux within a glacier fjord of any chemical which is released into bottom water from sediments, for example Fe, Mn (Wehrmann et al., 2014), dissolved organic phosphorous (DOP), dissolved organic nitrogen (DON) (Koziorowska et al., 2018) or Si (Hendry et al., 2019), could potentially be increased to varying degrees depending on sediment composition (Wehrmann et al., 2014; Glud et al., 2000) and the interrelated nature of fjord circulation, topography and the depth range over which entrainment occurs.

Where such benthic upwelling coupling does occur close to glacier termini it may be challenging to quantify from water column observations due to the overlap with other processes causing nutrient enrichment. For example, the moderately high dissolved Fe concentrations observed close to Antarctic ice shelves were classically attributed mainly to direct freshwater inputs, but it is now thought that the direct freshwater input and the Fe entering surface waters from entrainment of Fe-enriched near-bottom waters could be comparable in magnitude (St-Laurent et al., 2017), although with large uncertainty. This adds further complexity to the role of coastal, fjord and glacier geometry in controlling nutrient bioaccessibility, and determining the significance of such coupling is a priority for hybrid model–field studies.

### 5.2 From pelagic primary production to the carbon sink

Whilst primary production is a major driver of $CO_2$ drawdown from the atmosphere to the surface ocean, much of this C is subject to remineralization and, following bacterial or photochemical degradation of organic carbon, re-enters the

atmosphere as $CO_2$ on short timescales. The biological C pump refers to the small fraction of sinking C which is sequestered in the deep ocean or in sediments. There is no simple relationship between primary production and C export into the deep ocean as a range of primary-production–C-export relationships have been derived globally with the underlying cause subject to ongoing discussion (Le Moigne et al., 2016; Henson et al., 2019).

Irrespective of global patterns, glacier fjords are notable for their extremely high rates of sedimentation due to high lithogenic particle inputs (Howe et al., 2010). In addition to terrestrially derived material providing additional organic carbon for burial in fjords (Table 3), ballasting of sinking POC (particulate organic carbon) by lithogenic material generally increases the efficiency of the biological C pump by facilitating more rapid transfer of C to depth (Iversen and Robert, 2015; Pabortsava et al., 2017). With high sediment loads and steep topography, fjords are therefore expected to be efficient POC sinks, especially when normalized with respect to their surface area (Smith et al., 2015). Organic carbon accumulation rates in Arctic glacier fjords are far lower than temperate fjord systems, likely due to a combination of generally lower terrestrially derived carbon inputs and sometimes lower marine primary production, but Arctic fjords with glaciers still exhibit higher C accumulation than Arctic fjords without glaciers (Włodarska-Kowalczuk et al., 2019).

The limited available POC fluxes for Arctic glacier fjords support the hypothesis that they are efficient regions of POC export (Wiedmann et al., 2016; Seifert et al., 2019). POC equivalent to 28 %–82 % of primary production was found to be transferred to > 100 m depth in Nordvestfjord (west Greenland) (Seifert et al., 2019). This represents medium-to-high export efficiency compared to other marine environments on a global scale (Henson et al., 2019). High lithogenic particle inputs into Arctic glacier fjords could therefore be considered to maintain a low-primary-production–high-C-export-efficiency regime. On the one hand, they limit light availability and thus contribute to relatively low levels of primary production (Table 1), but concurrently they ensure that a relatively high fraction of C fixed by primary producers is transferred to depth (Seifert et al., 2019).

Beyond the potent impact of high sedimentation on benthic ecosystems (Włodarska-Kowalczuk et al., 2001, 2005), which is beyond the scope of this review, and the ballasting effect, which is sparsely studied in this environment to date (Seifert et al., 2019), relatively little is known about the interactive effects of concurrent biogeochemical processes on glacier-derived particle surfaces occurring during their suspension (or resuspension) in near-shore waters. Chemical processes occurring at turbid freshwater–saline interfaces such as dissolved Fe and DOM scavenging onto particle surfaces and phosphate or DOM co-precipitation with Fe oxyhydroxides (e.g. Sholkovitz et al., 1978; Charette and Sholkovitz, 2002; Hyacinthe and Van Cappellen, 2004) have yet to be extensively studied in Arctic glacier estuaries where

they may exert some influence on nutrient availability and C cycling.

## 6 Contrasting Fe- and $NO_3$-limited regions of the ocean

Whether or not nutrients transported to the ocean surface have an immediate positive effect on marine primary production depends on the identity of the resource(s) that limits marine primary production. Light attenuation is the ultimate limiting control on marine primary production and is exacerbated close to turbid glacial outflows (Hop et al., 2002; Arimitsu et al., 2012; Murray et al., 2015). However the spatial extent of sediment plumes and/or ice mélange, which limit light penetration into the water column, is typically restricted to within kilometres of the glacier terminus (Arimitsu et al., 2012; Hudson et al., 2014; Lydersen et al., 2014). Beyond the turbid, light-limited vicinity of glacial outflows, the proximal limiting resource for summertime marine primary production will likely be a nutrient, the identity of which varies with location globally (Moore et al., 2013). Increasing the supply of the proximal limiting nutrient would be expected to have a positive influence on marine primary production, whereas increasing the supply of other nutrients alone would not – a premise of "the law of the minimum" (Debaar, 1994). Although proximal limiting nutrient availability controls total primary production, organic carbon and nutrient stoichiometry nevertheless has specific effects on the predominance of different phytoplankton and bacterial groups (Egge and Aksnes, 1992; Egge and Heimdal, 1994; Thingstad et al., 2008).

The continental shelf is a major source of Fe into the ocean (Lam and Bishop, 2008; Charette et al., 2016), and this results in clear differences in proximal limiting nutrients between Arctic and Antarctic marine environments. The isolated Southern Ocean is the world's largest high-nitrate, low-chlorophyll (HNLC) zone where Fe extensively limits primary production even in coastal polynyas (Sedwick et al., 2011) and macronutrients are generally present at high concentrations in surface waters (Martin et al., 1990a, b). Conversely, the Arctic Ocean is exposed to extensive broad shelf areas with associated Fe input from rivers and shelf sediments and thus generally has a greater availability of Fe relative to macronutrient supply (Klunder et al., 2012). Fe-limited summertime conditions have been reported in parts of the Arctic and sub-Arctic (Nielsdottir et al., 2009; Ryan-Keogh et al., 2013; Rijkenberg et al., 2018) but are spatially and temporally limited compared to the geographically extensive HNLC conditions in the Southern Ocean.

However, few experimental studies have directly assessed the nutrient limitation status of regions within the vicinity of glaciated Arctic catchments. With extremely high Fe input into these catchments, $NO_3$ limitation might be expected year-round. However, $PO_4$ limitation is also plausible close to glaciers in strongly stratified fjords (Prado-Fiedler, 2009), due to the low availability of $PO_4$ in freshwater rel-

ative to NO$_3$ (Ren et al., 2019). Conversely, in the Southern Ocean, it is possible that Fe-limited conditions occur extremely close to glaciers and ice shelves (Fig. 6). High-NO$_3$, low-Fe water can be found in the immediate vicinity of Antarctica's coastline (Gerringa et al., 2012; Marsay et al., 2017) and even in inshore bays (Annett et al., 2015; Höfer et al., 2019). Macronutrient data from Maxwell Bay (King George Island, South Shetland Islands), for example, suggest that Fe from local glaciers mixes with high-NO$_3$, high-Si ocean waters, providing ideal conditions for phytoplankton blooms in terms of nutrient availability. The lowest surface macronutrient concentrations measured in Maxwell Bay in a summer campaign were 17 µM NO$_3$, 1.4 µM PO$_4$ and 47 µM Si (Höfer et al., 2019). Similarly, in Ryder Bay (Antarctic Peninsula), the lowest measured annual macronutrient concentrations – occurring after strong drawdown during a pronounced phytoplankton bloom (22 mg m$^{-3}$ chlorophyll $a$) – were 2.5 µM NO$_3$ and 0.4 µM PO$_4$ (Annett et al., 2015). This contrasts starkly with the summertime surface macronutrient distribution in glaciated fjords in the Arctic, including Kongsfjorden (Fig. 3), where surface macronutrient concentrations are typically depleted throughout summer. These differences may explain why some Antarctic glacier fjords have significantly higher chlorophyll and biomass than any of the Arctic glacier fjord systems considered herein (Mascioni et al., 2019). However, we note a general lack of seasonal and interannual data for Antarctic glacier fjord systems precludes a comprehensive inter-comparison of these different systems.

For a hypothetical nutrient flux from a glacier, the same flux could be envisaged in two endmember scenarios: one several kilometres inside an Arctic fjord (e.g. Godthåbsfjord or Kongsfjorden); and one at the coastline of an isolated Southern Ocean island such as the Kerguelen (Bucciarelli et al., 2001; Bowie et al., 2015), Heard (van der Merwe et al., 2019) or South Shetland Islands (Höfer et al., 2019). In the Arctic fjord, a pronounced Fe flux from summertime discharge would likely have no immediate positive effect upon fjord-scale marine primary production because Fe may already be replete (Hopwood et al., 2016; Crusius et al., 2017). This is consistent with the observation that Fe-rich discharge from land-terminating glaciers around west Greenland does not have a positive fjord-scale fertilization effect (Meire et al., 2017) and may possibly be associated with a negative effect (Table 1). Conversely, the same Fe input into coastal waters around the Kerguelen Islands would be expected to have a pronounced positive effect upon marine primary production, because the islands occur within the world's largest HNLC zone. Where Fe is advected offshore in the wake of the islands, a general positive effect on primary production is expected (Blain et al., 2001; Bucciarelli et al., 2001) even though there are marked changes in the phytoplankton community composition between the Fe-enriched bloom region (dominated by microphytoplankton) and the offshore HNLC area (dominated by small diatoms and nanoflagellates) (Uitz

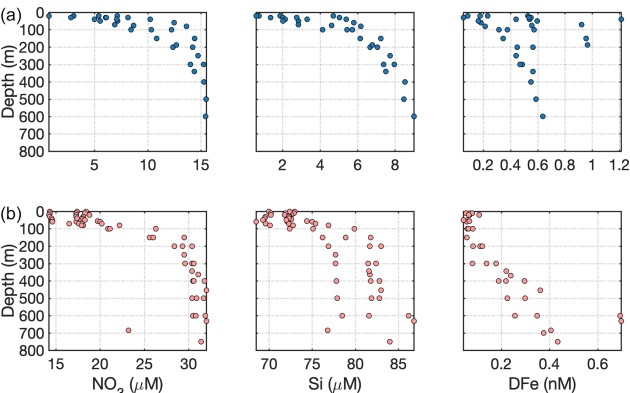

**Figure 6.** Contrasting nutrient properties of water on the **(a)** southeast Greenland shelf (data from Achterberg et al., 2018) with **(b)** the Ross Sea shelf (data from Marsay et al., 2017). Note the different scales used on the $x$ axes.

et al., 2009). However, even in these HNLC waters there are also other concurrent factors that locally mitigate the effect of glacially derived Fe in nearshore waters, because light limitation from near-surface particle plumes may locally offset any positive effect of Fe fertilization (Wojtasiewicz et al., 2019).

### 6.1 The subglacial discharge pump; from macronutrients to iron

The effect of the subglacial discharge nutrient pump may similarly vary with location. Contrasting the NO$_3$ and DFe concentrations of marine environments observed adjacent to different glacier systems suggests substantial variations in the proximal limiting nutrient of these waters on a global scale (Fig. 7). In Antarctic shelf regions, such as the western Antarctic Peninsula, a high log-transformed ratio of summertime NO$_3$ : DFe (median value 2) is indicative of Fe limitation. Across the Arctic there is a broader range of ratios (median values $-1.2$ to 1.3) indicating spatial variability in the balance between Fe and NO$_3$ limitation (Fig. 7). Variation is evident even within specific regions. The range of NO$_3$ : DFe ratios for both the Gulf of Alaska ($\log_{10} -2.5$ to 1.7) and the south Greenland shelf ($\log_{10} -1.5$ to 1.8) includes values that are indicative of the full spectrum of responses from NO$_3$ limitation to Fe/NO$_3$ co-limitation to Fe limitation (Browning et al., 2017). This suggests a relatively rapid spatial transition from excess to deficient DFe conditions.

How would the marine-terminating glacier upwelling effect operate in an Fe-limited system? The physical mechanism of a nutrient pump would be identical for glaciers with the same discharge and grounding line: one in a high-Fe, low-NO$_3$ Arctic system and one in a low-Fe, high-NO$_3$ Antarctic system. However, the biogeochemical consequences with respect to marine primary production would be different (Table 5). In the case of subglacial discharge, for simplicity, we consider a mid-depth glacier (grounding line of 100–250 m

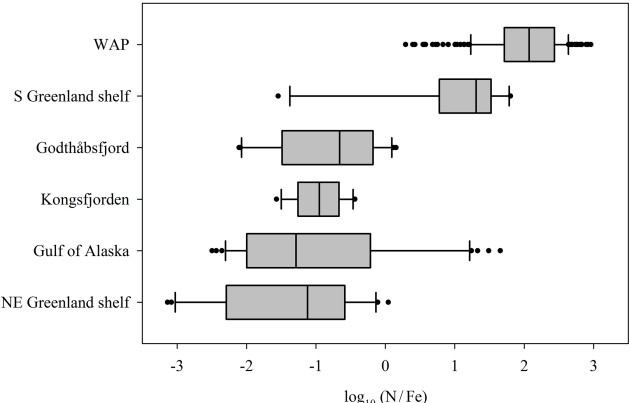

**Figure 7.** Variations in the ratio of dissolved $NO_3$ and Fe in surface waters ($< 20$ m) adjacent to glaciated regions: whiskers show the 10th and 90th percentiles; bars shows the median, 25th percentile and 75th percentile; and dots show all outliers. Data from the western Antarctic Peninsula (WAP, Annett et al., 2017; Ducklow et al., 2017), the south Greenland shelf (Achterberg et al., 2018; Tonnard et al., 2020), Godthåbsfjord (Hopwood et al., 2016), Kongsfjorden (Hopwood et al., 2017), the Gulf of Alaska (Lippiatt et al., 2010) and the NE Greenland shelf (Hopwood et al., 2018). For Kongsfjorden, $NO_3$ and Fe data were interpolated using the $NO_3$–salinity relationship.

below sea level) with a constant discharge rate of 250 $m^3 s^{-1}$. An entrainment factor of 6–10 would then be predicted by plume theory (Fig. 5) (Carroll et al., 2016). In a Greenland fjord with no sill to constrain circulation and a residence time short enough that inflowing nutrient concentrations were not changed significantly prior to entrainment, an average $NO_3$ concentration of 5–12 μM is predicted in the entrained water compared to $\sim 2$ μM in glacier discharge (Hopwood et al., 2018). Over a 2-month discharge period, this would produce a $NO_3$ flux of 40–160 Mmol $NO_3$, with 2 %–6 % of the $NO_3$ flux arising from meltwater discharge and 94 %–98 % from plume entrainment. Complete utilization of this $NO_3$ by phytoplankton according to the Redfield ratio (106 C : 16 N) (Redfield, 1934) would correspond to a biological sink of 0.27–1.0 Gmol C.

In an analogous HNLC environment, surface $NO_3$ requirements would already vastly exceed phytoplankton requirements (Fig. 7) due to extensive Fe limitation of primary production. Thus, whilst the upwelled $NO_3$ flux would be larger in an Fe-limited system, due to higher concentrations of $NO_3$ in the water column (see Fig. 6), the short-term biological effect of upwelling $NO_3$ alone would be negligible. More important would be the upwelling of the proximal limiting nutrient Fe. If we assume that dissolved Fe in the marine water column is in a stable, bioavailable form and that additional dissolved Fe from freshwater is delivered to the marine environment with a 90 %–99 % loss during estuarine mixing (Table 3), the upwelled Fe flux can be estimated. Upwelled unmodified water from a depth of 100–250 m would be ex-

pected to contain 0.06–0.12 nM Fe (Marsay et al., 2017). The freshwater endmember in the context of an Antarctic calving ice front would largely consist of ice melt (rather than subglacial discharge, Hewitt, 2020), so we use an intermediate freshwater Fe endmember of 33–680 nM in ice melt (Annett et al., 2017; Hodson et al., 2017). Upwelling via the same hypothetical 250 $m^3 s^{-1}$ discharge as per the Arctic scenario would generate a combined upwelled and discharge flux (after estuarine removal processes) of 0.89–89 kmol Fe with 2 %–52 % of the Fe arising from upwelling and 48 %–98 % from freshwater. Using an intermediate Fe : C value of 5 mmol Fe $mol^{-1}$C, which is broadly applicable to the coastal environment (Twining and Baines, 2013), this would correspond to a biological pool of 0.019–1.9 Gmol C. It should be noted that the uncertainty on this calculation is particularly large because, unlike $NO_3$ upwelling, there is a lack of in situ data to constrain the simultaneous mixing and non-conservative behaviour of Fe.

For a surface discharge of 250 $m^3 s^{-1}$, nutrient entrainment is assumed to be negligible. In the case of Fe outflow into a low-Fe, high-$NO_3$ system, we assume that the glacier outflow is the dominant local Fe source over the fertilized area during the discharge period (i.e. changes to other sources of Fe such as the diffusive flux from shelf sediments are negligible). For the case of surface discharge into a low-$NO_3$, high-Fe system, this is not likely to be the case for $NO_3$. Stratification induced by discharge decreases the vertical flux of $NO_3$ from below, thus negatively affecting $NO_3$ supply, although there are to our knowledge no studies quantifying this change in glacially modified waters.

It is clear from these simplified discharge scenarios (Table 5) that both the depth at which glacier discharge is released into the water column and the relative availabilities of $NO_3$ and Fe in downstream waters could be critical for determining the response of primary producers. The response of primary producers in low-Fe regimes is notably subject to much larger uncertainty, mainly because of uncertainty in the extent of Fe removal during estuarine mixing (Schroth et al., 2014; Zhang et al., 2015). Whilst the effects of the marine-terminating glacier nutrient pump on macronutrient fluxes have been defined in numerous systems, its effect on Fe availability is poorly constrained (Gerringa et al., 2012; St-Laurent et al., 2017, 2019). Furthermore, Fe bioavailability is conceptually more complicated than discussed herein, as marine organisms at multiple trophic levels affect the speciation, bioaccessibility and bioavailability of Fe, as well as the transfer between less-labile and more-labile Fe pools in the marine environment (Poorvin et al., 2004; Vraspir and Butler, 2009; Gledhill and Buck, 2012). Many microbial species release organic ligands into solution, which stabilize dissolved Fe as organic complexes. These feedbacks are challenging to model (Strzepek et al., 2005) but may exert a cap on the lateral transfer of Fe away from glacier inputs (Lippiatt et al., 2010; Thuroczy et al., 2012). To date, Fe fluxes from glaciers into the ocean have primarily been constructed from an inor-

**Table 5.** Suppositional effect of different discharge scenarios calculated from the Redfield ratio 106 C : 16 N : 1 P : 0.005 Fe (Redfield, 1934; Twining and Baines, 2013). A steady freshwater discharge of $250\,\mathrm{m^3\,s^{-1}}$ is either released from a land-terminating glacier or from a marine-terminating glacier at 100–250 m depth, in both cases for two months into Fe-replete, $NO_3$-deficient or Fe-deficient, $NO_3$-replete marine environments. Freshwater endmembers are defined as $2\,\mu M\,NO_3$ and 33–675 nM dissolved Fe (Annett et al., 2017; Hodson et al., 2017; Hopwood et al., 2018). Ambient water column conditions are defined as Greenland (Achterberg et al., 2018) (i.e. high-Fe, low-$NO_3$) and Ross Sea (Marsay et al., 2017) (i.e. low-Fe, high-$NO_3$) shelf profiles.

|  | Surface discharge | Subglacial discharge |
|---|---|---|
| High-Fe, low-$NO_3$ environment (predominant Arctic condition) | e.g. Young Sound $< 0$–0.017 Gmol C | e.g. Bowdoin Fjord, Sermilik 0.27–1.0 Gmol C |
| Low-Fe, high-$NO_3$ environment (predominant Antarctic condition) | e.g. Antarctic Peninsula 0.009–1.9 Gmol C | e.g. Antarctic Peninsula 0.019–1.9 Gmol C |

ganic, freshwater perspective (Raiswell et al., 2006; Raiswell and Canfield, 2012; Hawkings et al., 2014). Yet to understand the net change in Fe availability to marine biota, a greater understanding of how ligands and estuarine mixing processes moderate the glacier-to-ocean Fe transfer will evidently be required (Lippiatt et al., 2010; Schroth et al., 2014; Zhang et al., 2015).

## 7 Effects on the carbonate system

Beyond its impact on inorganic nutrient dynamics, glacial discharge also affects the inorganic carbon system, commonly referred to as the carbonate system, in seawater. The carbonate system describes the seawater buffer system and consists of dissolved $CO_2$ and carbonic acid, bicarbonate ions and carbonate ions. These components buffer pH and are the main reason for the ocean's capacity to absorb atmospheric $CO_2$. The interaction between these chemical species, which varies with physical conditions including temperature and salinity (Dickson and Millero, 1987), dictates the pH of seawater and the saturation state of biologically important carbonate minerals such as aragonite and calcite ($\Omega$Ar and $\Omega$Ca, respectively). Discharge generally reduces the total alkalinity (TA, buffering capacity) of glacially modified waters mainly through dilution (Fig. 8), which results in a decreased carbonate ion concentration. Since carbonate ions are the main control on the solubility of $CaCO_3$, decreasing carbonate ion availability due to meltwater dilution negatively impacts the aragonite and calcite saturation state (Doney et al., 2009; Fransson et al., 2015). Glacier discharge can also moderate the carbonate system indirectly, as higher primary production leads to increased biological dissolved inorganic carbon (DIC) uptake, lower $pCO_2$ and thus higher pH in seawater. Therefore increasing or decreasing primary production also moderates pH and the aragonite and calcite saturation state of marine surface waters.

Total alkalinity measurements of glacial discharge across the Arctic reveal a range from 20 to $550\,\mu mol\,kg^{-1}$ (Yde et al., 2005; Sejr et al., 2011; Rysgaard et al., 2012; Evans et al., 2014; Fransson et al., 2015, 2016; Meire et al., 2015; Turk et al., 2016). Similar to Si concentrations, the broad range is likely explained by different degrees of interaction between meltwater and bedrock, with higher alkalinity corresponding to greater discharge–bedrock interaction (Wadham et al., 2010; Ryu and Jacobson, 2012), and also reflects local changes in bedrock geology (Yde et al., 2005; Fransson et al., 2015). However, in absolute terms even the upper end of the alkalinity range reported in glacial discharge is very low compared to the volume-weighted average of Arctic rivers, $1048\,\mu mol\,kg^{-1}$ (Cooper et al., 2008). In an Arctic context, meltwater is therefore relatively corrosive. In addition to low total alkalinity, glacier estuaries can exhibit undersaturation of $pCO_2$ due to the non-linear effect of salinity on $pCO_2$ (Rysgaard et al., 2012; Meire et al., 2015). This undersaturation arises even when the freshwater endmember is in equilibrium with atmospheric $pCO_2$ and thus part of the $CO_2$ drawdown observed in Arctic glacier estuaries is inorganic and not associated with primary production. In Godthåbsfjord this effect is estimated to account for 28 % of total $CO_2$ uptake within the fjord (Meire et al., 2015).

By decreasing the TA of glacially modified waters (Fig. 8), glacier discharge reduces the aragonite and calcite saturation states, thereby amplifying the effect of ocean acidification (Fransson et al., 2015, 2016; Ericson et al., 2019). High primary production can mitigate this impact as photosynthetic $CO_2$ uptake reduces DIC and $pCO_2$ (e.g. Fig. 9) in surface waters and increases the calcium carbonate saturation state (Chierici and Fransson, 2009; Rysgaard et al., 2012; Meire et al., 2015). In relatively productive fjords, the negative effect of TA dilution may therefore be counter balanced. However, in systems where discharge-driven stratification is responsible for low productivity, increased discharge may create a positive feedback on ocean acidification state in the coastal zone resulting in a lower saturation state of calcium carbonate (Chierici and Fransson, 2009; Ericson et al., 2019).

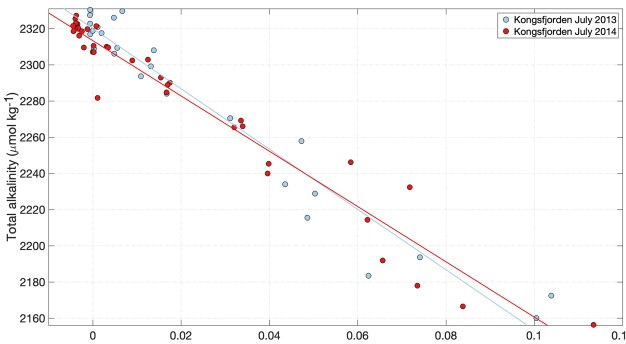

**Figure 8.** Total alkalinity in Kongsfjorden during the meltwater season (data from Fransson and Chierici, 2019). A decline in alkalinity is evident with increasing freshwater fraction in response to the low alkalinity concentrations in glacier discharge. Freshwater fraction was calculated using an average marine salinity endmember of 34.96; hence some slightly negative values are calculated in the outer fjord associated with the higher salinity of unmodified Atlantic water. Linear regression details are shown in Table S1.

Low-calcium carbonate saturation states ($\Omega < 1$; i.e. corrosive conditions) have been observed in the inner part of Glacier Bay (Alaska), demonstrating that glaciers can amplify seasonal differences in the carbonate system and negatively affect the viability of shell-forming marine organisms (Evans et al., 2014). Low $\Omega Ar$ has also been observed in the inner parts of Kongsfjorden, coinciding with high glacial discharge (Fransson et al., 2016). Such critically low $\Omega Ar$ ($< 1.4$) conditions have negative effects on aragonite-shell-forming calcifiers such as the pteropod *Limacina helicina* (Comeau et al., 2009, 2010; Lischka et al., 2011; Lischka and Riebesell, 2012; Bednaršek et al., 2014). Under future climate scenarios, in addition to the effect of increased glacier drainage in glacier fjords, synergistic effects with a combination of increased ocean $CO_2$ uptake and warming will further amplify changes to the ocean acidification state (Fransson et al., 2016; Ericson et al., 2019), resulting in increasingly pronounced negative effects on calcium carbonate shell formation (Lischka and Riebesell, 2012).

## 8 Organic matter in glacial discharge

In addition to inorganic ions, glacial discharge also contains many organic compounds derived from biological activity on glacier surfaces and overridden sediments (Barker et al., 2006; Lawson et al., 2014b). Organic carbon stimulates bacterial activity, and remineralization of organic matter is a pathway to resupply labile nitrogen and phosphorous to microbial communities. Similar to macronutrient concentrations, DOM concentrations in glacial discharge are generally low (Table 2) compared to runoff from large Arctic rivers, which have DOM concentrations 1–2 orders of magnitude higher (Dittmar and Kattner, 2003; Le Fouest et al.,

2013). This is evidenced in Young Sound where dissolved organic carbon (DOC) concentrations increase with salinity in surface waters, demonstrating that glaciers are a relatively minor source of DOM to the fjord (Paulsen et al., 2017).

While DOM concentrations are low in glacial discharge, the bioavailability of this DOM is much higher than its marine counterpart (Hood et al., 2009; Lawson et al., 2014b; Paulsen et al., 2017). This is likely due to the low C : N ratio of glacial DOM, as N-rich DOM of microbial origin is generally highly labile (Lawson et al., 2014a). It has been suggested that as glaciers retreat and the surrounding catchments become more vegetated, DOC concentrations in these catchments will increase (Hood and Berner, 2009; Csank et al., 2019). However, DOM from non-glacial terrestrial sources has a higher composition of aromatic compounds and thus is less labile (Hood and Berner, 2009; Csank et al., 2019). Furthermore, glacier coverage in watersheds is negatively correlated with DOC : DON ratios, so a reduction in the lability of DOM with less glacial coverage is also expected (Hood and Scott, 2008; Hood and Berner, 2009; Ren et al., 2019).

While DOC is sufficient to drive bacterial metabolism, bacteria also depend on nitrogen and phosphorus for growth. In this respect, bacteria are in direct competition with phytoplankton for macronutrients, and increasing additions of labile DOM downstream of glaciers could give bacteria a competitive edge. This would have important ecological consequences for the function of the microbial food web and the biological carbon sink (Larsen et al., 2015). Experiments with Arctic fjord communities, including Kongsfjorden, have shown that when bacteria are supplied with additional subsidies of labile carbon under nitrate limitation, they outcompete phytoplankton for nitrate (Thingstad et al., 2008; Larsen et al., 2015). This is even the case when there is an addition of excess Si, which might be hypothesized to give diatoms a competitive advantage. The implications of such competition for the carbon cycle are however complicated by mixotrophy (Ward and Follows, 2016; Stoecker et al., 2017). An increasing number of primary producers have been shown to be able to simultaneously exploit inorganic resources and living prey, combining autotrophy and phagotrophy in a single cell. Mixotrophy allows protists to sustain photosynthesis in waters that are severely nutrient limited and provides an additional source of carbon as a supplement to photosynthesis. This double benefit decreases the dependence of primary producers on short-term inorganic nutrient availability. Moreover, mixotrophy promotes a shortened, and potentially more efficient, chain from nutrient regeneration to primary production (Mitra et al., 2014). Whilst mixotrophy is sparsely studied in Arctic glacier fjords, both increasing temperatures and stratification are expected to favour mixotrophic species (Stoecker and Lavrentyev, 2018), and thus an understanding of microbial food web dynamics is vital to predict the implications of increasing discharge on the carbon cycle in glacier fjord systems.

Regardless of the high bioavailability of DOM from glacial discharge, once glacial DOM enters a fjord and is diluted by ocean waters, evidence of its uptake forming a significant component of the microbial food web in the Arctic has yet to be observed. Work from several outlet glacier fjords around Svalbard shows that the stable isotopic C ratio of bacteria does not match that of DOC originating from local glaciers, suggesting that glacially supplied DOC is a minor component of bacterial consumption compared to autochthonous carbon sources (Holding et al., 2017; Paulsen et al., 2018). Curiously, a data synthesis of taxonomic populations for glaciated catchments globally suggests a significant positive effect of glaciers on bacterial populations in glacier fjords but a negative effect in freshwaters and glacier forefields (Cauvy-Frauniè and Dangles, 2019). This suggests that multiple ecological and physical–chemical processes are at play, such that a simplistic argument that increasing glacial supply of DOC favours bacterial activity is moderated by other ecological factors. This is perhaps not surprising as different taxonomic groups may respond differently to perturbations from glacier discharge leading to changes in food web dynamics. For example, highly turbid glacial waters have particularly strong negative effects on filter-feeding (Arendt et al., 2011; Fuentes et al., 2016) and phagotrophic organisms (Sommaruga, 2015) and may also lead to reduced viral loads in the water column due to adsorption onto particle surfaces (Maat et al., 2019).

Whilst concentrations of DOM are low in glacier discharge, DOM-sourced nitrogen and phosphorous could still be relatively important in stratified outlet glacier fjords simply because inorganic nutrient concentrations are also low (e.g. Fig. 3). Refractory DON in rivers that is not directly degraded by bacteria can be subsequently broken down by photoammonification processes releasing ammonium (Xie et al., 2012). In large Arctic rivers, this nitrogen supply is greater than that supplied from inorganic sources (Le Fouest et al., 2013). For glacier discharge, processing of refractory DOM could potentially produce a comparable nitrogen flux to inorganic sources (Table 2, Wadham et al., 2016). Similarly, in environments where inorganic $PO_4$ concentrations are low, DOP may be a relatively more important source of phosphorous for both bacteria and phytoplankton. Many freshwater and marine phytoplankton species are able to synthesize the enzyme alkaline phosphatase in order to efficiently utilize DOP (Hoppe, 2003; Štrojsová et al., 2005). In the context of stratified, low-salinity inner-fjord environments, where inorganic $PO_4$ concentrations are potentially low enough to limit primary production (Prado-Fiedler, 2009), this process may be particularly important – yet DOP dynamics are understudied in glaciated catchments with limited data available (Stibal et al., 2009, Hawkings et al., 2016).

Finally, whilst DOC concentrations in glacier discharge are low, POC concentrations, which may also impact microbial productivity in the marine environment and contribute to the C sink within fjords, are less well characterized. Down-

stream of Leverett Glacier, mean runoff POC concentrations are reported to be 43–346 μM – 5 times higher than DOC (Lawson et al., 2014b). However, the opposite is reported for Young Sound, where DOC concentrations in three glacier-fed streams were found to be 7–13 times higher than POC concentrations (Paulsen et al., 2017). Similarly, low POC concentrations of only 5 μM were found in supraglacial discharge at Bowdoin glacier (Kanna et al., 2018). In summary, relatively little is presently known about the distribution, fate and bioavailability of POC in glaciated catchments.

## 9   Insights into the long-term effects of glacier retreat

Much of the present interest in Arctic ice–ocean interactions arises because of the accelerating increase in discharge from the Greenland Ice Sheet, captured by multi-annual to multi-decadal time series (Bamber et al., 2018). This trend is attributed to atmospheric and oceanic warming due to anthropogenic forcing, at times enhanced by persistent shifts in atmospheric circulation (Box, 2002; Ahlström et al., 2017). From existing observations, it is clear that strong climate variability patterns are at play, such as the North Atlantic Oscillation/Arctic Oscillation, and that, in order to place recent change in context, time series exceeding the satellite era are required. Insight can be potentially gained from research into past sedimentary records of productivity from high-latitude marine and fjord environments. Records of productivity and the dominance of different taxa as inferred by microfossils, biogeochemical proxies and genetic records from those species that preserve well in sediment cores can help establish long-term spatial and temporal patterns around the present-day ice sheet periphery (Ribeiro et al., 2012). Around Greenland and Svalbard, sediment cores largely corroborate recent fjord-scale surveys suggesting that inner-fjord water column environments are generally low-productivity systems (Kumar et al., 2018), with protist taxonomic diversity and overall productivity normally higher in shelf waters than in inner-fjord environments (Ribeiro et al., 2017).

Several paleoclimate archives and numerical simulations suggest that the Arctic was warmer than today during the early to mid-Holocene thermal maximum ($\sim 8000$ years ago), which was registered by $\sim 1$ km thinning of the Greenland Ice Sheet (Lecavalier et al., 2017). Multiproxy analyses performed on high-resolution and well-dated Holocene marine sediment records from contrasting fjord systems are therefore one approach to understand the nature of such past events, as these sediments simultaneously record climate and some long-term biotic changes representing a unique window into the past. However, while glacial–interglacial changes can provide insights into large-scale ice–ocean interactions and the long-term impact of glaciers on primary production, these timescales are of limited use to understanding more recent variability at the ice–ocean interface of fjord systems such as those mentioned in this review. The five well-

characterized Arctic fjords used as case studies here (Fig. 1; Bowdoin, Kongsfjorden, Sermilik, Godthåbsfjord and Young Sound), for example, did not exist during the Last Glacial Maximum $\sim$ 19 000 years ago (Knutz et al., 2011).

On long timescales, glacier–ocean interactions are subject to marked temporal changes associated with glacial–interglacial cycles. In the short term, the position of glacier termini shifts inland during ice sheet retreat or outwards during ice sheet expansion, and in the long-term proglacial regions respond to isostatic uplift and delta progradation. The uplift of fine-grained glaciomarine and deltaic sediments is a notable feature of landscape development in fjord environments following the retreat of continental-scale ice sheets (Cable et al., 2018; Gilbert et al., 2018). This results in the gradual exposure and subsequent erosion of these sediment infills and their upstream floodplains, releasing labile organic matter to coastal ecosystems. Whilst the direct biogeochemical significance of such chemical fluxes may be limited in the marine environment on interannual timescales (Table 2), potentially more important is the Fe fertilization following wind erosion and dust emittance from glacial floodplains.

Ice core records from Greenland and Antarctica, spanning several climatic cycles, suggest that aeolian deposition rates at high latitudes were as much as 20 times greater during glacial than interglacial periods (Kohfeld and Harrison, 2001). Elevated input of terrigenous Fe during windy glacial episodes, and associated continental drying, has therefore been hypothesized to stimulate oceanic productivity through time and thus modify the oceanic and atmospheric $CO_2$ balance (Martin, 1990). While there seems to be a pervasive dust–climate feedback on a glacial–interglacial planetary scale (Shaffer and Lambert, 2018), glacier retreat also exposes new areas of unconsolidated glacial sediments leading to an increase in both dust storm events and sediment yields from glacial basins locally. The spatial scale over which this glacially derived dust can be transported (100–500 km) far exceeds that of discharge-carried nutrients (Crusius et al., 2011; Prospero et al., 2012; Bullard, 2013).

## 10   A need for new approaches?

The pronounced temporal and spatial variations evident in the properties of glacially modified waters emphasize the need for high-resolution data on both short (hourly to daily) and long (seasonal to interannual) timescales in order to understand glacial processes and their downstream effects. In Godthåbsfjord, Juul-Pedersen et al. (2015) provide a detailed study of seasonal primary production dynamics. This monthly monitoring programme captures seasonal, annual and interannual trends in the magnitude of primary production. Whilst such a time series clearly highlights a strong interannual stability in both seasonal and annual primary production ($103.7 \pm 17.8$ g C m$^{-2}$ yr$^{-1}$; Juul-Pedersen et al., 2015), it is unable to fully characterize shorter (i.e. days to

weeks) timescale events such as the spring bloom period. Yet higher data resolution cannot feasibly be sustained by shipboard campaigns.

Low-frequency, high-discharge events are known to occur in Godthåbsfjord, and other glacier fjords (Kjeldsen et al., 2014), but are challenging to observe from monthly resolution data, and thus there is sparse data available to quantify their occurrence and effects or to quantify the short-term variation in discharge rates at large, dynamic marine-terminating glaciers. Consequently, modelled subglacial discharge rates and glacier discharge derived from regional models (e.g. RACMO, Noël et al., 2015), which underpin our best-available estimates of the subglacial nutrient pump (e.g. Carroll et al., 2016), do not yet consider such variability. Time lapse imagery shows that the lifetimes and spatial extents of subglacial discharge plumes can vary considerably (Schild et al., 2016; Fried et al., 2018). While buoyant plume theory has offered important insights into the role of subglacial plumes in the nutrient pump, buoyant plume theory does not characterize the lateral expansion of plume waters. Furthermore, determining the influence of discharge, beyond the immediate vicinity of glacial outflows, is a Lagrangian exercise, yet the majority of existing observational and modelling studies have been conducted primarily in the Eulerian reference frame (e.g. ship-based profiles and moored observations that describe the water column at a fixed location). Moving towards an observational Lagrangian framework will require the deployment of new technology such as the recent development of low-cost GPS trackers which, especially when combined with in situ sensors, may improve our understanding of the transport and mixing of heat, freshwater, sediment and nutrients downstream of glaciers (Carlson et al., 2017; Carlson and Rysgaard, 2018). For example, GPS trackers deployed on "bergy bits" have revealed evidence of small-scale, retentive eddies in Godthåbsfjord (Carlson et al., 2017) and characterized the surface flow variability in Sermilik Fjord (Sutherland et al., 2014).

Unmanned aerial vehicles and autonomous surface/underwater vehicles can also be used to observe the spatio-temporal variability of subglacial plumes at high resolution (Mankoff et al., 2016; Jouvet et al., 2018). Complementing these approaches are developments in the rapidly maturing field of miniaturized chemical sensors suitable for use in cryosphere environments (Beaton et al., 2012). Such technology will ultimately reduce much of the uncertainty associated with glacier–ocean interactions by facilitating more comprehensive, more sustainable field campaigns (Straneo et al., 2019), with reduced costs and environmental footprints (Nightingale et al., 2015; Grand et al., 2017, 2019). This is evidenced by a successful prolonged mooring deployment in the Santa Inés Glacier fjord system (Fig. 9).

The Santa Inés Glacier fjord sits adjacent to the open water of the Straits of Magellan in southwest Patagonia. Moored high-resolution measurements are now collected in situ us-

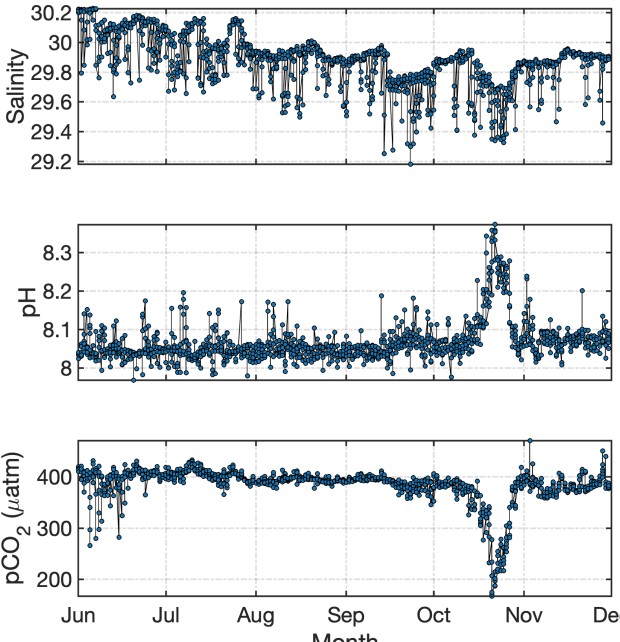

**Figure 9.** Winter–spring dynamics of salinity, pH and $p\mathrm{CO}_2$ at the Santa Inés Glacier fjord, Ballena (Patagonia). High-resolution $p\mathrm{CO}_2$ and pH measurements (every three hours) were taken in situ using autonomous SAMI-$\mathrm{CO}_2$ and SAMI-pH sensors (as per Vergara-Jara et al., 2019) (Sunburst Sensors, LLC) starting in the austral autumn (March 2018). All sensors were moored at 10 m depth.

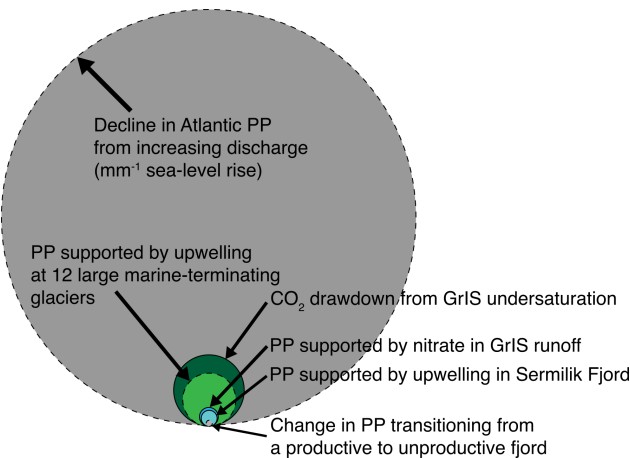

**Figure 10.** A scale comparison of the significance of different chemical/physical processes driven by glacial discharge in terms of the resulting effects on annual marine primary production (PP) or $\mathrm{CO}_2$ drawdown (units $\mathrm{Tg\,C\,yr^{-1}}$). Bold lines indicate mean estimates based on multiple independent studies; dashed lines are based on only one. Green–blue colours are positive; grey colours are negative. Calculated changes (largest–smallest) are determined from glacial discharge superimposed on a modelled global RCP8.5 scenario (Kwiatkowski et al., 2019), $p\mathrm{CO}_2$ uptake due to meltwater-induced undersaturation scaled to the Greenland Ice Sheet (Meire et al., 2015), computed upwelled $\mathrm{NO}_3$ fluxes (assuming 100 % utilization at Redfield ratio, Hopwood et al., 2018), mean freshwater $\mathrm{NO}_3$ (Greenland) inventory (Table 3), $\mathrm{NO}_3$ anomaly due to upwelling in Sermilik Fjord (Cape et al., 2019), and contrasting the mean PP for groups II and IV (Table 1) for a fjord the size of Young Sound.

ing sensor technology and a mooring within the fjord. Measurements include the carbonate system parameters $p\mathrm{CO}_2$ and pH. The 2018 winter to spring time series (Fig. 9) demonstrates a sharp decline in $p\mathrm{CO}_2$ and corresponding increase in pH, associated with the onset of the spring bloom in early October. Such a pronounced event, occurring over $\sim 2$ weeks, would be impossible to characterize fully with monthly sampling of the fjord. Over winter, pH and $p\mathrm{CO}_2$ were more stable, but sensor salinity data still reveal short-term dynamics within the fjords' surface waters (Fig. 9). A general decline in salinity is evident moving from winter into spring. Short-term changes on diurnal timescales – presumably linked to tidal forcing – and also on daily–weekly timescales – possibly linked to weather patterns – are also evident (Fig. 9). Much work remains to be done to deduce the role of these short-term drivers on primary production.

Finally, we note that the different scales over which the processes discussed herein operate raises the critical question of how importantly the different effects of glacial discharge on the marine environment are perceived in different research fields. Herein we have largely focused on local- to regional-scale processes operating on seasonal to interannual timescales in the marine environment at individual field sites (Fig. 1). A very different emphasis may have been placed on the relative importance of different processes if a different spatial/temporal perspective had been adopted, for

example considering the decadal–centennial effects of increasing meltwater addition to the Atlantic Ocean, or conversely the seasonal effect of meltwater solely within terrestrial systems. One conceptual way of comparing some of the different processes and effects occurring as a result of glacial discharge is to consider a single biogeochemical cycle on a global scale, for example the carbon drawdown associated with marine primary production (Fig. 10).

A net decrease in primary production is predicted over the 21st century at the Atlantic scale on the order of $> 60\,\mathrm{Tg\,C\,yr^{-1}\,mm^{-1}}$ of annual sea-level rise from Greenland due solely to the physical effects of freshwater addition (Kwiatkowski et al., 2019). An example of a potential negative effect on primary production operating on a much smaller scale would be the retreat of marine-terminating glaciers and the associated loss of $\mathrm{NO}_3$ upwelling (Torsvik et al., 2019). The effect of switching a modest glacier fjord the size of Young Sound from being a higher-productivity marine-terminating glacier fjord environment to a low-productivity glacier fjord environment receiving runoff only from land-terminating glaciers (using mean primary production values from Table 1) would be a change of $\sim 0.01\,\mathrm{Tg\,C\,yr^{-1}}$. Conversely, potential positive effects of glacier discharge on primary production can be estimated us-

ing the Redfield ratio (Redfield, 1934) to approximate how much primary production could be supported by $NO_3$ supplied to near-surface waters from meltwater-associated processes. Adding all the $NO_3$ in freshwater around Greenland (Table 3) into the ocean, in the absence of any confounding physical effects from stratification, would be equivalent to primary production of $\sim 0.09\,\mathrm{Tg\,C\,yr^{-1}}$. Using the same arbitrary conversion to scale other fluxes, the primary production potentially supported by upwelling of $NO_3$ at Sermilik (Cape et al., 2019) is approximately $0.13\,\mathrm{Tg\,C\,yr^{-1}}$ and that supported by upwelling of $NO_3$ at 12 large Greenlandic marine-terminating systems (Hopwood et al., 2018) is approximately $1.3\,\mathrm{Tg\,C\,yr^{-1}}$. Finally the inorganic $CO_2$ drawdown due to $p CO_2$ undersaturation in glacier estuaries around Greenland is approximately $1.8\,\mathrm{Tg\,C\,yr^{-1}}$ (Meire et al., 2015).

These values provide a rough conceptual framework for evaluating the relative importance of different processes operating in parallel but on different spatial scales (Fig. 10). Whilst a discussion of glacial weathering processes is beyond the scope of this review, we note that these estimates of annual C fluxes (Fig. 10) are comparable to, or larger than, upper estimates of the $CO_2$ drawdown/release associated with weathering of carbonate, silicate and sulfide minerals in glaciated catchments globally (Jones et al., 2002; Tranter et al., 2002; Torres et al., 2017). The implication of this is that shifts in glacier–ocean inter-connectivity could be important compared to changes in weathering rates in glaciated catchments in terms of feedbacks in the C cycle on inter-annual timescales.

### 10.1 A link between retreating glaciers and harmful algal blooms?

Shifts between different microbial groups in the ocean can have profound implications for ecosystem services. For example, addition of DOM can induce shifts in the microbial loop to favour bacteria in their competition with phytoplankton for macronutrient resources, which directly affects the magnitude of $CO_2$ uptake by primary producers (Thingstad et al., 2008; Larsen et al., 2015). Similarly, changing the availability of Si relative to other macronutrients affects the viability of diatom growth and thus, due to the efficiency with which diatom frustules sink, potentially the efficiency of the biological carbon pump (Honjo and Manganini, 1993; Dugdale et al., 1995).

A particularly concerning hypothesis, recently proposed from work across Patagonian fjord systems and the first evaluations of harmful algal bloom (HAB)-associated species around Greenland, is that changes in glacier discharge and associated shifts in stratification and temperature could affect HAB occurrence (Richlen et al., 2016; León-Muñoz et al., 2018; Joli et al., 2018). In the Arctic, very little work has been done to specifically investigate HAB occurrence and drivers in glacier-discharge-affected regions. Yet HAB-associated species are known to be present in Arctic waters (Lefebvre et al., 2016; Richlen et al., 2016), including *Alexandrium tamarense*, which has been implicated as the cause of toxin levels exceeding regulatory limits in scallops from west Greenland (Baggesen et al., 2012), and *Alexandrium fundyense*, cysts of which have been found at low concentrations in Disko Bay (Richlen et al., 2016). Around Greenland, low temperatures are presently thought to be a major constraint on HAB development (Richlen et al., 2016). Yet increasing meltwater discharge into coastal regions drives enhanced stratification and thus directly facilitates the development of warm surface waters through summer. This meltwater-driven stratification has been linked to the occurrence of HAB species including the diatoms *Pseudonitzschia* spp. (Joli et al., 2018). Thus, increasing freshwater discharge from Greenland could increase HAB viability in downstream stratified marine environments (Richlen et al., 2016; Joli et al., 2018; Vandersea et al., 2018), potentially with negative impacts on inshore fisheries.

Given the ongoing intensification of climate change and the interacting effects of different environmental drivers of primary production in glacier fjord systems (e.g. surface warming, carbonate chemistry, light availability, stratification, nutrient availability and zooplankton distribution), it is however very challenging to predict future changes on HAB event frequency and intensity. Furthermore, different HAB-associated groups (e.g. toxin-producing diatom and flagellate species) may show opposite responses to the same environmental perturbation (Wells et al., 2015). Moreover, many known toxin-producing species in the Arctic are mixotrophic, further complicating their interactions with other microbial groups (Stoecker and Lavrentyev, 2018). Fundamental knowledge gaps clearly remain concerning the mechanisms of HAB development, and there are practically no time series or studies to date investigating changes specifically in glaciated Arctic catchments. Given the socioeconomic importance of glacier-fjord-scale subsistence fisheries, especially around Greenland, one priority for future research in the Arctic is to establish to what extent HAB-associated species are likely to benefit from future climate scenarios in regions where freshwater runoff is likely to be subject to pronounced ongoing changes (Baggesen et al., 2012; Richlen et al., 2016; Joli et al., 2018).

### 11 Understanding the role of glaciers alongside other manifestations of climate change

In order to comprehensively address the questions posed in this review, it is evident that a broader perspective than a narrow focus on freshwater discharge alone, and its regional biogeochemical effects, is required (Fig. 10). Freshwater discharge is not the sole biogeochemical connection between the glaciers and the ocean (Fig. 11). Dust plumes from proglacial terrain supply glacial flour to the ocean on scales

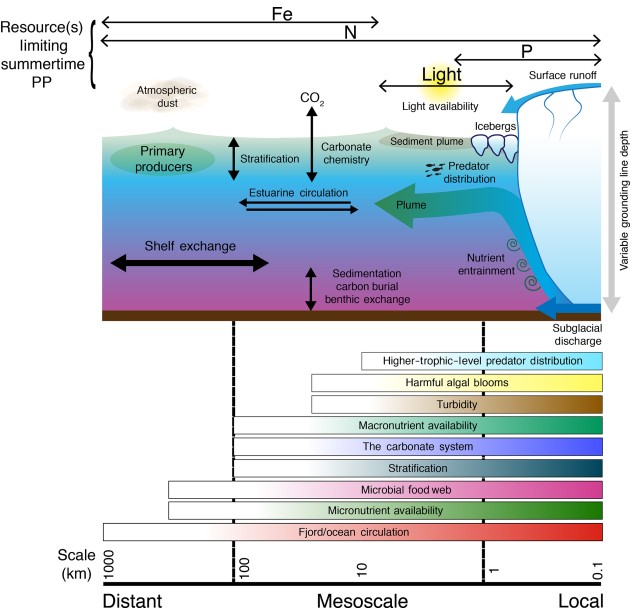

**Figure 11.** The approximate spatial scale over which glaciers directly affect different drivers of marine primary production (PP) compared to the likely limiting resources constraining primary production.

of $> 100 \, \text{km}$ and thus act as an important source of Fe to the ocean at high latitudes, where other atmospheric dust sources are scarce (Prospero et al., 2012; Bullard, 2013). Similarly, icebergs have long been speculated to act as an important source of Fe to the offshore ocean (Hart, 1934; Raiswell et al., 2008; Lin et al., 2011) and induce mixing of the surface ocean (Helly et al., 2011; Carlson et al., 2017). Whilst freshwater discharge is a driver of biogeochemical changes in nearshore and fjord environments downstream of glaciers (Arimitsu et al., 2016), the distant ($> 100 \, \text{km}$ scale) biogeochemical effects of glaciers on the marine environment are likely dominated by these alternative mechanisms (Fig. 11). Furthermore, the distal physical effects of adding increasingly large volumes of glacier discharge into the Atlantic may have biogeochemical feedbacks which, whilst poorly studied, are potentially far larger than individual regional-scale processes discussed herein (Fig. 10) (Kwiatkowski et al., 2019).

Discharge-derived effects must also be interpreted in the context of other controls on primary production in the high-latitude marine environment. Sea-ice properties, and particularly the timing of its breakup and the duration of the ice-free season, are a key constraint on the seasonal trend in primary production in the Arctic (Rysgaard et al., 1999; Rysgaard and Glud, 2007). Similarly, whilst discharge affects multiple aspects of the three-dimensional water column including fjord-scale circulation and mixing (Kjeldsen et al., 2014; Carroll et al., 2017), stratification (Meire et al., 2016b; Oliver et al., 2018), and boundary current properties (Sutherland et

al., 2009), other changes in the Earth system including wind patterns (Spall et al., 2017; Sundfjord et al., 2017; Le Bras et al., 2018), sea-ice dynamics, regional temperature increases (Cook et al., 2016), and other freshwater sources (Benetti et al., 2019) are driving changes in these parameters on similar spatial and temporal scales (Stocker et al., 2013; Hop et al., 2019).

Several key uncertainties remain in constraining the role of glaciers in the marine biogeochemical system. Outlet glacier fjords are challenging environments in which to gather data, and there is a persistent deficiency of both physical and biogeochemical data within kilometres of large marine-terminating glacier systems, where glacier discharge first mixes with ocean properties. Subglacial discharge plume modelling and available data from further downstream can to some extent evade this deficiency for conservative physical (e.g. salinity and temperature) and chemical (e.g. noble gases, $NO_3$ and $PO_4$) parameters in order to understand mixing processes (Mortensen et al., 2014; Carroll et al., 2017; Beaird et al., 2018). However, the mixing behaviour of non-conservative chemical parameters (e.g. pH, Si, and Fe) is more challenging to deduce from idealized models. Furthermore, the biogeochemical effects of low-frequency, high-discharge events and small-scale mixing, such as that induced around icebergs, remain largely unknown. There is a critical need to address this deficiency by the deployment of new technology to study marine-terminating glacier mixing zones and downstream environments.

The uniqueness of individual glacier fjord systems, due to highly variable fjord circulation and geometry, is itself a formidable challenge in scaling up results from Arctic field studies to produce a process-based understanding of glacier–ocean interactions. A proposed solution, which works equally well for physical, chemical and biological perspectives, is to focus intensively on a select number of key field sites at the land–ocean interface rather than mainly on large numbers of broadscale, summertime-only surveys (Straneo et al., 2019). In addition to facilitating long-term time series, focusing in detail on fewer systems facilitates greater seasonal coverage to understand the changes in circulation and productivity that occur before, during and after the melt season. However, the driving rationale for the selection of key glacier field sites to date was in many cases their contribution to sea-level rise. Thus, well-studied sites account for a large fraction of total Arctic glacier discharge into the ocean but only represent a small fraction of the glaciated coastline. For example, around the Greenland coastline, the properties of over 200 marine-terminating glaciers are characterized (Morlighem et al., 2017). Yet just 5 glaciers (including Helheim in Sermilik Fjord) account for 30 % of annual combined meltwater and ice discharge from Greenland, and 15 account for $> 50$ % (year 2000 data, Enderlin et al., 2014). The relative importance of individual glaciers changes when considering longer time periods (e.g. 1972–2018, Mouginot et al., 2019), yet, irrespective

of the timescale considered, a limited number of glaciers account for a large fraction of annual discharge. Jakobshavn Isbræ and Kangerlussuaq, for example, are among the largest four contributors to ice discharge around Greenland over both historical (1972–2018) and recent (2000–2012) time periods (Enderlin et al., 2014; Mouginot et al., 2019). Whilst small glaciated catchments, such as Kongsfjorden and Young Sound, are far less important for sea-level rise, similar "small" glaciers occupy a far larger fraction of the high-latitude coastline and are thus more representative of glaciated coastline habitat.

## 12  Conclusions

### 12.1  Where and when does glacial freshwater discharge promote or reduce marine primary production?

In the Arctic, marine-terminating glaciers are associated with the enhanced vertical fluxes of macronutrients, which can drive summertime phytoplankton blooms throughout the meltwater season.

In the Arctic, land-terminating glaciers are generally associated with the local suppression of primary production, due to light limitation and stratification impeding vertical nutrient supply from mixing. Primary production in Arctic glacier fjords without marine-terminating glaciers is generally low compared to other coastal environments.

In contrast to the Arctic, input of Fe from glaciers around the Southern Ocean is anticipated to have a positive effect on marine primary production, due to the extensive limitation of primary production by Fe.

In some brackish, inshore waters, DOM from glaciated catchments could enhance bacterial activity at the expense of phytoplankton, but a widespread effect is unlikely due to the low DOM concentration in freshwater.

Glacier discharge reduces the buffering capacity of glacially modified waters and amplifies the negative effects of ocean acidification, especially in low-productivity systems, which negatively effects calcifying organisms.

### 12.2  How does spatio-temporal variability in glacial discharge affect marine primary production?

Glacier retreat associated with a transition from marine- to land-terminating systems is expected to negatively affect downstream productivity in the Arctic, with long-term inland retreat also changing the biogeochemical composition of freshwater.

Low-frequency, high-discharge events are speculated to be important drivers of physical and biogeochemical processes in the marine environment, but their occurrence and effects are poorly constrained.

HAB viability may increase in future Arctic glacier fjords in response to increasing discharge driving enhanced strati-

fication, but there are very limited data available to test this hypothesis.

A time series in Godthåbsfjord suggests that, on interannual timescales, fjord-scale primary production is relatively stable despite sustained increases in glacier discharge.

### 12.3  How far-reaching are the effects of glacial discharge on marine biogeochemistry?

Local effects of glaciers (within a few kilometres of the terminus, or within glacier fjords) include light suppression, impediment of filter-feeding organisms and influencing the foraging habits of higher organisms.

Mesoscale effects of glaciers (extending tens to hundreds of kilometres from the terminus) include nutrient upwelling, Fe enrichment of seawater, modification of the carbonate system (both by physical and biological drivers) and enhanced stratification.

Remote effects are less certain. Beyond the 10–100 km scale over which discharge plumes can be evident, other mechanisms of material transfer between glaciers and the ocean, such as atmospheric deposition of glacial flour and icebergs, are likely more important than meltwater (Fig. 11). Fully coupled biogeochemical and physical global models will be required to fully assess the impacts of increasing discharge into the ocean on a pan-Atlantic scale (Fig. 10).

*Data availability.* CE1 Data sources are cited within the text. For primary production data, see Andersen (1977), Nielsen and Hansen (1995), Jensen et al. (1999), Nielsen (1999), Levinsen and Nielsen (2002), Juul-Pedersen et al. (2015), Meire et al. (2017), Lund-Hansen et al. (2018), Hop et al. (2002), Iversen and Seuthe (2011), Hodal et al. (2012), van de Poll et al. (2018), Seifert et al. (2019), Smoła et al. (2017), Rysgaard et al. (1999), Holding et al. (2019), Harrison et al. (1982), and Reisdorph and Mathis (2015). For chemical data and associated fluxes, see Fransson et al. (2016), van de Poll et al. (2018), Cantoni et al. (2019), Cauwet and Sidorov (1996), Emmerton et al. (2008), Hessen et al. (2010), Hopwood et al. (2016, 2017, 2018), Kanna et al. (2018), Cape et al. (2019), Hawkings et al. (2014, 2017), Lund-Hansen et al. (2018), Meire et al. (2015, 2016a), Brown et al. (2010), Paulsen et al. (2017), Stevenson et al. (2017), Statham et al. (2008), Bhatia et al. (2010, 2013a, 2013b), Lawson et al. (2014b), Hood et al. (2015), Csank et al. (2019), Wadham et al. (2016), Achterberg et al. (2018), Marsay et al. (2017), Annett et al. (2017), Ducklow et al. (2017), Tonnard et al. (2020), Lippiatt et al. (2010), Fransson and Chierici (2019), Vergara-Jara et al. (2019), and Kwiatkowski et al. (2019). For discharge plume properties, see Carroll et al. (2016), Halbach et al. (2019), Kanna et al. (2018), Mankoff et al. (2016), Meire et al. (2016b), Jackson et al. (2017), Bendtsen et al. (2015), Beaird et al. (2018), and Schaffer et al. (2020).

*Supplement.* The supplement related to this article is available online: https://doi.org/10.5194/tc-14-1-2020-supplement.

*Author contributions.* TD coordinated workshop activities and designed questions to structure the review paper. MJH coordinated manuscript writing. All authors contributed to writing at least one section of the review and assisted with the revision of other sections. DC edited all figures.

*Competing interests.* The authors declare that they have no conflict of interest.

*Acknowledgements.* The authors thank all conveners and participants of the IASC cross-cutting activity "The importance of Arctic glaciers for the Arctic marine ecosystem" hosted by the Cryosphere Working Group/Network on Arctic Glaciology and the Marine Working Group. IASC funding to support early career scientist attendance is gratefully acknowledged. Figure 7 and all linear regressions were produced in SigmaPlot.

*Financial support.* Mark Hopwood was financed by the DFG (award number HO 6321/1-1). Andy Hodson was supported by Joint Programming Initiative (JPI-Climate Topic 2: Russian Arctic and Boreal Systems) award 71126 and Research Council of Norway grant 294764. Johnna Holding was supported by Marie Curie grant GrIS-Melt (752325). Lorenz Meire was supported by the VENI program from the Dutch Research Council (NWO grant 016.Veni.192.150). José L. Iriarte received support from the FONDECYT 1170174 project. Sofia Ribeiro received support from Geocenter Denmark (project GreenShift). Thorben Dunse was supported by the Nordforsk-funded project (GreenMAR).

The article processing charges for this open-access publication were covered by a Research Centre of the Helmholtz Association.

*Review statement.* This paper was edited by Evgeny A. Podolskiy and reviewed by Jon Hawkings and Kiefer Forsch.

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

**Remarks from the language copy-editor**

CE1    Please verify this new section carefully.

**Remarks from the typesetter**

TS1    **Editors approval necessary. I would send the explanation to the editor, which you wrote in your last proofreading. Please confirm or write a new statement for the editor why this change needs to be made.**

TS2    Please check the rows here. Or would you like to swap the complete rows?