# Peer review of "Review article: How does glacier discharge affect marine biogeochemistry and primary production in the Arctic?"

_The Cryosphere, 2019_

## Short Comment (SC1) · 25 Jun 2019

A paper,

The biogeochemical impact of glacial meltwater from Southwest Greenland,

by Hendry et al has just been accepted by Progress in Oceanography.

https://doi.org/10.1016/j.pocean.2019.102126

The highlights are:

"Novel multi-disciplinary approach to tracing freshwater and particle transport into

boundary currents.

Significant glacial inputs reach coastal waters and are transported rapidly offshore.

Low surface water dissolved silicon concentrations maintained by diatom activity despite strong glacial and benthic supplies".

I guess that the processes described in this paper and their consequences should be included in the review for completeness.
* * *

---

## Referee Comment (RC1) · Jon Hawkings (Referee) · 1 Sep 2019

This review article is timely and presents an opportunity for the authors to summarise the current "state of play" and highlight potential future research direction in the Arctic with regard to ice-ocean biogeochemical interaction. I broadly agree with most of the main points raised and the authors touch on most of the key research areas. The review is well written, the figures largely appropriate (apart from some points below) and I'm supportive of its publication after revision. However, I have suggestions for improvement. When and where some of the literature quoted is relatively selective and the way it is contextualised in certain circumstances misses nuance. One major omission is

a discussion of particulate fluxes (both as part of nutrient budgets, and importance in ballasting and C burial) and indirect processing of glacial inputs (related to particulate inputs; i.e. benthic recycling and/or burial). Given the context of these environments (dominated by inputs of products of physical weathering), and the existence of literature in other glacially influenced regions (e.g. Laura Wehrmann's and associated groups ongoing work in Svalbard; e.g. Wehrmann et al., 2014), this could have been an opportunity to start a balanced discussion. This is an oversight, especially for a review article, and given recent interest in particulate fluxes (not just in glacial locations), even if the authors do not think these flux terms are important. As a previous reviewer indicated, there is also a need to discuss and incorporate more recent publications (i.e. Hendry et al., 2019, but also Seifert et al., 2019, Wadham et al., 2019 amongst some others I suggest below), and some key papers have been omitted or not referenced where they should have been. I have some major reservations about section 8, which feels incredibly speculative, and think it should be toned down and incorporated into section 9 in a much reduced form. Specific comments to be addressed are below.

L65: Calcium carbonate is not an ion. This should be corrected to "inorganic salts...".

L64-68: These plumes also carry large quantities of reactive particular material, including labile particulate nutrients. Whatever you think of their ultimate fate (which can be discussed) I think this is important to note as it is an important characteristic of glacial meltwaters. In this context I'm sure the authors will be aware of the literature (some suggestions for inclusion are Hendry et al., 2019, Seifert et al., 2019, Jeandel and Oelkers, 2015, Grimm et al., 2019, Schoenfelt et al 2017, Morgan et al., 2014, Eiriksdottir et al., 2015).

Figure 1: I'd like to see the quality of this figure improved before publication. As the first figure and a key map of study areas it's also a little too basic at present.

Section 3: I find the referencing in the first paragraph curious. Although by no means do I think that the authors should be referencing some work ahead of others, the first
reference of a particular group's work is page 8, where it's critiqued, despite the number of publications from this group that are suitable for referencing before (in this context). The authors discuss the need for seasonal datasets to contextual flux information, yet there are already several studies currently available that contain temporal datasets over several months and several years of monitoring for hydro chemical parameters, macronutrients and Fe. The concentrations used on Table 2 are from some of these studies, and are discharge weighted mean concentrations derived from a seasonal dataset (the only DWM concentrations in Greenlandic meltwaters that I know to exist at present). There is certainly a debate that can be had with regard to the particulate nutrient inputs (which the authors should deal with in a more balanced manner), but I do not fully understand why other aspects of those papers have been overlooked. These might not be datasets that span whole melt seasons (typically early May to early September), but they are the longest available at the moment and should be acknowledged as such. I would like to see the current literature discussed in a more nuanced way in the next version of the manuscript.

L163: Semantics but I think this should be "dissolved macronutrients". Again, the role of particulate macronutrients can be critiqued, but this is an important distinction to make. Glacial meltwaters have high concentrations of particulate nutrients (save N), and low concentrations of dissolved nutrients, and it's important to highlight that whatever you think of the eventual fate.

L164-166: There is a push here to emphasise that the PO4 concentrations in glacial meltwaters are particularly low. I'm not arguing against this (they are compared to some marine waters), but the PO4 concentrations in glacial meltwaters from large catchments (see Leverett Glacier) are similar to the global river mean (0.32  $\mu$ M; Meybeck, 1982), and also similar to (or exceeding) PO4 concentrations in Arctic rivers (0.03-0.76  $\mu$ M). Further, the annual yields (normalised to catchment area) are very high (see Table 4 in Hawkings et al., 2016). Again, not all this information may be needed in the context of the review, but it's important to not single out glacial inputs as being particularly
nutrient deplete as is currently done.

L167: This needs a reference and some contextual information. See point above.

L178: I'm not sure if I'd call these measurements "extensive" given they are from two small glaciers in a fjord with many meltwater inputs (the major inputs coming from much larger tidewater glaciers). The references given for studies of Svalbard meltwaters also have listed LoD for PO4 is 5 ppb (0.16  $\mu$ M), and a limit of quantification likely even higher (although not mentioned) making those figures difficult to compare to the fjord measurements when the LoD is typically better.

L188-189: As above point, I don't really understand the referencing here. There are other appropriate studies that emphasise the existence of reactive particulate Fe that should be referenced here (Bhatia et al., 2013, Schroth et al., 2012, Schroth et al., 2014, Hawkings et al., 2014, Hawkings et al., 2018).

L198-199: Low nM concentrations are still fairly significant in a marine context, especially when  $\sim$ 100 km from the main inputs at the head of the fjord. Surface open ocean waters and even some coastal systems are typically

tion/other removal processes have been observed, especially in similar high sediment, deltaic environments (Treguer et al., 2013, Kamatani et al., 1984), apart from when strong benthic Si fluxes have been inferred (e.g. Eyre and Balls, 1999). I'm not saying dissolution of particulate material is not important in other systems, but the authors need more than one reference to support this generalisation.

L230-234: I welcome balanced debate, however, it's disappointing the review makes no mention of incubation experiments performed in this study, which show release of DSi from particulate material to seawater over a period of 30 days in samples that weren't treated to remove ASi. This doesn't necessarily mean DSi is released in the fjord surface, but it's worth consideration especially given the recent findings of Hendry et al. (2019) and Gruber et al. (2019) among others. The former shows strong evidence bottom water modification for example. The benthic environment is currently ignored and the lack of discussion of this is an oversight.

L242-254 (and Figure 3): I don't disagree with most of the interpretation here, but given that some of the low salinity end members are not dissimilar to Hawkings et al. (2017) (where there are no high salinity end members) it seems curious that the authors explain this by lack of data and complexity of fjord systems in these instances. Simply drawing linear regressions through points in Figure 3 is also misleading and doesn't tell the whole story that is being shown in each dataset. e.g. if you drew a linear regression through the Bowdoin Fjord plots at the same salinities (

higher dissolved silica concentrations (Wadham et al., 2010). Pedantic, but I'm also not too keen on the term "surface discharge", as it could indicate any meltwater entering the fjord via surface rivers. Surpaglacial meltwater would be a better term. Most supraglacial meltwater is also routed to the glacier bed (and the subglacial drainage system), so I would think this is unlikely to be a large contributor. By "ice melt" I assume the reference is to iceberg melt?

L261-264: Discussion of Hendry et al. (2019) would be useful here. I think the wording misses nuances given flux estimates of Si for ice sheets did not exist before Meire et al. (2016) and Hawkings et al. (2017), and so were considered zero in biogeochemical models and estimates of the global silica cycle. I would consider no estimate an underestimate. Table 2: I was not aware that Lawson et al. (2014) measured dissolved organic nitrogen (DON). The discharge weighted DON concentrations of Wadham et al. (2016) need to be included here (1.7  $\mu$ M). No mention has been made of NH4 concentrations. They are minor but should be discussed for completeness. I think some discussion of methodology with regard to Fe concentrations would also be appropriate here. As the authors know, it is complicated to simply compare concentrations of Fe where measurements are conducted via different methodologies, for example size fractionation (

flux differences for Fe in particular are due to an arbitrarily applied fjord removal in the papers. The 11-fold flux difference between Stevenson et al. (2017) and Hawkings et al. (2014) is due entirely to the application of an arbitrary fjord removal factor - the flux-at-gate (i.e. the flux from the river into the fjord) are very similar between the studies (note the  $\sim$ 90% removal is also discussed in Hawkings et al., 2014, and an estimate of flux after removal given).

L288-293: As several points above. A point is made that seasonal datasets are needed, yet the only publications with datasets >2 months in length have been omitted in the referencing. This needs to be rectified.

L298-302: I understand why the authors want to make this point. In defence of these studies, the flux calculations are made at the "gate" and therefore represent a first order estimate for inputs into the fjord (which is how elemental fluxes from rivers are almost universally calculated). The elemental estimates for ice sheet fluxes (previously assumed to be inconsequential) are also some of the first, so in that context glacial estimates were underestimated (as they weren't estimated at all before). As is touched upon, the largest flux term in the papers cited is the particulate loading (and the particulate fluxes from the ice sheet are massive – estimated at 8% of global sediment fluxes to the ocean; Overeem et al., 2017), which clearly isn't observed in the surface samples and on the timescales the authors discuss. It is a shame the fate of these particulates are not discussed in a balanced manner, and only somewhat negatively, if at all.

L303-307: I think this should come earlier - after line 283.

L319-323: As the authors mention elsewhere, turbidity is important in suppressing surface productivity (via light limitation), which should be mentioned here. Discussion of new work by Seifert et al. (2019) should also be discussed in the context of carbon removal.

L417-421: This is the first reference to any benthic processes occurring. This is an
oversight of the current manuscript, and this deserves discussion. Studies in both polar regions have investigated benthic recycling and diagenetic processes and the authors should discuss this as well (see Wehrmann et al., 2014, Henkel et al., 2018, Buongiorno et al., 2019).

L449: Can be a few 10s km where turbulent plume is observed and can be spatially variable with time (Tedstone et al., 2012, Hudson et al., 2014).

L456-458: This is one of several reasons why the limiting nutrients are likely differ. What about riverine inputs, dust inputs etc...?

L471-472: This is slightly misleading. The coastal regions of Antarctica have low Fe concentrations but there are now several studies highlighting the potential importance of glacial inputs. Further Figure 5 misses out PFe concentrations from Marsay et al. which are consistently >1 nM. Further the authors in this study comment that measurements come following 2 months of intense primary productivity (i.e. these are not traditionally limiting waters, but a productive coastal ecosystem).

L552: This is a rather low estimate of DFe from a grounding line and there is very little information available on concentration estimates. I'm not quite sure how the authors came to this value from Marsay et al. (2017), so it would be useful to provide a sentence to elaborate.

L584-596: Completely agree and pertinent point to make given we know almost nothing about ligand binding in glacial fjords (and very little in estuaries more generally). However, I think the perspective here is mainly focused on the idea of bioavailability in "open ocean" waters, which is almost certainly controlled by ligand binding (what this does to the bioavailability I think is still poorly understood given the wide range and complexity of metal stabilising ligands). An increasing number of studies (Kranzler et al. 2011, 2016, Shoenfelt et al. 2017, Grimm et al. 2019) are demonstrating the importance of accessing Fe from particulate pools yet there is very little discussion of this. Surely in coastal areas the particulate pool is likely to be very important given the high TCD
concentrations (Schroth et al., 2014) and is almost as poorly understood as the ligand pool? Some balanced discussion of this is important.

L608-609: I don't know what other bedrock types the author's think are likely, but carbonate and silicate bedrock broadly covers them all.

L639: Some repetition here. Please rephrase.

L620-643 p1: Linked again to my point about lack of discussion on the importance of benthic cycling, I think there should be some discussion of the potential role of alkalinity production in sedimentary environments (e.g. via denitrification and sulfate reduction).

L620-643 p2: Some contextualisation is needed here. To be my knowledge (and I am definitively not an expert in this) but there tends to be a conservative decline in alkalinity in most estuarine settings (see Cai et al., 2010 and Thomas et al. 2009 for example), so this is not unique. The trend of decreasing alkalinity with increasing freshwater is therefore not particularly surprising in the context of freshwater-saltwater continuum environments as a whole. I agree that monitoring these changes with increasing meltwater discharge will be an important future undertaking.

L620-643 p3: I think some additional detail in this section would be useful for readers. Could the authors also consider an alternative scenario whereby glacial meltwater have low pCO2 and high pH as glacial meltwater tend to be elevated in pH and corresponding low pCO2 (Tranter et al. 1993, Sharp and Tranter, 2017)? For example, there is currently no mention of the conclusions of Meire et al. (2015), which shows glacial melt water associated with low pCO2 regions of the fjord. Also see recent studies by Pilcher et al. (2018) and St Pierre et al. (2019).

L649: What is meant by a DOM concentration? Do you mean DOC concentration?

L689-695: All of the samples in the Holding et al. (2016) bar one are taken from salinities above 34 therefore it's not particularly surprising a clear signature of glacial DOC is observed in bacteria here. Additionally, there is no mention of a glacial DOM

TCD
in algae, some of which are likely to be mixotrophic as commented in this manuscript (i.e. the interpretation is not straightforward). In this context I really don't think you can consider the Holding et al. estimate of  $\sim$ 11% of bacterial OC in marine waters to be from glacial DOM as minor. It is worth mentioning that other studies (e.g. Fellman et al., 2015, Hagvar et al., 2016) much closer to glacial inputs have found assimilation of glacial DOM into food webs. This is much debated, but one part of the story is that glacial DOM is highly bioavailable (as observed by a number of studies) and is therefore likely consumed very close to the glacier front.

L695: Paulsen et al. (2018) isn't the correct reference to use in this context and is slightly misleading. This study shows that bioavailability is influenced by glacial melt-water inputs not that it is a minor component of bacterial consumption.

Section 8: This whole section feels extremely speculative to me and is not actually correlated to real world observations, nor with any observations from the Arctic. Most of the literature cited provides tenuous links with the only evidence that I can see based on the observation that HABs occur in Patagonia and that there are glaciers in Patagonia as well (but not in the same locations at the HABs). The main study cited (Leon-Munoz et al., 2018) was conducted in fjords with very little or no glacial cover, and contains no reference to glaciers, or meltwater inputs. I'm not against the inclusion of some points from this section into the next section (long-term effects of glacier retreat), as it's important to form hypotheses for testing (especially when anticipating future change), but it needs to be significantly toned down, reduced and explicit mentioned that the hypotheses are speculative.

L749-750: I disagree with some of the glaciological interpretation in this paragraph. The study cited (Bliss et al., 2014) is a modelling study to predict future meltwater runoff terms, with no observed data presented (yes future estimates of mass change and runoff are given and are useful, but that is not how this study was cited). This is especially problematic in Patagonia, where there is a relative dearth of data to use for model inputs/validation. There is no evidence to suggest that glacial runoff is in long

TCD
term decline in this region. The opposite is actually likely to be true with regard to the Patagonian Ice Fields (see recent studies of Forresta et al., 2018, Richter et al., 2019, Li et al., 2019), which are currently the largest contributor to sea level rise per unit area in the world. Glacial meltwater runoff is not intricately linked to precipitation as per non-glacial rivers, but reduced precipitation is likely to amplify mass balance losses.

L773: As above – this paper does not demonstrate this and cannot be used in this context. Section 9: I like this section, but some alterations are needed based on my previous comments above.

L829-830: I don't disagree but references needed here to substantiate point.

Figure 9: Nice looking figure, but I'd really like to see more balance in the interpretation of the literature represented in it. One major omission (again I'm going back to it) is any lack of benthic feedback. "Sedimentation and Carbon(/nutrient) [burial]" is seen as a one way process here, which is unlikely to be true (see works by Wehrmann amongst many others).

L945-947: Recommend updating these figures with new data available in Mouginot et al. (2019).

References Cited: Bell, R. G. (1994). Behaviour of dissolved silica, and estuarine/coastal mixing and exchange processes at Tairua Harbour, New Zealand. New Zealand Journal of Marine and Freshwater Research, 28(1), 55–68. https://doi.org/10.1080/00288330.1994.9516596

Bhatia, M. P., Kujawinski, E. B., Das, S. B., Breier, C. F., Henderson, P. B., & Charette, M. A. (2013). Greenland meltwater as a significant and potentially bioavailable source of iron to the ocean. Nature Geoscience, 6(4), 274–278. https://doi.org/10.1038/ngeo1746

Buongiorno, J., Herbert, L. C., Wehrmann, L. M., Michaud, A. B., Laufer, K., Røy, H., ... Lloyd, K. G. (2019). Complex Microbial Communities Drive Iron and Sulfur Cycling in TCD
Arctic Fjord Sediments. Applied and Environmental Microbiology, 85(14), e00949-19. https://doi.org/10.1128/AEM.00949-19

Burton, J. D., Liss, P. S., & Venugopalan, V. K. (1970). The Behaviour of Dissolved Silicon During Estuarine Mixing I. Investigations in Southampton Water. ICES Journal of Marine Science, 33(2), 134–140. https://doi.org/10.1093/icesjms/33.2.134

Cai, W.-J., Hu, X., Huang, W.-J., Jiang, L.-Q., Wang, Y., Peng, T.-H., & Zhang, X. (2010). Alkalinity distribution in the western North Atlantic Ocean margins. Journal of Geophysical Research: Oceans, 115(C8). https://doi.org/10.1029/2009JC005482

Cloern, J. E., Jassby, A. D., Schraga, T. S., Nejad, E., & Martin, C. (2017). Ecosystem variability along the estuarine salinity gradient: Examples from long-term study of San Francisco Bay. Limnology and Oceanography, 62(S1), S272–S291. https://doi.org/10.1002/lno.10537

Edmond, J. M., Spivack, A., Grant, B. C., Ming-Hui, H., & Zexiam; Chen Sung; Zeng Xiushau, C. (1985). Chemical dynamics of the Changjiang estuary. Continental Shelf Research, 4(1), 17–36. https://doi.org/https://doi.org/10.1016/0278-4343(85)90019-6

Eiriksdottir, E. S., Gislason, S. R., & Oelkers, E. H. (2015). Direct evidence of the feedback between climate and nutrient, major, and trace element transport to the oceans. Geochemica et Cosmochimica Acta, 166, 249–266.

Eyre, B., & Balls, P. (1999). A comparative study of nutrient behavior along the salinity gradient of tropical and temperate estuaries. Estuaries, 22(2A), 313–326. https://doi.org/Doi 10.2307/1352987

Fellman, J. B., Hood, E., Raymond, P. A., Hudson, J., Bozeman, M., & Arimitsu, M. (2015). Evidence for the assimilation of ancient glacier organic carbon in a proglacial stream food web. Limnology and Oceanography, 60(4), 1118–1128. https://doi.org/10.1002/lno.10088

Foresta, L., Gourmelen, N., Weissgerber, F., Nienow, P., Williams, J. J., Shepherd, A.,
... Plummer, S. (2018). Heterogeneous and rapid ice loss over the Patagonian Ice Fields revealed by CryoSat-2 swath radar altimetry. Remote Sensing of Environment, 211, 441–455. https://doi.org/https://doi.org/10.1016/j.rse.2018.03.041

Grimm, C., Martinez, R. E., Pokrovsky, O. S., Benning, L. G., & Oelkers, E. H. (2019). Enhancement of cyanobacterial growth by riverine particulate material. Chemical Geology, 525, 143–167. https://doi.org/https://doi.org/10.1016/j.chemgeo.2019.06.012

Gruber, C., Harlavan, Y., Pousty, D., Winkler, D., & Ganor, J. (2019). Enhanced chemical weathering of albite under seawater conditions and its potential effect on the Sr ocean budget. Geochimica et Cosmochimica Acta. https://doi.org/https://doi.org/10.1016/j.gca.2019.06.049

Hågvar, S., Ohlson, M., & Brittain, J. E. (2016). A melting glacier feeds aquatic and terrestrial invertebrates with ancient carbon and supports early succession. Arctic, Antarctic, and Alpine Research, 48(3), 551–562. https://doi.org/10.1657/AAAR0016-027

Hawkings, J. R., Benning, L. G., Raiswell, R., Kaulich, B., Araki, T., Abyaneh, M., ... Tranter, M. (2018). Biolabile ferrous iron bearing nanoparticles in glacial sediments. Earth and Planetary Science Letters, 493. https://doi.org/10.1016/j.epsl.2018.04.022

Hendry, K. R., Huvenne, V. A. I., Robinson, L. F., Annett, A., Badger, M., Jacobel, A. W., ... Malcolm S. Woodward, E. (2019). The biogeochemical impact of glacial meltwater from Southwest Greenland. Progress in Oceanography, 102126. https://doi.org/https://doi.org/10.1016/j.pocean.2019.102126

Henkel, S., Kasten, S., Hartmann, J. F., Silva-Busso, A., & Staubwasser, M. (2018). Iron cycling and stable Fe isotope fractionation in Antarctic shelf sediments, King George Island. Geochimica Et Cosmochimica Acta, 237, 320–338. https://doi.org/https://doi.org/10.1016/j.gca.2018.06.042

Hudson, B., Overeem, I., McGrath, D., Syvitski, J. P. M., Mikkelsen, A., & Hasholt, B.
(2014). MODIS observed increase in duration and spatial extent of sediment plumes in Greenland fjords. The Cryosphere, 8(4), 1161–1176. https://doi.org/10.5194/Tc-8-1161-2014

Jeandel, C., & Oelkers, E. H. (2015). The influence of terrigenous particulate material dissolution on ocean chemistry and global element cycles. Chemical Geology, 395, 50–66. https://doi.org/10.1016/j.chemgeo.2014.12.001

Johnson, K. S., Gordon, R. M., & Coale, K. H. (1997). What controls dissolved iron concentrations in the world ocean? Marine Chemistry, 57(3–4), 137–161. https://doi.org/Doi 10.1016/S0304-4203(97)00043-1

Jones, M. T., Gislason, S. R., Burton, K. W., Pearce, C. R., Mavromatis, V., Pogge von Strandmann, P. A. E., & Oelkers, E. H. (2014). Quantifying the impact of riverine particulate dissolution in seawater on ocean chemistry. Earth and Planetary Science Letters, 395, 91–100. https://doi.org/http://dx.doi.org/10.1016/j.epsl.2014.03.039

Kamatani, A., & Takano, M. (1984). The behaviour of dissolved silica during the mixing of river and sea waters in Tokyo Bay. Estuarine, Coastal and Shelf Science, 19(5), 505–512. https://doi.org/https://doi.org/10.1016/0272-7714(84)90012-X

Kranzler, C, Lis, H., Shaked, Y., & Keren, N. (2011). The role of reduction in iron uptake processes in a unicellular, planktonic cyanobacterium. Environmental Microbiology, 13(11), 2990–2999. https://doi.org/Doi 10.1111/J.1462-2920.2011.02572.X

Kranzler, Chana, Kessler, N., Keren, N., & Shaked, Y. (2016). Enhanced ferrihydrite dissolution by a unicellular, planktonic cyanobacterium: A biological contribution to particulate iron bioavailability. Environmental Microbiology, n/a-n/a. https://doi.org/10.1111/1462-2920.13496

Li, J., Chen, J., Ni, S., Tang, L., & Hu, X. (2019). Long-term and inter-annual mass changes of Patagonia Ice Field from GRACE. Geodesy and Geodynamics, 10(2), 100–109. https://doi.org/https://doi.org/10.1016/j.geog.2018.06.001
Marsay, C. M., Barrett, P. M., McGillicuddy Jr., D. J., & Sedwick, P. N. (2017). Distributions, sources, and transformations of dissolved and particulate iron on the Ross Sea continental shelf during summer. Journal of Geophysical Research: Oceans, 122(8), 6371–6393. https://doi.org/10.1002/2017JC013068

Meybeck, M. (1982). Carbon, Nitrogen, and Phosphorus Transport by World Rivers. American Journal of Science, 282(4), 401–450.

Overeem, I., Hudson, B. D., Syvitski, J. P. M., Mikkelsen, A. B., Hasholt, B., van den Broeke, M. R., ... Morlighem, M. (2017). Substantial export of suspended sediment to the global oceans from glacial erosion in Greenland. Nature Geoscience, 10, 859–863. https://doi.org/10.1038/ngeo3046https://www.nature.com/articles/ngeo3046#supplementary-information

Mouginot et al. (2019). Forty-six years of Greenland Ice Sheet mass balance from 1972 to 2018. PNAS, 116(19),9239-9244

Pilcher, D. J., Siedlecki, S. A., Hermann, A. J., Coyle, K. O., Mathis, J. T., & Evans, W. (2018). Simulated Impact of Glacial Runoff on CO2 Uptake in the Gulf of Alaska. Geophysical Research Letters, 45(2), 880–890. https://doi.org/10.1002/2017GL075910

Ragueneau, O., Lancelot, C., Egorov, V., Vervlimmeren, J., Cociasu, A., Déliat, G., ... Cauwet, G. (2002). Biogeochemical Transformations of Inorganic Nutrients in the Mixing Zone between the Danube River and the Northwestern Black Sea. Estuarine, Coastal and Shelf Science, 54(3), 321–336. https://doi.org/https://doi.org/10.1006/ecss.2000.0650

Richter, A., Groh, A., Horwath, M., Ivins, E., Marderwald, E., Hormaechea, L. J., ... Dietrich, R. (2019). The Rapid and Steady Mass Loss of the Patagonian Icefields throughout the GRACE Era: 2002–2017. Remote Sensing, Vol. 11. https://doi.org/10.3390/rs11080909

Schroth, A W, Crusius, J., Chever, F., Bostick, B. C., & Rouxel, O. J. (2011). Glacial
influence on the geochemistry of riverine iron fluxes to the Gulf of Alaska and effects of deglaciation. Geophysical Research Letters, 38(16), L16605. https://doi.org/Artn L16605Doi 10.1029/2011gl048367

Schroth, Andrew W, Crusius, J., Hoyer, I., & Campbell, R. (2014). Estuarine removal of glacial iron and implications for iron fluxes to the ocean. Geophysical Research Letters, 41, 3951–3958. https://doi.org/10.1002/2014GL060199

Seifert, M., Hoppema, M., Burau, C., Elmer, C., Friedrichs, A., Geuer, J. K., ... Iversen, M. H. (2019). Influence of Glacial Meltwater on Summer Biogeochemical Cycles in Scoresby Sund, East Greenland . Frontiers in Marine Science , Vol. 6, p. 412. Retrieved from https://www.frontiersin.org/article/10.3389/fmars.2019.00412

Sharp, M., & Tranter, M. (2017). Glacier biogeochemistry. Geochemical Perspectives. https://doi.org/10.7185/geochempersp.6.2

Shiller, A. M. (2003). Syringe Filtration Methods for Examining Dissolved and Colloidal Trace Element Distributions in Remote Field Locations. Environmental Science & Technology, 37(17), 3953–3957. https://doi.org/10.1021/es0341182

Shoenfelt, E. M., Sun, J., Winckler, G., Kaplan, M. R., Borunda, A. L., Farrell, K. R., ... Bostick, B. C. (2017). High particulate iron(II) content in glacially sourced dusts enhances productivity of a model diatom. Science Advances, 3(6). https://doi.org/10.1126/sciadv.1700314

St. Pierre, K. A., St. Louis, V. L., Schiff, S. L., Lehnherr, I., Dainard, P. G., Gardner, A. S., ... Sharp, M. J. (2019). Proglacial freshwaters are significant and previously unrecognized sinks of atmospheric CO&It;sub>2&It;/sub> Proceedings of the National Academy of Sciences, 201904241. https://doi.org/10.1073/pnas.1904241116

Statham, P J, Skidmore, M., & Tranter, M. (2008). Inputs of glacially derived dissolved and colloidal iron to the coastal ocean and implications for primary productivity. Global Biogeochemical Cycles, 22(3), GB3013. https://doi.org/Artn Gb3013Doi TCD
**10.1029/2007gb003106**

Statham, Peter J. (2012). Nutrients in estuaries — An overview and the potential impacts of climate change. Science of The Total Environment, 434(0), 213–227. https://doi.org/10.1016/j.scitotenv.2011.09.088

Tagliabue, A., Bowie, A. R., Boyd, P. W., Buck, K. N., Johnson, K. S., & Saito, M. A. (2017). The integral role of iron in ocean biogeochemistry. Nature, 543(7643), 51–59. https://doi.org/10.1038/nature21058

Tedstone, A. J., & Arnold, N. S. (2012). Automated remote sensing of sediment plumes for identification of runoff from the Greenland ice sheet. Journal of Glaciology, 58(210), 699–712. https://doi.org/DOI: 10.3189/2012JoG11J204

Thomas, H., Schiettecatte, L.-S., Suykens, K., Koné, Y. J. M., Shadwick, E. H., Prowe, A. E. F., ... Borges, A. V. (2009). Enhanced ocean carbon storage from anaerobic alkalinity generation in coastal sediments. Biogeosciences, 6(2), 267–274. https://doi.org/10.5194/bg-6-267-2009

Tranter, M., Brown, G., Raiswell, R., Sharp, M., & Gurnell, A. (1993). A conceptual model of solute acquisition by Alpine glacial meltwaters. Journal of Glaciology, 39, 573–581.

Treguer, P. J., & De La Rocha, C. L. (2013). The World Ocean Silica Cycle. Annual Review of Marine Science, 5, 477–501. https://doi.org/Doi 10.1146/Annurev-Marine-121211-172346

Wadham, J. L., Hawkings, J. R., Tarasov, L., Gregoire, L. J., Spencer, R. G. M., Gutjahr, M., ... Kohfeld, K. E. (2019). Ice sheets matter for the global carbon cycle. Nature Communications, 10(1), 3567. https://doi.org/10.1038/s41467-019-11394-4

Wadham, J. L., Tranter, M., Skidmore, M., Hodson, A. J., Priscu, J., Lyons, W. B., ... Jackson, M. (2010). Biogeochemical weathering under ice: Size matters. Global Biogeochemical Cycles, 24(3), GB3025. https://doi.org/10.1029/2009gb003688

**TCD**
Wehrmann, L. M., Formolo, M. J., Owens, J. D., Raiswell, R., Ferdelman, T. G., Riedinger, N., & Lyons, T. W. (2014). Iron and manganese speciation and cycling in glacially influenced high-latitude fjord sediments (West Spitsbergen, Svalbard): Evidence for a benthic recycling-transport mechanism. Geochimica Et Cosmochimica Acta, 141, 628–655. https://doi.org/http://dx.doi.org/10.1016/j.gca.2014.06.007

---

## Referee Comment (RC2) · Kiefer Forsch (Referee) · 15 Sep 2019

This review was comprehensive in its scope of biogeochemical impacts of freshwater discharge in the cryosphere. Using multiple case studies of Arctic fjords, Hopwood et al. capture the range of biogeochemical settings, and in doing so, identify and summarize multiple drivers for diverse phytoplankton response. The review was written with a broad audience in mind, with detailed discussions and patient explanations. The figures aided the discussion and were generally appropriate to the text. I support the publication of this much-needed review pending the appropriate revisions are made.

I feel the authors were diligent in their discussion of state-of-knowledge and take a conservative stance when estimating fluxes of dissolved nutrients. I support this approach. However, I am cautious about language which aims to describe ecosystem function as similar in both the Arctic and Antarctic. Few studies exist which focus on the ice-ocean interface (within 1km of marine-terminating glaciers) in the Antarctic. The geochemical gradients are intense here and logistically more challenging to study. It is apparent to me that the Antarctic lacks a robust assessment of the fjords, and so the authors should acknowledge that comparatively less is known about the Antarctic.

I think the authors should include in their discussion mention of katabatic wind events and the efficiency at which they mix the upper water column, and the result this would have on export of the surface layer and upwell subsurface sources (see Lundesgaard et al. 2019). Katabatic wind events are important interactions between the atmosphere and ice sheets.

Lastly, I am pleased with the discussion about new approaches being required to address these highly dynamic ecosystems. Namely, higher resolution (in space and time) studies are needed to understand how this system function and will respond to climate forcing.

Specific comments:

L281-282: I do not think we have a well-constrained estimate for the Antarctic. Subglacial discharge is one of the critical fluxes discussed in this review. Recent attention has been given to the subglacial environment and I think it is worth mentioning the uncertainty which surrounds this source. There are biotic and abiotic factors which influence the quality and quantity of iron released to the ocean. Weathering rates are controlled in part by regional geology, but also the microbial communities (namely, chemolithoautotrophs) and exposure to oxygen may be important controls. (Wadham et al. 2010; Tranter, Skidmore, and Wadham 2005)

Further, it is nearly impossible to differentiate the effects of tidal uplift, sediment resuspension, glacial calving and subsequent scouring of the sediments at the glacier

terminus from purely subglacial discharge. Our understanding of these effects would be greatly increased if measurements were made proximal to cold-based, low velocity marine-terminating glaciers. We can then begin to pick apart the contributions of these different processes.

L288: Seasonal variation may be an important theme for future directions, both in the Arctic and Antarctic. The authors make this note. Without the aid of the ocean modeling community, we do not yet know how subglacial discharge responds to climate forcing.

My general feeling is that while the comparisons may be obvious, there are important functional differences between the Antarctic and Arctic. And so supporting information should be appropriate. For example L203-206 has two well-known studies of particulate iron in the Antarctic (Gerringa et al., 2012, and Annett et al., 2017). The authors may choose to mention this is an important question in general for particle-enriched iron sources. (Fitzsimmons et al. 2017)

L192-195: I think it is important here to discuss the potential for dissolved-particle exchange, facilitated by undersaturated organic ligands or by dissolution in the guts of zooplankton. (Gledhill and Buck 2012; Barbeau et al. 1996)

L452-455: This discussion is accurate, however nutrient stoichiometry (both supply and demand) is what drives primary production and selects for specific phytoplankton taxa, especially in enriched environments. In the instance of diatoms, the N:Fe ratio is a good predictor of iron limitation, where a threshold describes the point at which diatoms begin to grow sub-optimally. The application of geochemical proxies (N:Fe, Siex) for nutrient stress should be applied where such data exists (see King and Barbeau 2007; Hogle et al. 2018).

L511-512: This is indicative of Fe-limitation of the phytoplankton community, which is dominated by diatoms during the sampled summer growth season. Please indicate this is a log-transformed ratio.

L496: "islands occur within"

The phytoplankton community must meet several requirements for a pronounced increase in growth to occur. They must be physiologically adapted to use glacially-derived iron sources. It is unknown the degree to which phytoplankton in the Antarctic use colloidal iron, which would require biotic and abiotic processes to transform it in to a bioavailable form (ie organic complexation, dissolution, photoreduction). I challenge the simplistic view of HNLCs and acknowledge this to be a grand question of our time.

L584-596: This is a great discussion on the uncertainties which remain largely in marine iron biogeochemistry.

L672: "...additional subsidies of labile carbon..."

L731: Our data for Antarctica is spares, and biased towards summer growth periods. We have little information about the community dynamics throughout the ice-free growth season.

L742: We see the same in Antarctic fjords, but lack an early Spring diatom bloom. Instead, flagellates dominate the fjords. A pronounced diatom bloom and sedimentation event spans ~2 weeks, and overall production falls dramatically early-Fall.

L758: "of Patagonia"

L792: It is becoming more apparent that fjords in the Antarctic are highly productive relative to their Arctic counterparts. Primary production in the fjords rivals that of the Fe-limited shelf regions during the summer. Indeed, we find that organic carbon export is greatest in the inner-fjord environment (unpublished). This is more evidence of the differences in behavior between the Arctic and Antarctic.

L819-822: How then do we reconcile the expansion of the icesheets and the decreased availability of sediments eroded by wind?

L862-863, 865: Autonomous gliders with optical backscatter and seawater sampling

capabilities would we a great way to begin to address this. I agree!

References:

Barbeau, K, J W Moffett, D A Caron, P L Croot, and D L Erdner. 1996. "Role of Protozoan Grazing in Relieving Iron Limitation of Phytoplankton." Nature 380 (6569): 61–64. https://doi.org/10.1038/380061a0.

Fitzsimmons, Jessica N, Seth G John, Christopher M Marsay, Colleen L Hoffman, Sarah L Nicholas, Brandy M Toner, Christopher R German, and Robert M Sherrell. 2017. "Iron Persistence in a Distal Hydrothermal Plume Supported by Dissolved-Particulate Exchange." Nature Geosci 10 (3): 195–201. http://dx.doi.org/10.1038/ngeo2900.

Gledhill, Martha, and Kristen Buck. 2012. "The Organic Complexation of Iron in the Marine Environment: A Review ." Frontiers in Microbiology . http://journal.frontiersin.org/article/10.3389/fmicb.2012.00069. Hogle, Shane L., Christopher L. Dupont, Brian M. Hopkinson, Andrew L. King, Kristen N. Buck, Kelly L. Roe, Rhona K. Stuart, et al. 2018. "Pervasive Iron Limitation at Subsurface Chlorophyll Maxima of the California Current." Proceedings of the National Academy of Sciences of the United States of America. https://doi.org/10.1073/pnas.1813192115.

King, Andrew L., and Katherine Barbeau. 2007. "Evidence for Phytoplankton Iron Limitation in the Southern California Current System." Marine Ecology Progress Series 342: 91–103. https://doi.org/10.3354/meps342091.

Lundesgaard, Øyvind, Brian Powell, Mark Merrifield, Lisa Hahn-Woernle, and Peter Winsor. 2019. "Response of an Antarctic Peninsula Fjord to Summer Katabatic Wind Events." Journal of Physical Oceanography. https://doi.org/10.1175/jpo-d-18-0119.1.

Tranter, Martyn, Mark Skidmore, and Jemma Wadham. 2005. "Hydrological Controls on Microbial Communities in Subglacial Environments." Hydrological Processes. https://doi.org/10.1002/hyp.5854.

Wadham, J. L., M. Tranter, M. Skidmore, A. J. Hodson, J. Priscu, W. B. Lyons, M. Sharp, P. Wynn, and M. Jackson. 2010. "Biogeochemical Weathering under Ice: Size Matters." Global Biogeochemical Cycles 24 (3). https://doi.org/10.1029/2009GB003688.

---

## Author Comment (AC1) · 22 Oct 2019

The reviewer is thanked for detailed and thorough comments on the text. A point-by-point response to issues raised follows. As some general issues are raised across multiple comments, a few key topics are discussed here first which we will refer back to.

**(a) Linking glacier particulate nutrient fluxes to primary production**

The issue of links between glacier-derived particles and marine primary production in environments downstream of glaciers has been raised extensively in several recent papers around Greenland (Hawkings et al., 2016, 2017; Wadham et al., 2019), implying that glacier-derived particles contain 'bioavailable' nutrients and are contributing to 'high productivity' in meltwater affected waters. But where is a positive link between glacier-derived particles and Arctic marine primary production (PP) established?

For example, looking at the recent (Wadham et al., 2019) review, which focuses extensively on glacier particulate fluxes where discharge enters the ocean (i.e. near zero salinity) and does a good job of bringing together geochemical work in this field, the justification for the conceptual link between glacier-derived particles and Arctic marine PP is summarised in the following paragraph, which underpins the large emphasis on these fluxes in recent manuscripts and their link to the global C cycle:

*"Observations of heightened biological activity in marine waters surrounding glaciers has been noted as early as 1938, when brown zones (representing turbid melt plumes) in front of tidewater glaciers were noted to be particularly productive regions for biota[70,71], with more recent research reinforcing this connection[10,72]."*

But the references in this paragraph do not support this statement with respect to primary producers — those in the Arctic show the exact opposite. The Lydersen et al. (2014) reference states that glacier-associated particle plumes downstream of Arctic glaciers negatively affect marine primary production. The argument presented in Lydersen et al. (2014) is that secondary production (not primary production) is high around glacier plumes in Arctic fjords because microbial biota are 'stunned' by the stressful, turbulent conditions in these plumes, which then make easy prey for higher-level organisms. This is consistent with all other literature in the study location (Kongsfjorden — a site we already discuss extensively). Primary production measurements, and extensive chl a data, show these 'brown zones' are low PP environments (e.g., Hop et al., 2002). The Meire et al.(2017) reference similarly does not support the statement, as the lowest primary production in the region studied (W Greenland) is associated with heavy particle inputs. This paper attributes the increased primary production in some glacier fjords solely to upwelling, not freshwater runoff and/or associated particles— which is suggested to have no clear significant positive effect on productivity assessed in terms of higher trophic levels. The other references referred to (and references therein) either discuss Antarctic catchments (where a fertilizing effect arises from Fe, although in early references this is not extensively discussed correctly as it wasn't widely accepted until the 1980/90s that Fe-limitation was extensive across the Southern Ocean), or the upwelling effect. As acknowledged later in the review, there is no explicit evidence of a link between glacier derived particles and increased marine PP around Greenland. Yet data is available to assess this, so we should perhaps use it herein.

It is challenging to produce an Arctic 'average' PP for reference, because PP primarily occurs during intense bloom periods, and thus annual PP averages for any specific area are biased low compared to typical PP during any bloom period. Furthermore, observations in glacier fjords are generally temporally biased towards bloom periods. We can however define two rough reference PP values.

One, a low threshold for 'high' PP (as it integrates high and low PP time periods) is a simple pan-Arctic mean for the period March–September (420 mg C m$^{-2}$ day$^{-1}$) (Pabi et al., 2008) (which we already reference in the original text). This provides a very low benchmark for assessing 'high' PP for an individual PP measurement. A more meaningful comparison, because the limited primary production data downstream of glaciers is generally biased towards summer, and because fjords are shelf environments, is to look at the range of shelf PP across the Arctic in May–August (360–1500 mg C m$^{-2}$ day$^{-1}$) (Pabi et al., 2008) (Figure 1).

[Figure]

*Figure Discussion 1: All available Arctic primary production data by month for regions with glacial influence. Data from* (Harrison et al., 1982; Hodal et al., 2012; Holding et al., 2019; Hop et al., 2002; Iversen and Seuthe, 2011; Jensen et al., 1999; Juul-Pedersen et al., 2015; Levinsen and Nielsen, 2002; Lund-Hansen et al., 2018; Meire et al., 2017; Nielsen, 1999; van de Poll et al., 2018; Rysgaard et al., 1999; Seifert et al., 2019; Smoła et al., 2017; TG and Hansen, 1995)*. Circles represent glacier-fjords, triangles sites beyond glacier-fjords, bold shapes are <80 km from a marine-terminating glacier. Error bars are standard deviations for stations where multiple measurements were made in the same both. Hashed line is the pan-Arctic mean primary production (March-September). Shaded area is the pan-Arctic shelf range of primary production for May-August.*

We summarise this by categorising data according to whether PP was determined close to a marine-terminating glacier, or within a glacier-fjord producing the following statistics for the peak of the discharge season (July-September) (Table 1).

| Category | Mean primary production (± standard deviation) mg C m$^{-2}$ day$^{-1}$ | n | Data from |
|---|---|---|---|
| (I) Marine-terminating glacier influence, non-fjord | 817 ± 892 | 10 | Disko Bay, Scoresby Sund, Glacier Bay, North Greenland, Canadian Arctic Archipelago |
| (II) Marine-terminating glacier influence, glacier fjord | 480 ± 403 | 33 | Godthåbsfjord, Kongsfjorden, Scoresby Sund, Glacier Bay, Hornsund, |
| (III) No marine terminating glacier influence, non-fjord | 305 ± 268 | 29 | Godthåbsfjord, Young Sound, Scoresby Sund, Disko Bay, |

| | | | Canadian Arctic Archipelago |
|---|---|---|---|
| (IV) No marine terminating glacier influence, glacier fjord | 108 ± 88 | 40 | Godthåbsfjord, Young Sound, Kangerlussuaq |

*Table 1. Primary production from studies conducted in glaciated Arctic regions and pooled according to whether <80 km of marine-terminating glacier ('marine-terminating glacier influence'), and whether within a glacier fjord. Data sources as per Figure 2. n = number of data points, where studies report primary production measurements at the same station for the same month in multiple years (e.g. Juul-Pedersen et al., 2015) a single mean is used in the data compilation (i.e. n=1 irrespective of the historical extent of the time series).*

| P values | I | II | III |
|---|---|---|---|
| IV | <0.001 | 0.001 | 0.153 |
| III | 0.001 | 0.211 | |
| II | 0.071 | | |

Table 2. P values for inter-group comparisons (for Table 1)

It is apparent that glaciated regions are low productivity environments during the meltwater season with the exception of sites close to marine-terminating glaciers. Higher PP is generally found over the shelf, beyond the zone over which extensive sedimentation has occurred. Inner-fjord environments, with strong freshwater glacier plumes where glacier particles remain largely in suspension, are almost universally low PP (Lund-Hansen et al., 2018; Meire et al., 2017), and include the lowest Arctic coastal PP values measured (Holding et al., 2019). Inter-fjord variability is inevitably an issue and hence why we opted for the structure of the review considering the best-studied cases. But we note that even in the fjord region where the most extensive comments have been written about the potential positive fertilizing effects of particles downstream of glaciers (Kangerlussuaq (e.g. Hawkings et al., 2015, 2017; Wadham et al., 2016)), we see low PP during the meltwater season (Lund-Hansen et al., 2018) which is explicitly linked to the negative effects of glacier-derived particle plumes. Over the spatial range where these particles are visibly present in the water column (i.e. prior to extensive sedimentation improving water clarity), the photic zone is diminished, nutrient uptake is reduced, and chlorophyll a and primary production are low (Murray et al., 2015). Irrespective of their geochemistry, the general effect of glacier particle plumes is a suppression of marine primary production on this spatio-temporal scale, so we are unclear why such emphasis is being placed on the importance on a potential fertilizing effect of these plumes.

If Arctic glacier fjords were included as an ecological province alongside shelf/open waters/etc. (e.g. as per Pabi et al., 2008), they would (according to literature data), be the lowest PP province in the Arctic. We therefore disagree with some of the reviewers' comments on this topic, including those alluding to particulate fluxes summarised by (Wadham et al., 2019). This is problematic from the perspective of writing a review. It would be difficult to focus extensively on the geochemistry of glacier-derived particles across the Arctic and how they could fuel marine primary production, only to go on to show that the major effect of these particles determined from oceanographic data is light limitation (or other, less well characterised, negative effects).

**(b) The effect of stratification and the influence on flux gate calculations**

Similarly, some recent studies, as summarised by (Wadham et al., 2019), have argued that the nutrient flux from Greenland is larger than currently considered in the oceanographic community and have taken a freshwater orientated approach, typically multiplying the concentration of labile particulate phases in freshwater by freshwater discharge volume to define a flux into the marine environment. For inter-catchment comparisons this is perfectly valid. Yet concerning the net change

in PP, the only change that matters is the net change in nutrient availability in the surface mixed layer or photic zone over the growth season. This is not the same thing as the flux flowing out of a glacier precisely because freshwater changes the depth of the surface mixed layer, increases stratification, and generally decreases the depth of the photic zone. This is a well-recognised problem in the Arctic as it affects all freshwater nutrient 'sources' (McClelland et al., 2011a).

[Freshwater volume × freshwater concentration] will always produce a positive flux of any nutrient, but the net change in the availability of that summertime nutrient supply downstream can (and often is) still negative because of the effect of freshwater stratifying marine waters. Therefore comments concerning how marine PP will change based on large freshwater derived fluxes alone [freshwater volume × freshwater concentrations] are usually misleading as they often do not capture the direction of change (dilution of macronutrients within surface waters and loss of vertical supply) concerning the availability of a nutrient to marine primary producers. A flux gate at the point of discharge into the ocean can therefore be used for inter-catchment comparisons, but it can rarely be used to make comments about changing Arctic marine primary production. We have therefore deliberately avoided commenting extensively on work which falls into this category.

We do not dispute that freshwater contains macronutrients, or that freshwater contains labile particulates, but all of the field evidence we can find shows unambiguously that these 'direct' inputs are associated with negative (or no) changes to Arctic marine primary production and negative (or very limited) changes to regional Arctic nutrient budgets. Any plausible exceptions (which are limited in the Arctic) and reasons for this are already highlighted in the text (alongside comments on stratification and light limitation and a brief contrast with the Southern Ocean where Fe-fuelled phytoplankton blooms downstream of glaciers are well documented).

**(c) Calculating nutrient fluxes from particle dissolution**

Glacier-to-ocean fluxes for non-conservative components (e.g. Fe and Si) are more challenging to determine compared to generally conservative components because of the pronounced changes that occur in concentration and speciation across the estuarine salinity gradient. There are several approaches to deducing the flux from a glacier we can find in the literature to date:

1. Measuring freshwater dissolved and particulate concentrations and then using an estimated removal or dissolution factor to account for the extent to which the dissolved and particle phases interact over the estuarine gradient (e.g. a removal factor for Fe, or a dissolution factor for amorphous Si).
2. Using an estuarine mixing diagram to deduce the estuarine endmember which, in simple terms, accounts for the net dissolution/removal occurring across the salinity gradient and produces an effective concentration that can be multiplied by freshwater discharge volume to produce a flux.
3. A water mass analysis to determine the relative enrichment of a parameter in glacially-modified waters (by contrasting properties in a water mass before and after the addition of glacial meltwater).

All of these approaches can be used to derive a flux into the ocean. Each has advantages and disadvantages, and is subject to different data requirements. (1) Requires zero salinity dissolved and particulate data, and some knowledge of how the dissolved and particulate phases equilibrate over the salinity gradient. This method is therefore more problematic for estuaries with multiple freshwater endmembers and for parameters where the estuarine behaviour of a chemical is spatially/temporally variable. (2) Requires data across the salinity gradient, works particularly well if

the non-conservative effects are confined to low-salinity values, but is more challenging to apply if multiple processes are occurring on different scales. (3) Can be conducted only with extensive water column profiles and a water mass analysis, with most of the endmembers well defined in terms of their chemical properties prior to mixing.  This is the most conclusive approach in terms of determining the net input of any chemical into the ocean, but also has the highest data requirement.

Using Si as an example, we can illustrate the present problem in the literature with existing fluxes.

Taking the Kongsfjorden Si data (in Figure 2) as an example, where both estuarine and freshwater dissolved Si endmembers have been determined, approach (2) suggests an estuarine endmember of 6.4 µM. An enrichment factor of 13% can then be calculated for use with method (1) by comparing 6.4 µM to measured freshwater endmembers.

For the larger catchments discussed, the same approach (method 2) can be applied to derive estuarine endmembers of 25 µM (Godthabsfjord), 29.1 µM (Gulf of Alaska), 8.49 µM (Bowdoin) and 7.47 µM (Sermilik) (Table 3) (Brown et al., 2010; Cape et al., 2019; Kanna et al., 2018; Meire et al., 2016). The largest caveats with this method are that it may cause an overestimate in catchments where strong diatom drawdown of dissolved Si occurs in the higher-salinity end of an estuary or an under-estimate where strong drawdown occurs over the full salinity gradient, and that it is generally conducted only for surface waters (thus, if there are large differences in the dissolved Si behaviour between runoff entering at the fjord surface, and subglacial discharge entering sub-surface layers, these endmember values only apply to the surface runoff component).

The only complete application of method 3 we can find in the literature is for Sermilik (Cape et al., 2019) where, in simple terms, the upstream, downstream, and fjord profiles of macronutrients ($NO_3$/$PO_4$/Si) are considered in order to deduce the enrichment arising from glacial meltwater. Fitting to observed export out of the fjord is consistent with a Si freshwater endmember of 4–10 µM. Because the fit is forced with all macronutrients, the Si endmember cannot be much higher than this, as the fit solution must account for both the observed Si, and $NO_3$ and $PO_4$ enrichments which arise from marine endmembers and don't include significant particle-dissolution on the same timescale as Si. The Meire et al. (2016) approach is a hybrid use of methods 2/3, considering the local perturbations to Si with some knowledge of freshwater Si endmembers producing an estuarine endmember of 28.7 µM. Method 3 can also be applied to extensive data from the Geotraces process study at the 79° North glacier, which is now available online[1] (we don't discuss this data because, unlike other catchments herein, hasn't been extensively discussed in the literature- but we note the concentrations of Si close to the glacier terminus are similar to, or lower than, for Sermilik and no discernible Si enrichment is evident in glacially modified water at the shelf-break).

Work on Si for Kangerlussuaq (Hawkings et al., 2017) uses method (1), with a dissolution factor for labile particulate Si of 100%. The freshwater endmember then becomes 395 µM. Applying the other methods to this fjord region is challenging given the limited data available, but if we plot the few estuarine datapoints available (Hawkings et al., 2017; Lund-Hansen et al., 2018) they are not dissimilar from other glacier fjords (Figure 3) — although (Hatton et al., 2019) implies they are. The saline endmember selected by (Hawkings et al., 2017) is odd as it refers to offshore surface seawater, but should refer to inflowing sub-surface seawater (summertime fjord circulation involves inflow at depth and net outflow at the surface). In any case, plotting either a 2 µM saline endmember as used (Hawkings et al., 2017), or a more realistic estimate of 10 µM based on other Greenland fjord systems, the maximum possible intercept at zero salinity (using method 2) is 25—30
* * *
[1] https://doi.pangaea.de/10.1594/PANGAEA.884128 and https://doi.pangaea.de/10.1594/PANGAEA.879197

µM. This would suggest a dissolution factor of 6—7% for the purpose of applying method 1. As noted by the reviewer, some incubation results are used to support the claim of 100% dissolution in the manuscript, but we would ask how relevant the timescale of the incubation (and incubation conditions with particles kept in suspension during this time) are to a fjord environment in which glacier particles are subject to rapid sedimentation? We suggest there is much work to be done to reconcile this high estimate of Si export with other work around the Arctic (Table 3). The simplest explanations that could be proposed for why this value is so high are that either it wasn't intended to construct an annual flux (i.e. represents Si dissolving on slower timescales), or that it is meant to be inclusive of all subsequent dissolution that could possibly occur from glacier-particles (i.e. direct dissolution whilst in suspension plus any subsequent benthic re-working). But it is apparent from the presentation of data in (Hawkings et al., 2017) and subsequent work by the same group, that neither of these explanations is the case. The flux is presented as an annual flux, discussed in terms of short-term PP, and is added to (no included within) additional benthic fluxes (Hendry et al., 2019).

| Catchment | Method | Enrichment of dissolved Si due to dissolution | Estuarine endmember for flux calculations |
|---|---|---|---|
| Kongsfjorden | 2 | +13% | 6.4 µM |
| Godthabsfjord | 2 | | 25 µM |
| Gulf of Alaska | 2 | | 29 µM |
| Bowdoin | 2 | | 8.5 µM |
| Sermilik | 2 | | 7.5 µM |
| Sermilik | 3 | | 4–10 µM |
| Godthabsfjord | 2/3 | | 28.7 µM |
| Kangerlussuaq | 1 | >150% | 395 µM |
| Kangerlussuaq | 2* (with large error due to limited data) | | 25–30 µM |

*Table 3* To summarize, the estuarine endmember that we would multiply by a Greenland freshwater discharge of x km$^3$ year$^{-1}$, inclusive of dissolved Si and particulate Si that dissolves in estuarine waters. Excluding the 395 µM value, all other work suggests an endmember of 6–30 µM for flux calculations.

We cannot find much discussion that quantitatively explains why this apparently large flux (Table 3) is more appropriate than values that can derived from other studies, which are universally much lower. As noted, much more extensive Si coverage is available for some other fjords (Kongsfjorden/Bowdoin/Godthabsfjord/Young Sound/79°NG) (Cape et al., 2019; Kanna et al., 2018; Meire et al., 2016). It is unclear why oceanographic data from these catchments, which is far more extensive, hasn't been used to test the hypothesis that global Si cycling is 'missing' a substantial glacial component given that now extensive literature has been written about this hypothesis (Hatton et al., 2019; Hawkings et al., 2017; Hendry et al., 2019; Wadham et al., 2019). We therefore maintain that our discussion of these cycles is balanced until a clearer reconciliation between 'high' estimates of fluxes can be made with large datasets.

**Detailed comments on review**

*This review article is timely and presents an opportunity for the authors to summarise the current "state of play" and highlight potential future research direction in the Arctic with regard to ice-ocean biogeochemical interaction. I broadly agree with most of the main points raised and the authors touch on most of the key research areas. The review is well written, the figures largely appropriate (apart from some points below) and I'm supportive of its publication after revision. However, I have suggestions for improvement.*

*When and where some of the literature quoted is relatively selective and the way it is contextualised in certain circumstances misses nuance. One major omission is a discussion of particulate fluxes (both as part of nutrient budgets, and importance in ballasting and C burial) and indirect processing of glacial inputs (related to particulate inputs; i.e. benthic recycling and/or burial). Given the context of these environments (dominated by inputs of products of physical weathering), and the existence of literature in other glacially influenced regions (e.g. Laura Wehrmann's and associated groups ongoing work in Svalbard; e.g. Wehrmann et al., 2014), this could have been an opportunity to start a balanced discussion. This is an oversight, especially for a review article, and given recent interest in particulate fluxes (not just in glacial locations), even if the authors do not think these flux terms are important. As a previous reviewer indicated, there is also a need to discuss and incorporate more recent publications (i.e. Hendry et al., 2019, but also Seifert et al., 2019, Wadham et al., 2019 amongst some others I suggest below), and some key papers have been omitted or not referenced where they should have been. I have some major reservations about section 8, which feels incredibly speculative, and think it should be toned down and incorporated into section 9 in a much reduced form. Specific comments to be addressed are below.*

**The reviewer is thanked for detailed comments on the text. A point-by-point response to issues raised follows. We are critical of most recent work in the literature attempting to link particulate nutrient fluxes into the Arctic to marine primary production as a link between the two is not clear to us, primary production (PP) data does not support the hypothesis that glacially derived particles are a positive influence on marine primary production (see new section), it supports the opposite conclusion.**

**Benthic processes/fluxes was something raised during the initial potential list of topics alongside sea-ice, icebergs, glacial dust and others. As can be seen for the length of the review it is simply not possible to cover all of these 'glacier-related' topics in one review. We selected topics covered (and the title) to be those which were specific to the direct effect of freshwater discharge in the ocean, and those which have broad-scale effects. We had noted in the original draft that in some cases (e.g. the case of Kongsfjorden), there is direct coupling between benthic nutrient cycling and pelagic primary production - although we neglected to explicitly label this as a 'benthic' topic. We can rephrase to highlight this issue in general terms and to add some general comments on the interactions between glaciers and benthic recycling in the summary section where we link to work in related areas (dust/icebergs etc). A new sub-section ('**Benthic pelagic-coupling enhanced by subglacial discharge'**) is added following and expanding on the NH$_4$ work of** (Halbach et al., 2019)**.**

**We have of course updated the text with 2019 references in the revised text which were published between the original and revised submission dates. The** (Seifert et al., 2019) **reference is particularly interesting, as the reviewer notes, a lithogenic 'ballasting' effect of particles is widely hypothesized in other contexts, but there wasn't much material in the literature to discuss this specifically with respect to glacier-derived particles at the time we wrote the review (we found only a few references, which lacked much direct evidence and weren't specific to meltwater).**

Replies

*L65: Calcium carbonate is not an ion. This should be corrected to "inorganic salts. . .".*

**L 65 rephrased 'inorganic components'**

*L64-68: These plumes also carry large quantities of reactive particular material, including labile particulate nutrients. Whatever you think of their ultimate fate (which can be discussed) I think this is important to note as it is an important characteristic of glacial meltwaters. In this context I'm sure*

*the authors will be aware of the literature (some suggestions for inclusion are Hendry et al., 2019, Seifert et al., 2019, Jeandel and Oelkers, 2015, Grimm et al., 2019, Schoenfelt et al 2017, Morgan et al., 2014, Eiriksdottir et al., 2015).*

**L64-68 (See general comment) Whilst these 'reactive' fluxes may be of intense interest to glaciologists the main effect of particle plumes on marine primary production (the subject of this review) is light limitation and thus we question how important the lability of these particles is. It would be an odd addition to the paper to discuss in detail the geochemical composition of particles, only to summarize that it doesn't matter in terms of its immediate effect on primary production because of light suppression. Many of these references refer to processes occurring on geological, not inter-annual timescales and are common to all ocean shelves, not specific to glacial-meltwater affected areas, and therefore a little beyond the scope of the current title.**

**Some comments concerning a geological timeframe could be added, but it would be very confusing to mix discussion of processes that affect PP on these two scales, recent literature concerning glacially-derived particles has focused almost entirely on inter-annual timescales which is what we discuss herein. There are fundamental differences in the relative importance of processes operating on these two different timescales which are not (particularly with respect to geological timescales) well agreed upon in the literature** (e.g. see Tyrrell, 1999)**, and as noted, on these timescales the glacier-fjord systems we discuss herein cease to exist as ocean-glacier interfaces. We do not think it would be useful to discuss them together.**

**It is also misleading to refer to labile lithogenic particulates as 'nutrients', as this implies they are actively taken up into biological systems, which is not generally the case either in glacial freshwater or marine systems. 'Labile particulate nutrients' in a marine context refers to organic P/N etc and not lithogenic elemental particles.**

*Figure 1: I'd like to see the quality of this figure improved before publication. As the first figure and a key map of study areas it's also a little too basic at present.*

**A revised figure 1 is added, (this was previously a last minute add-on at the request of the editor).**

[Figure]

*Section 3: I find the referencing in the first paragraph curious. Although by no means do I think that the authors should be referencing some work ahead of others, the first reference of a particular*

*group's work is page 8, where it's critiqued, despite the number of publications from this group that are suitable for referencing before (in this context).*

**Section 3. The reviewer is referring to the very extensive freshwater nutrient work by Bristol University. We remind the reviewer that this is a marine review and thus some work which may be of major importance to freshwater environments is of much less importance in the marine environment. We started the manuscript with every piece of literature we could find concerning the impact of meltwater specifically in the marine environment. Supporting literature concerning ancillary fields of course comes later in the text. If it were the case that freshwater derived nutrient fluxes were particularly large, or a particularly dominant feature of meltwater of course this would come earlier, but we note that the direct input of freshwater constitutes <1% of nitrate inputs into these meltwater affected regions and therefore isn't particularly important for marine PP, and that the dominant effect of freshwater in the marine environment on nutrient availability is stratification (see general comment). It would be very odd to start a review with a detailed discussion of a budget component that makes a negligible effect to Arctic marine PP.**

*The authors discuss the need for seasonal datasets to contextual flux information, yet there are already several studies currently available that contain temporal datasets over several months and several years of monitoring for hydro chemical parameters, macronutrients and Fe. The concentrations used on Table 2 are from some of these studies, and are discharge weighted mean concentrations derived from a seasonal dataset (the only DWM concentrations in Greenlandic meltwaters that I know to exist at present). There is certainly a debate that can be had with regard to the particulate nutrient inputs (which the authors should deal with in a more balanced manner), but I do not fully understand why other aspects of those papers have been overlooked. These might not be datasets that span whole melt seasons (typically early May to early September), but they are the longest available at the moment and should be acknowledged as such. I would like to see the current literature discussed in a more nuanced way in the next version of the manuscript.*

**There are longer datasets (multi-year) available in the marine environment for the case studies investigated (e.g. GEM portal). It is not an argument we agree with that time series of freshwater data (without corresponding marine data) should be used to make key conclusions about the marine environment when freshwater fluxes make a very small (for N/P negligible) component of nutrient input and when estuarine transformations and stratification (see general comment) fundamentally change the effect this material has in the marine environment. The term 'particulate nutrient' used by the reviewer, referring to labile particulate phases, is not used in a marine context because these phases are not actively up-taken by cells in the environments discussed, questioning the extent to which they can be described as 'nutrients'. They are more commonly described as a source of nutrients (Si etc).**

*L163: Semantics but I think this should be "dissolved macronutrients". Again, the role of particulate macronutrients can be critiqued, but this is an important distinction to make. Glacial meltwaters have high concentrations of particulate nutrients (save N), and low concentrations of dissolved nutrients, and it's important to highlight that whatever you think of the eventual fate.*

**L 163 'Labile' elements (rather than organic C/N/P particulates) in particles are not generally considered as nutrients because they are not widely uptaken by cellular processes. With respect to PO4, NO3 and Si, measurements of these compounds are usually conducted unfiltered in the marine environment so it is more common to refer to 'macronutrients' than 'dissolved macronutrients'. Labile particulates cannot be referred to as 'nutrients' unless it can be demonstrated that they actively are taken up into biological systems.**

*L164-166: There is a push here to emphasise that the PO4 concentrations in glacial meltwaters are particularly low. I'm not arguing against this (they are compared to some marine waters), but the PO4 concentrations in glacial meltwaters from large catchments (see Leverett Glacier) are similar to the global river mean (0.32 μM; Meybeck, 1982), and also similar to (or exceeding) PO4 concentrations in Arctic rivers (0.03-0.76 μM). Further, the annual yields (normalised to catchment area) are very high (see Table 4 in Hawkings et al., 2016). Again, not all this information may be needed in the context of the review, but it's important to not single out glacial inputs as being particularly nutrient deplete as is currently done*

**L164 In the context of the ocean, all freshwater (with very few exceptions) is PO4-deficient (and often thought to be PO4-limited) so we are not sure what point is being made here. Meltwater is PO4-deplete as demonstrated by both concentrations and cellular C:P and P:N stoichiometry** (Ren et al., 2019)**. Riverwater contains generally higher concentrations of organic P and thus meltwater is at the low end of freshwater 'bioavailable' PO$_4$ concentrations.**

*L167: This needs a reference and some contextual information. See point above.*

**We can add some general background here. Focusing specifically on Arctic catchments, river estuaries tend to show much higher Si, higher NO3 and similar PO4 concentrations (Cauwet and Sidorov, 1996; Emmerton et al., 2008; Tank et al., 2012). We have replotted the original Figure to include some example Arctic river catchments. It is difficult to select a 'typical' Arctic river, so we selected 3 catchments: the Lena, Mackenzie and Yenisey.**

*L178: I'm not sure if I'd call these measurements "extensive" given they are from two small glaciers in a fjord with many meltwater inputs (the major inputs coming from much larger tidewater glaciers). The references given for studies of Svalbard meltwaters also have listed LoD for PO4 is 5 ppb (0.16 μM), and a limit of quantification likely even higher (although not mentioned) making those figures difficult to compare to the fjord measurements when the LoD is typically better.*

**L178 Re-phrased (the references were given as examples, there are a very large number of references giving freshwater nutrient concentrations for Kongsfjorden). Yes the LOD of PO4 is often problematic in these studies and we suspect if field blanks were properly/consistently reported through the literature the calculated PO4 concentrations in glacial freshwater would change. Whilst we do not particularly want to dig into methodological reviews herein, for Fe, PO4 and Si there are potential well-known problems to raise and so a brief comment on filtration/method artefacts for those compounds/elements where this may be an issue for data quality (Fe/PO4/Si) is now added alongside the data compilation (Tables 2/3 in the text).**

*L188-189: As above point, I don't really understand the referencing here. There are other appropriate studies that emphasise the existence of reactive particulate Fe that should be referenced here (Bhatia et al., 2013, Schroth et al., 2012, Schroth et al., 2014, Hawkings et al., 2014, Hawkings et al., 2018).*

**There is, we thought obviously, a strong bias throughout the text to studies conducted in the marine environment at the key fieldsites mentioned. This is explicit because we want to discuss the effect of meltwater in the marine environment, and it is very difficult to contextualise studied that don't have extensive (or any) marine data. It is also very difficult to contextualise studies that don't have accompanying data concerning salinity and other key parameters available (especially for nutrients like Fe and Si that experience significant modification within estuarine zones). The Hawkings and Bhatia works are freshwater based. The Schroth work is more useful in this context and is extensively discussed extensively concerning estuarine mixing (although the accompanying data is not available online or from the author so we cannot comment in as much depth).**

**To quantify why it is better to use marine/estuarine studies to study Fe/Si in the ocean, consider the following. Estuarine removal flocculates between 60 and 99% of dissolved Fe, which is highly variable between (and even within) different estuarine gradients** (Schroth et al., 2014; Sholkovitz et al., 1978; Zhang et al., 2015)**. Thus the same dissolved Fe concentration measured at zero salinity could plausibly produce values varying by a factor of 40 in saline waters, which is generally much less than seasonal changes in Fe concentrations of any fraction** (Hawkings et al., 2014; Statham et al., 2008)**. Similarly, total Fe shows no straightforward relationship to salinity or to dissolved Fe. Hence it is very difficult to make conclusions about the fate of Fe from freshwater data alone, and improving accuracy in the freshwater endmember doesn't really improve this much, whereas marine studies unambiguously show the actual enrichment irrespective of what the freshwater endmember was.**

*L198-199: Low nM concentrations are still fairly significant in a marine context, especially when 100 km from the main inputs at the head of the fjord. Surface open ocean waters and even some coastal systems are typically <0.5 nM and often much lower (Johnson, Gordon, & Coale, 1997; Tagliabue et al., 2017). These concentrations would usually be considered very high for marine systems - an important point worth making I think*

**Costal Fe values are always high relative to offshore waters, this is not unique to near-glacier systems, and whilst these values are 100 km from the nearest glacier, they are only 1 km from the coastline and much less than this (50-200 m) from the sea-floor making it an interesting assumption that they definitely have a direct meltwater origin. Fe concentrations across a salinity gradient should always be discussed with salinity in mind. Normalised to salinity, dissolved Fe concentrations at these locations are not particularly high. Considering that Arctic concentrations (offshore) peak within the transpolar drift at 4-5 nM dFe (in saline waters) (Rijkenberg et al., 2018; Slagter et al., 2017) , glacier estuary concentrations of 1-3 nM dFe are-perhaps surprisingly- low. Total Fe concentrations are higher, but are more challenging to interpret given that they don't behave conservatively and are less relevant to determining Fe availability to primary producers. A full discussion of the Fe-cycle is beyond the scope of this text given the limited relevance to Arctic primary production, it is of course much more relevant in the context of Fe-limitation immediately adjacent to glaciers in the Southern Ocean, but we have deliberately kept an Arctic focus to avoid getting side-tracked.**

*L205-206: Schroth et al. (2014) should be referenced in this context as well.*

**Yes the study is relevant, although we have tried to limit general points to 3 references (we are already well over the suggested limit) and in this context the Crusius data cited covers the same region more extensively.**

*L211-212: What about biological uptake?*

**This of course results in some drawdown in most environments, hence why we started the sentence 'In the absence of biological processes'**

*L222-230: The first assertation in this paragraph (that Si is generally released from the particulate phase over a salinity gradient) is based on one referenced paper (Windom et al., 1991). Other review articles on estuarine environments (e.g. review article of Statham, 2012) and many other estuarine papers (e.g. Edmond et al., 1985, Burton et al., 1970, Cloern et al., 2017, Bell, 1994, Raguenau et al., 2002 to list a few) note that conservative behaviour, or in some circumstances reverse weathering and/or adsorp-tion/other removal processes have been observed, especially in similar high sediment, deltaic environments (Treguer et al., 2013, Kamatani et al., 1984), apart from when strong benthic Si*

*fluxes have been inferred (e.g. Eyre and Balls, 1999). I'm not saying dissolution of particulate material is not important in other systems, but the authors need more than one reference to support this generalisation.*

**L222- added. We also note the salinity range and spatial scale. Si dissolution is normally evident only at low salinities across rapidly changing salinity gradients, which are often missed or poorly sampled in estuarine studies, e.g. (Brown et al., 2010) shows conservative behaviour of Si in a glacier catchment, but this concerns salinities of >25. (Windom et al., 1991) is quite nice in actually capturing the low-S changes in multiple rivers, few of the other suggest references do this. We have deliberately avoided estuaries with strong anthropogenic influences as this is beyond the scope of the text, but can add a few more sentences discussing observed Si trends in other estuaries. There are adequate Arctic studies to show varying degrees of (non-)conservative behaviour (Cauwet and Sidorov, 1996; Emmerton et al., 2008). A paragraph here is added to discuss differences as, as alluded to, there are differences between different rivers and thus showing a single 'typical' river scenario is not possible. As requested earlier, we have also added 3 different Arctic rivers into the salinity vs nutrients plot to show the range of concentrations/estuarine behaviour across multiple systems.**

*L230-234: I welcome balanced debate, however, it's disappointing the review makes no mention of incubation experiments performed in this study, which show release of DSi from particulate material to seawater over a period of 30 days in samples that weren't treated to remove ASi. This doesn't necessarily mean DSi is released in the fjord surface, but it's worth consideration especially given the recent findings of Hendry et al. (2019) and Gruber et al. (2019) among others. The former shows strong evidence bottom water modification for example. The benthic environment is currently ignored and the lack of discussion of this is an oversight.*

**L230- We are not disputing that some particulate Si is released from particles, this is beyond doubt and evident from the shape of the Si/salinity curves. (See general points). The incubation experiments in question are extrapolated over several times the time period over which these particles would remain in suspension in a glacier fjord. It is very difficult to reconcile with large datasets elsewhere.**

**As the Hawkings et al flux is inclusive of 100% dissolution of the annual labile particulate Si discharge we would think this already includes any directly related benthic flux, but this is not what the later work from this group suggests** (Hendry et al., 2019) **and thus we find these studies hard to understand and to reconcile with large Si datasets around Greenland.**

**A sub-section has been added to expand on the direct benthic-pelagic linkages highlighted by** (Halbach et al., 2019) **where we now also mention briefly more general benthic processes affected by glaciers (i.e. high sedimentation), yet benthic cycling is not unique to environments affected by meltwater so it would be beyond the scope of the review to extensively cover benthic pelagic processes (as per dust, icebergs and sea-ice), especially looking at shelf environments over long timescales (geological rather than seasonal/interannual). We explicitly titled and focused the review on 'meltwater' to keep a tight focus.**

*L242-254 (and Figure 3): I don't disagree with most of the interpretation here, but given that some of the low salinity end members are not dissimilar to Hawkings et al. (2017) (where there are no high salinity end members) it seems curious that the authors explain this by lack of data and complexity of fjord systems in these instances. Simply drawing linear regressions through points in Figure 3 is also misleading and doesn't tell the whole story that is being shown in each dataset. e.g. if you drew a*

*linear regression through the Bowdoin Fjord plots at the same salinities (<10) then it would look very different. It's generally inappropriate to draw a regression line beyond where the data points lie and I'd like to see this corrected for relevant fjords. It would be better to use a GAM model to fit the surface data in Figure 3 and the authors should consider doing so (and not plotting beyond the dataset). In addition it would advantageous to indicate which samples on this figure are taken at the surface and which are taken at depth to avoid confusion. "Leverett" should be Søndre Strømfjord.*

**We do not think that use of new modelling approaches is appropriate for a review article on an ancillary topic, and as noted there is very little data for this fjord to force such a model, but we agree it would be a more useful exercise to do for catchments with more extensive data (any of the case studies herein). Extrapolating to zero is a standard oceanographic method for flux calculations. Yes we agree the (Hawkings et al., 2017) data are similar to other fjords (although (Hatton et al., 2019) suggests they aren't). Hence the problem, the high fluxes in the (Hawkings et al., 2017) paper arise from how they are modelled and the assumptions made in this calculation, not because the mid-salinity datapoints are particularly high compared to other datasets (see general point). The plot is replotted, no regression is shown for the 2016 data as it is limited in scope, surface data is re-shaped to distinguish surface from sub-surface data. Some higher salinity Si data for the Kangerlussuaq site is added** (Lund-Hansen et al., 2018)**.**

*L252-253: Worth pointing out this is from a small land terminating glacier. Although there's a lot of debate, larger glaciers seem to export meltwaters with comparatively higher dissolved silica concentrations (Wadham et al., 2010). Pedantic, but I'm also not too keen on the term "surface discharge", as it could indicate any meltwater entering the fjord via surface rivers. Surpaglacial meltwater would be a better term. Most supraglacial meltwater is also routed to the glacier bed (and the subglacial drainage system), so I would think this is unlikely to be a large contributor. By "ice melt" I assume the reference is to iceberg melt?*

**This was already touched on briefly in the original text, but we can expand the sentence slightly (original line 604-606). We had much discussion with respect to how to define meltwater as terms used between the oceanographic and glacial communities differ widely. We use 'supraglacial' when specifically referring to samples which are supraglacial, in a marine context we refer to 'surface' and 'subsurface' to define where freshwater enters the water column. These terms are more vague in a glaciological context, but reflect the reality that not much can be determined about the origin of this water from marine profiles alone.**

*L261-264: Discussion of Hendry et al. (2019) would be useful here. I think the wording misses nuances given flux estimates of Si for ice sheets did not exist before Meire et al. (2016) and Hawkings et al. (2017), and so were considered zero in biogeochemical models and estimates of the global silica cycle. I would consider no estimate an underestimate.*

**L261 'no estimate' can also be interpreted to mean that data was available (e.g. for earlier Si data (Brown et al., 2010; Azetsu-Scott and Syvitski, 1999) and the GEM portal), but it was apparent that it was not an important term and this assumed negligible for the purposes of processes occurring at the model scale (in-fjord processes are sub-grid, and (Hendry et al., 2019) notes the lack of pronounced Si export out of the fjord in question meaning that these processes are definitively sub-grid for global biogeochemical models). When stratification is considered, it is widely recognised that freshwater fluxes are not significant positive influences on macronutrients (McClelland et al., 2011b) especially for N and P. Hence a reason why extensive fluxes cannot be found in the oceanographic literature is because the [freshwater volume × freshwater concentration approach] referred to herein lacks relevance in a broader marine context.**

**The comment concerning models is not strictly correct. Ocean models are forced with observed macronutrient distributions which are available around most of Greenland (excluding the North coastline) and have informed global biogeochemical models for decades, thus any distant effect of meltwater derived material (i.e. beyond sub-model-grid resolution in fjords around the coast) is inherently included in model descriptions of Atlantic macronutrients. It's just not parametrized explicitly, but this is different from being considered zero, a similar comment could be made about many processes that influence nutrient distributions.**

*Table 2: I was not aware that Lawson et al. (2014) measured dissolved organic nitrogen (DON).*

**Lawson reference corrected (wrong reference order)**

*The discharge weighted DON concentrations of Wadham et al. (2016) need to be included here (1.7 µM). No mention has been made of NH4 concentrations. They are minor but should be discussed for completeness.*

**Added, with respect to NH4 we add a comment earlier in the text to clarify NH4 is usually below detection in the marine environment, hence why the case of Kongsfjorden with respect to benthic NH4 release being detectable at the surface is particularly interesting** (Halbach et al., 2019)**. For this reason NH4 fluxes aren't included here (now expanded in the new 'benthic-pelagic coupling' sub-section).**

*I think some discussion of methodology with regard to Fe concentrations would also be appropriate here. As the authors know, it is complicated to simply compare concentrations of Fe where measurements are conducted via different methodologies, for example size fractionation (<0.2 µm, <0.45 µm), and filter type (e.g. PES, PVDF, PC), without noting as such. Polycarbonate (PC) filters (as used in Statham et al., 2008) are particularly problematic as the effective pore size of them reduces sharply upon filtration of even small amounts of sample, especially in highly turbid waters (see Shiller, 2003, for some discussion of this). Further, it is also worth considering representative glacier sample collection. This should be discussed in terms of future research direction. For example, from what I can ascertain, the glaciers samples in Hopwood et al. (2016) that form the Fe concentration estimate in Table 2 are all 1-2 km2, are not ice sheet catchments, and represent insignificant inputs into the fjord. It's questionable how representative a 1-2 km2 glacial catchment is in the context of an ice sheet.*

**Filtration issues are raised earlier as suggested, as this is a very specific issue it is raised alongside brief comments on other methodological issues (e.g. low PO4 detection limits, NH4 contamination, Si freezing problems) in the data compilation (Table 3 in the text). The Hopwood 2016 text shows full surface transects of a fjord in addition to a few freshwater samples. The large uncertainty in estuarine removal factors for glacial dFe (which range 60-99%) adds up to a 40-fold uncertainty on to how large Fe export is when determined from freshwater concentrations. As noted, there is no significant differences in the fluxes calculated, and if the freshwater endmember for this fjord were back-calculated (again, with inevitable large uncertainty), these concentrations would be within the estimated endmember range. As summarised, for Fe/Si if substantial non-conservative behaviour is occurring at low salinities, high accuracy in the freshwater endmember is of limited use because it doesn't provide much insight into the net addition to these elements occurring over the estuarine salinity gradient.**

*L281-286: This is not strictly true, as the authors comment later on L319-323. The flux differences for Fe in particular are due to an arbitrarily applied fjord removal in the papers. The 11-fold flux difference between Stevenson et al. (2017) and Hawkings et al. (2014) is due entirely to the*

*application of an arbitrary fjord removal factor - the fluxat-gate (i.e. the flux from the river into the fjord) are very similar between the studies (note the 90% removal is also discussed in Hawkings et al., 2014, and an estimate of flux after removal given).*

*We have swapped the order of this section to make it clear that there is no meaningful difference between these fluxes.*

***The flux that matters to primary producers in the marine environment is the flux after any removal processes that occur on short timescales (minutes-days) after/during mixing in the ocean. The Hawkings 2014 paper indeed mentions a 90% removal flux as noted in the table, but goes on to state that 'and lastly, a number of Greenlandic glaciers discharge directly into the ocean, avoiding estuarine processing' which is an odd comment as estuarine mixing refers to the process of mixing fresh and saline waters and does not require the physical presence of an estuary (it can, for example, be mimicked in a laboratory mixing fresh and saline waters). Removal factors are not arbitrary as they dictate what Fe is available for marine primary production.***

*L288-293: As several points above. A point is made that seasonal datasets are needed, yet the only publications with datasets >2 months in length have been omitted in the referencing. This needs to be rectified.*

**L288 This is an interesting way of reading this paragraph and not the meaning that was meant. We note throughout need for discharge estimates (meaning physical data) alongside datasets in the marine environment. Given the limited concentrations of macronutrients in freshwater and the strong non-conservative behaviour of those nutrients which are present at high concentrations (Fe/Si), more freshwater data at high resolution doesn't discernibly reduce the uncertainty concerning meltwater effects on primary production. The key point was meant to be that freshwater discharge data is only useful when coupled to marine data for the same region/timescale.**

*L298-302: I understand why the authors want to make this point. In defence of these studies, the flux calculations are made at the "gate" and therefore represent a first order estimate for inputs into the fjord (which is how elemental fluxes from rivers are almost universally calculated). The elemental estimates for ice sheet fluxes (previously assumed to be inconsequential) are also some of the first, so in that context glacial estimates were underestimated (as they weren't estimated at all before). As is touched upon, the largest flux term in the papers cited is the particulate loading (and the particulate fluxes from the ice sheet are massive – estimated at 8% of global sediment fluxes to the ocean; Overeem et al., 2017), which clearly isn't observed in the surface samples and on the timescales the authors discuss.*

**L298 The problem with this 'gate' is that it is not appropriate to use it to speculate about PP in the marine environment, especially for N and P, which is widely recognised when dealing with fluxes from freshwater into the marine environment (McClelland et al., 2011b), hence why such fluxes aren't routinely found in marine literature.**

**It's not clear what the reviewer means by 'clearly isn't observed in surface samples and on the timescales' [we discuss]. We present full depth profiles for all the case studies close to the peak of the meltwater season in studies that were specifically designed to capture the water masses moving in and out of the glacier fjords, therefore any 'flux' that is occurring into the water column on seasonal/annual timescales should be strongly evident.**

*It is a shame the fate of these particulates are not discussed in a balanced manner, and only somewhat negatively, if at all.*

**The effect of particles on scales of 1-100 km from glaciers in the Arctic is overwhelmingly negative on PP (see general comment). It is very hard to reconcile this with the hypothesis the reviewer is referring to [that these particulates have a fertilizing effect]. Considering that in an above comment the reviewer acknowledges that these -potentially fertilizing particulate- fluxes are not strongly evident in any of our catchments at the peak of the meltwater season, it is difficult to find evidence that we are missing something.**

*L303-307 I think this should come earlier – after line 283.*

**Changes as noted above.**

*L319-323 As the authors mention elsewhere, turbidity is important in suppressing surface productivity (via light limitation), which should be mentioned here. Discussion of new work by Seifert et al. (2019) should also be discussed in the context of carbon removal.*

**In the specific sentences here, this (light limitation) is not particularly the case. Whether mid-summer productivity is controlled by only light-limitation or macronutrient-limitation can be assessed by looking at nutrient distribution in near-surface waters. Recent work in this fjord (Holding et al., 2019) suggests primary producers are well adapted to the light conditions in summer and that light-limitation was only a significant proximal-control on primary production at the inner-most fjord station. The Seifert work is now discussed.**

*L417-421. This is the first reference to any benthic processes occurring. This is an oversight of the current manuscript, and this deserves discussion. Studies in both polar regions have investigated benthic recycling and diagenetic processes and the authors should discuss this as well (see Wehrmann et al., 2014, Henkel et al., 2018, Buongiorno et al., 2019).*

**As noted, an extensive discussion of benthic processing is beyond the title of the current manuscript (as per other related themes). We chose our title to be as tightly defined as possible, an extensive review branching out to benthic processes, supra-glacial processes, sea-ice processes, icebergs and dust- would be more comprehensive, but far beyond what we can achieve in a single text. Comments can be easily added however to develop the benthic NH4 story alluded to** (Halbach et al., 2019) **but not flagged as a 'benthic process' in Kongsfjorden and to emphasize the overlapping nature of benthic inputs with meltwater inputs to the ocean (as we already allude to with Fe).**

*449  Can be a few 10s km where turbulent plume is observed and can be spatially variable with time (Tedstone et al., 2012, Hudson et al., 2014).*

**We are aware of this, a line is added (re-)emphasizing that these are very broad generalisations. It is not our intention to provide a 'standardised' conceptual model as we note throughout that glacier-fjords across the Arctic are all practically unique and plumes can vary from not being evident at all in surface waters, to surface plumes extending 10s of kilometres along fjords (but are typically more restricted).**

*L456-458 This is one of several reasons why the limiting nutrients are likely differ. What about riverine inputs, dust inputs…?*

**We have tried to keep the text as focused as possible on the Arctic and an in depth review of differences in Fe sources between the Arctic and Antarctic is detail we do not wish to go into. To a**

first approximation, the critical difference is the vast difference in remoteness, the increased shelf exposure of the Arctic covers the associated shelf/river/atmosphere influences (we have added a sentence to explain this).

*471-471 This is slightly misleading. The coastal regions of Antarctica have low Fe concentrations but there are now several studies highlighting the potential importance of glacial inputs. Further Figure 5 misses out PFe concentrations from Marsay et al. which are consistently >1 nM.*

**We disagree with this comment, even very close to some glaciers there is evidence of residual nitrate and the potential for dFe limitation. If Fe from glaciers is to have an immediate positive effect on marine PP, it has to mix into surface high macronutrient, low dFe waters. Under these circumstances, the dFe supplied will then rapidly be drawn down to low levels (unless macronutrients become depleted). In any case, low dFe concentrations cannot be used to infer a low total Fe supply. It is not misleading to state that dFe-limitation occurs close to Antarctic glaciers, on the contrary, if it didn't then there wouldn't be such a strong biological response to new dFe input in summer.**

**Figure 5 does not 'miss' anything essential for the interpretation with respect to Fe limitation or primary production. There are obviously only so many parameters we can show and these 3 are sufficient to see the general contrast between the two cases. Fe limitation in the ocean can be (and is) assessed in marine waters by looking at the ratio in availability of dissolved Fe to NO3 (Moore et al., 2013), this approach quantitatively assess the extent of Fe stress in cells even working across very broad Fe gradients (e.g. (Browning et al., 2017)) where particulate Fe concentrations vary from high to low alongside DFe gradients. This means either than the direct influence of particulate Fe is via the dissolved phase (i.e. the influence of particulates on Fe bioavailability is accounted for in dissolved Fe measurements) or that, if directly available to some organisms, direct particulate Fe uptake is very minor compared to that of dissolved Fe. Neither of these suppositions is surprising considering that most Fe-cellular uptake pathways are specific to organically complexed Fe or free Fe rendering particulate Fe far less accessible to most species (Shaked and Lis, 2012).**

*Further the authors in this study comment that measurements come following 2 months of intense primary productivity (i.e. these are not traditionally limiting waters, but a productive coastal ecosystem).*

**We referred to these waters as 'high nitrate, low dFe' which is correct. We did not refer to the levels of productivity. It is not clear what the reviewer means here. There is almost invariably a proximal limiting nutrient (except perhaps in extremely productive eastern-boundary upwelling systems) even in very productive waters. Highly productive regions of the Southern Ocean can still be (and often are) Fe-(co)-limited as Fe is still the proximal limiting nutrient and NO3 is not fully depleted during the growth season.**

***L552** This is a rather low estimate of DFe from a grounding line and there is very little information available on concentration estimates. I'm not quite sure how the authors came to this value from Marsay et al. (2017), so it would be useful to provide a sentence to elaborate*

**We can explain this better, the estimate of dFe released beneath an ice shelf is for freshwater ice melt, not for subglacial discharge– as this doesn't exist close to the edge of most ice shelves in the same way as it does for a marine-terminating glacier where subglacial discharge plumes are pronounced at the glacier terminus. It is therefore not derived from Marsay et al., it comes from the freshwater studies cited. We can acknowledge uncertainty in this value (there are no direct**

measurements to quantify it), but also again note that the vast majority of uncertainty in this calculation comes from the estuarine removal of Fe species during mixing between saline and fresh waters. This dwarfs the uncertainty from any other source.

*L584 Completely agree and pertinent point to make given we know almost nothing about ligand binding in glacial fjords (and very little in estuaries more generally). However, I think the perspective here is mainly focused on the idea of bioavailability in "open ocean" waters, which is almost certainly controlled by ligand binding (what this does to the bioavailability I think is still poorly understood given the wide range and complexity of metal stabilising ligands). An increasing number of studies (Kranzler et al. 2011, 2016, Shoenfelt et al. 2017, Grimm et al. 2019) are demonstrating the importance of accessing Fe from particulate pools yet there is very little discussion of this. Surely in coastal areas the particulate pool is likely to be very important given the high concentrations (Schroth et al., 2014) and is almost as poorly understood as the ligand pool? Some balanced discussion of this is important.*

**Direct accessibility of particulate Fe to pelagic phytoplankton is a bit of a misnomer, there are specific examples of mechanisms individual organisms have developed to capture Fe from particulate sources** (Rubin et al., 2011)**, but cellular uptake processes are overwhelmingly dependent upon dissolved Fe availability. This can be demonstrated in the ocean at large (including high particulate Fe coastal regions) by looking at the extent to which Fe stress corresponds to dFe concentrations; dFe availability explains almost perfectly the extent of Fe-limitation across regimes transiting from high to low particulate Fe, meaning that dFe is to a first approximation the principle factor in determining Fe-limitation** (Browning et al., 2017)**. The role of particulates is generally understood to be as a buffer of the dissolved pool. Further, as noted, we have specifically focused the text on the Arctic where Fe is not an extensive limiting factor for primary production and thus its biogeochemistry is of much less interest than were we reviewing a similar topic in the Southern Ocean.**

*L608-609 I don't know what other bedrock types the author's think are likely, but carbonate and silicate bedrock broadly covers them all.*

**More specific comments can be added about the bedrock associated with relatively high inorganic (Alkalinity/silicate) concentrations.**

*L620-643 p1: Linked again to my point about lack of discussion on the importance of benthiccycling,Ithinkthereshouldbesomediscussionofthepotentialroleofalkalinity production in sedimentary environments (e.g. via denitrification and sulfate reduction).*

**R: This is not directly connected to meltwater and is a generic shelf process which we think is well beyond the scope of the title.**

*L620-643 p2: Some contextualisation is needed here. To be my knowledge (and I am definitively not an expert in this) but there tends to be a conservative decline in alkalinity in most estuarine settings (see Cai et al., 2010 and Thomas et al. 2009 for example), so this is not unique. The trend of decreasing alkalinity with increasing freshwater is therefore not particularly surprising in the context of freshwater-saltwater continuum environments as a whole. I agree that monitoring these changes with increasing meltwater discharge will be an important future undertaking.*

*p3: I think some additional detail in this section would be useful for readers. Could the authors also consider an alternative scenario whereby glacial meltwater have low pCO2 and high pH as glacial meltwater tend to be elevated in pH and correspondinglowpCO2(Tranteretal. 1993,*

*SharpandTranter, 2017)? Forexample, there is currently no mention of the conclusions of Meire et al. (2015), which shows glacial melt water associated with low pCO2 regions of the fjord. Also see recent studies by Pilcher et al. (2018) and St Pierre et al. (2019).*

**620 Correct, it is a generally correct statement to state that freshwater generally amplifies ocean acidification, but as noted in the text glacial meltwater has a particularly low TA which means that meltwater is a much more potent acidifier than riverwater (by volume). We have added some sentences here to provide more basic detail on the carbonate cycle as it is easy to get confused. The freshwater pH doesn't really matter in terms of to what extent freshwater drives ocean acidification, i.e. it's not possible for freshwater with low TA and high pH to act as a counter-balance to ocean acidification, freshwater with low TA will always acidify because what matters is the buffering capacity. We would rather not raise confusion here by discussing freshwater pH which varies so much in glaciated catchments precisely because of the low TA. In meltwater-affected saline waters, several non-conservative effects come into play, and the carbonate system is further affected by the extent of primary production which lowers pCO2 and thereby pH, and the general under-saturation of pCO2 in meltwater (which increases pCO2 drawdown). The saturation state of meltwater and estuarine waters with respect to pCO2 is also now discussed briefly as a related issue.**

*L649 L649: What is meant by a DOM concentration? Do you mean DOC concentration*

**DOM refers to dissolved organic material, DOC refers explicitly to dissolved organic C, although these two are often used inter-changeably in the literature given that the majority of DOM is DOC. We will make sure to define these at first use.**

*L689 L689-695: All of the samples in the Holding et al. (2016) bar one are taken from salinities above 34 therefore it's not particularly surprising a clear signature of glacial DOC is observed in bacteria here. Additionally, there is no mention of a glacial DOM in algae, some of which are likely to be mixotrophic as commented in this manuscript (i.e. the interpretation is not straightforward). In this context I really don't think you can consider the Holding et al. estimate of ~11% of bacterial OC in marine waters to be from glacial DOM as minor. It is worth mentioning that other studies (e.g. Fellman et al., 2015, Hagvar et al., 2016) much closer to glacial inputs have found assimilation of glacial DOM into food webs. This is much debated, but one part of the story is that glacial DOM is highly bioavailable (as observed by a number of studies) and is therefore likely consumed very close to the glacier front.*

**Given the title of the text we are primary concerned herein with the effects of meltwater in the marine environment and are therefore much more interested in the broad-scale response of biogeochemistry across saline areas than in freshwater plumes. It seems obvious that in a freshwater system, all of the DOC will be freshwater-associated, the question we are interested in here is whether any freshwater signals can be detected offshore.**

*L695 Paulsen et al. (2018) isn't the correct reference to use in this context and is slightly misleading. This study shows that bioavailability is influenced by glacial meltwater inputs not that it is a minor component of bacterial consumption.*

**The study explicitly demonstrates that glaciers** *'are not a major contributor of carbon or of FDOM in the system'* **that** *'the significant amounts of BDOC in glacial runoff reported by Hood et al. (2009), Fellman et al. (2010), and Lawsonet al. (2014), may, in fact, be negligible compared to the degra-dation potential of the various autochthonous carbon sources that are already present in the fjord'.* **This supports the sentence as cited; we can add an extra few sentences to explain this more.**

*Section 8: This whole section feels extremely speculative to me and is not actually correlatedtorealworldobservations,norwithanyobservationsfromtheArctic. Mostof theliteraturecitedprovidestenuouslinkswiththeonlyevidencethatIcanseebasedon the observation that HABs occur in Patagonia and that there are glaciers in Patagonia as well (but not in the same locations at the HABs). The main study cited (Leon-Munoz et al., 2018) was conducted in fjords with very little or no glacial cover, and contains no reference to glaciers, or meltwater inputs. I'm not against the inclusion of some points from this section into the next section (long-term effects of glacier retreat), as it's important to form hypotheses for testing (especially when anticipating future change), but it needs to be significantly toned down, reduced and explicit mentioned that the hypotheses are speculative.*

**We do already describe this as a 'hypothesis' and only lines 754-755 speculates, the rest of the section is a description of reasonably un-controversial literature. The link between meltwater and stratification is well established, and the link between HABs and stratification is well established. We can of course flag that connecting these two observations with a hypothesis (that changes in glacier-discharge may affect HABs) is not well established. We can also expand the rationale behind this potentially being of relevance to the Arctic as there are HAB-forming species present in stratified areas around west-Greenland. The main study site used in** *Leon-Munoz et al.,* **is not an area where meltwater is the major source of freshwater, but the regional discussion covers areas which do have a majority of local freshwater inputs from glaciers and where changes in glacially derived freshwater inputs are affecting stratification and seasonal patterns of primary production-hence the link to long term changes in glacier fjords. We clarify this more in the revised text.**

*L749-750 I disagree with some of the glaciological interpretation in this paragraph. The study cited (Bliss et al., 2014) is a modelling study to predict future meltwater runoff terms, with no observed data presented (yes future estimates of mass change and runoff are given and are useful, but that is not how this study was cited). This is especially problematic in Patagonia, where there is a relative dearth of data to use for model inputs/validation. There is no evidence to suggest that glacial runoff is in long term decline in this region. The opposite is actually likely to be true with regard to the Patagonian Ice Fields (see recent studies of Forresta et al., 2018, Richter et al., 2019, Li et al., 2019), which are currently the largest contributor to sea level rise per unit area in the world. Glacial meltwater runoff is not intricately linked to precipitation as per non-glacial rivers, but reduced precipitation is likely to amplify mass balance losses.*

**Yes the wording here is incorrectly matched to the reference, we can split this sentence and separate the observations of glacier retreat and reduced runoff, and introduce the concept of peak discharge from future scenarios in model studies.**

*L829-830 I don't disagree but references needed here to substantiate point.*

 **We were referring to the studies already cited in the same sentences, but can repeat them for clarity.**

*Figure 9: Nice looking figure, but I'd really like to see more balance in the interpretation oftheliteraturerepresentedinit. Onemajoromission(againI'mgoingbacktoit)isany lack of benthic feedback. "Sedimentation and Carbon(/nutrient) [burial]" is seen as a one way process here, which is unlikely to be true (see works by Wehrmann amongst many others).*

Figure 9. **Yes this can be changed.**

*L945-947: Recommend updating these figures with new data available in Mouginot et al. (2019).*

**Yes these can be updated.**

References referred to:

[revised manuscript text omitted]

van de Poll, W. H., Kulk, G., Rozema, P. D., Brussaard, C. P. D., Visser, R. J. W. and Buma, A. G. J.: Contrasting glacial meltwater effects on post-bloom phytoplankton on temporal and spatial scales in Kongsfjorden, Spitsbergen, Elem Sci Anth, 6(1), 2018.

Ren, Z., Martyniuk, N., Oleksy, I. A., Swain, A. and Hotaling, S.: Ecological Stoichiometry of the Mountain Cryosphere , Front. Ecol. Evol. , 7, 360 [online] Available from: https://www.frontiersin.org/article/10.3389/fevo.2019.00360, 2019.

Rijkenberg, M. J. A., Slagter, H. A., Rutgers van der Loeff, M., van Ooijen, J. and Gerringa, L. J. A.: Dissolved Fe in the Deep and Upper Arctic Ocean With a Focus on Fe Limitation in the Nansen Basin, Front. Mar. Sci., 5, 88, doi:10.3389/fmars.2018.00088, 2018.

Rubin, M., Berman-Frank, I. and Shaked, Y.: Dust- and mineral-iron utilization by the marine dinitrogen-fixer Trichodesmium, Nat. Geosci., 4(8), 529–534, doi:10.1038/ngeo1181, 2011.

Rysgaard, S., Nielsen, T. and Hansen, B.: Seasonal variation in nutrients, pelagic primary production and grazing in a high-Arctic coastal marine ecosystem, Young Sound, Northeast Greenland, Mar. Ecol. Prog. Ser., 179, 13–25, doi:10.3354/meps179013, 1999.

Schroth, A. W., Crusius, J., Campbell, R. W. and Hoyer, I.: Estuarine removal of glacial iron and implications for iron fluxes to the ocean, Geophys. Res. Lett., 41(11), 3951–3958, doi:10.1002/2014GL060199, 2014.

Seifert, M., Hoppema, M., Burau, C., Elmer, C., Friedrichs, A., Geuer, J. K., John, U., Kanzow, T., Koch, B. P., Konrad, C., van der Jagt, H., Zielinski, O. and Iversen, M. H.: Influence of Glacial Meltwater on Summer Biogeochemical Cycles in Scoresby Sund, East Greenland , Front. Mar. Sci. , 6, 412 [online] Available from: https://www.frontiersin.org/article/10.3389/fmars.2019.00412, 2019.

Shaked, Y. and Lis, H.: Disassembling iron availability to phytoplankton, Front. Microbiol., 3, 123, doi:10.3389/fmicb.2012.00123, 2012.

Sholkovitz, E. R., Boyle, E. A. and Price, N. B.: The removal of dissolved humic acids and iron during estuarine mixing, Earth Planet. Sci. Lett., 40, 130–136, doi:10.1016/0012-821X(78)90082-1, 1978.

Slagter, H. A., Reader, H. E., Rijkenberg, M. J. A., Rutgers van der Loeff, M., de Baar, H. J. W. and Gerringa, L. J. A.: Organic Fe speciation in the Eurasian Basins of the Arctic Ocean and its relation to terrestrial DOM, Mar. Chem., 197, 11–25, doi:https://doi.org/10.1016/j.marchem.2017.10.005, 2017.

Smoła, Z. T., Tatarek, A., Wiktor, J. M., Wiktor, J. M. W., Kubiszyn, A. and Węsławski, J. M.: Primary producers and production in Hornsund and Kongsfjorden – comparison of two fjord systems, Polish Polar Res., 38, 351–373, doi:10.1515/popore-2017-0013, 2017.

Statham, P. J., Skidmore, M. and Tranter, M.: Inputs of glacially derived dissolved and colloidal iron to the coastal ocean and implications for primary productivity, Global Biogeochem. Cycles, 22(3), doi:Gb301310.1029/2007gb003106, 2008.

Tank, S. E., Raymond, P. A., Striegl, R. G., McClelland, J. W., Holmes, R. M., Fiske, G. J. and Peterson, B. J.: A land-to-ocean perspective on the magnitude, source and implication of DIC flux from major Arctic rivers to the Arctic Ocean, Global Biogeochem. Cycles, 26(4), n/a-n/a, doi:10.1029/2011GB004192, 2012.

TG, N. and Hansen, B.: Plankton community structure and carbon cycling on the western coast of Greenland during and after the sedimentation of a diatom bloom , Mar. Ecol. Prog. Ser., 125, 239–257 [online] Available from: https://www.int-res.com/abstracts/meps/v125/p239-257/, 1995.

Tyrrell, T.: The relative influences of nitrogen and phosphorus on oceanic primary production, Nature, doi:10.1038/22941, 1999.

Wadham, J. L., Hawkings, J., Telling, J., Chandler, D., Alcock, J., O'Donnell, E., Kaur, P., Bagshaw, E., Tranter, M., Tedstone, A. and Nienow, P.: Sources, cycling and export of nitrogen on the Greenland Ice Sheet, Biogeosciences, 13(22), 6339–6352, doi:10.5194/bg-13-6339-2016, 2016.

Wadham, J. L., Hawkings, J. R., Tarasov, L., Gregoire, L. J., Spencer, R. G. M., Gutjahr, M., Ridgwell, A. and Kohfeld, K. E.: Ice sheets matter for the global carbon cycle, Nat. Commun., 10(1), 3567, doi:10.1038/s41467-019-11394-4, 2019.

Windom, H., Byrd, J., Smith, R., Hungspreugs, M., Dharmvanij, S., Thumtrakul, W. and Yeats, P.: Trace metal-nutrient relationships in estuaries, Mar. Chem., 32(2), 177–194, doi:10.1016/0304-4203(91)90037-W, 1991.

Zhang, R., John, S. G., Zhang, J., Ren, J., Wu, Y., Zhu, Z., Liu, S., Zhu, X., Marsay, C. M. and Wenger, F.: Transport and reaction of iron and iron stable isotopes in glacial meltwaters on Svalbard near Kongsfjorden: From rivers to estuary to ocean, Earth Planet. Sci. Lett., 424, 201–211, doi:10.1016/j.epsl.2015.05.031, 2015.

---

## Author Comment (AC2) · 22 Oct 2019

This review was comprehensive in its scope of biogeochemical impacts of freshwater discharge in the cryosphere. Using multiple case studies of Arctic fjords, Hopwood et al. capture the range of biogeochemical settings, and in doing so, identify and summarize multiple drivers for diverse phytoplankton response. The review was written with a broad audience in mind, with detailed discussions and patient explanations. The figures aided the discussion and were generally appropriate to the text. I support the publication of this much-needed review pending the appropriate revisions are made.I feel the authors were diligent in their discussion of state-of-knowledge and take a con-

servative stance when estimating fluxes of dissolved nutrients. I support this approach. However, I am cautious about language which aims to describe ecosystem function as similar in both the Arctic and Antarctic. Few studies exist which focus on the ice-ocean interface (within 1km of marine-terminating glaciers) in the Antarctic. The geochemical gradients are intense here and logistically more challenging to study. It is apparent to me that the Antarctic lacks a robust assessment of the fjords, and so the authors should acknowledge that comparatively less is known about the Antarctic.

R: We thank the reviewer for detailed comments on the text. It was not our intention to apply similarities between Arctic and Antarctic systems, or to draw extensive comparisons between the two in general as there are obviously important biogeochemical differences in the marine context between the two. We add some brief discussion of this at appropriate points in the text e.g. lines 544 'These differences may explain why some Antarctic glacier-fjords have significantly higher chlorophyll and biomass than any of the Arctic glacier-fjord systems considered herein (Mascioni et al., 2019). However, we note a general lack of seasonal and interannual data for Antarctic glacier fjord systems precluding a comprehensive inter-comparison of these different systems.'

I think the authors should include in their discussion mention of katabatic wind events and the efficiency at which they mix the upper water column, and the result this would have on export of the surface layer and upwell subsurface sources (see Lundesgaard et al. 2019). Katabatic wind events are important interactions between the atmosphere and ice sheets.

R: We can add further details around this. We expand a brief mention of wind events in 'Fjords as critical zones for glacier-ocean interactions' to read: "energetic shelf forcing (i.e., from coastal/katabatic winds and coastally-trapped waves) can result in rapid exchange over synoptic timescales (Straneo et al., 2010; Jackson et al., 2014; Moffat, 2014) and similarly also affect productivity (Meire et al., 2016b). Katabatic winds are common features of glaciated fjords. Down-fjord wind events facilitate the removal of low salinity surface waters and ice from glacier fjords, and the inflow of warmer, saline

waters at depth (Johnson et al., 2011). The frequency, direction and intensity of wind events throughout the year thus add further complexity to the effect that fjord geometry potentially has on fjord-shelf exchange processes (Cushman-Roisin et al., 1994; Spall et al., 2017)"

Lastly, I am pleased with the discussion about new approaches being required to address these highly dynamic ecosystems. Namely, higher resolution (in space and time) studies are needed to understand how this system function and will respond to climate forcing. Specific comments: L281-282: I do not think we have a well-constrained estimate for the Antarctic. Subglacial discharge is one of the critical fluxes discussed in this review. Recent attention has been given to the subglacial environment and I think it is worth mentioning the uncertainty which surrounds this source. There are biotic and abiotic factors which influence the quality and quantity of iron released to the ocean. Weathering rates are controlled in part by regional geology, but also the microbial communities (namely, chemolithoautotrophs) and exposure to oxygen may be important controls. (Wadham et al. 2010; Tranter, Skidmore, and Wadham 2005) Further, it is nearly impossible to differentiate the effects of tidal uplift, sediment resuspension, glacial calving and subsequent scouring of the sediments at the glacier terminus from purely subglacial discharge. Our understanding of these effects would be greatly increased if measurements were made proximal to cold-based, low velocity marine-terminating glaciers. We can then begin to pick apart the contributions of these different processes.

R: Agreed, we have corrected this statement 'around the Arctic' (rather than 'globally')

L288: Seasonal variation may be an important theme for future directions, both in the Arctic and Antarctic. The authors make this note. Without the aid of the ocean modeling community, we do not yet know how subglacial discharge responds to climate forcing.

R: Agreed, we can add a comment to this effect in the text. (new lines 495): 'Yet determining the extent to which these events affect fjord-scale mixing, biogeochemistry and

how these rates change in response to climate forcing' My general feeling is that while the comparisons may be obvious, there are important functional differences between the Antarctic and Arctic. And so supporting information should be appropriate. For example L203-206 has two well-known studies of particulate iron in the Antarctic (Gerringa et al., 2012, and Annett et al., 2017). The authors may choose to mention this is an important question in general for particle-enriched iron sources. (Fitzsimmons et al. 2017)

R: Yes there are certainly differences, whilst we do not wish to extensively discuss Antarctic systems (precisely because they are so different), we flag the differences elsewhere and instead re-phrase this sentence to refer exclusively to Arctic studies.

L192-195: I think it is important here to discuss the potential for dissolved-particle exchange, facilitated by undersaturated organic ligands or by dissolution in the guts of zooplankton. (Gledhill and Buck 2012; Barbeau et al. 1996)

R: We can add a few lines discussing this in addition to our already present notes to the complexity of the Fe cycle. New lines 255: 'Furthermore, the mechanisms that promote transfer of particulate Fe into bioavailable dissolved phases, such as ligand mediated dissolution (Thuroczy et al., 2012) and biological activity (Schmidt et al., 2011); and the scavenging processes that return dissolved Fe to the particulate phase are both poorly characterized(Tagliabue et al., 2016).'

L452-455: This discussion is accurate, however nutrient stoichiometry (both supply and demand) is what drives primary production and selects for specific phytoplankton taxa, especially in enriched environments. In the instance of diatoms, the N:Fe ratio is a good predictor of iron limitation, where a threshold describes the point at which diatoms begin to grow sub-optimally. The application of geochemical proxies (N:Fe, Siex) for nutrient stress should be applied where such data exists (see King and Barbeau 2007; Hogle et al. 2018).

R: We can add some general comments on nutrient stoichiometry here, it is perhaps

a little detailed to specifically address the subtleties of Fe-diatom limitation, but in general terms the issue can be discussed as an influence on taxonomic groups; 'Although proximal limiting nutrient availability controls total primary production, organic carbon and nutrient stoichiometry has specific effects on the predominance of different phytoplankton and bacterial groups (Egge and Aksnes, 1992; Egge and Heimdal, 1994; Thingstad et al., 2008).'

L511-512: This is indicative of Fe-limitation of the phytoplankton community, which is dominated by diatoms during the sampled summer growth season. Please indicate this is a log-transformed ratio.

R: Explicitly stated.

L496: "islands occur within" The phytoplankton community must meet several requirements for a pronounced increase in growth to occur. They must be physiologically adapted to use glacially derived iron sources. It is unknown the degree to which phytoplankton in the Antarctic use colloidal iron, which would require biotic and abiotic processes to transform it in to a bioavailable form (ie organic complexation, dissolution, photoreduction). I challenge the simplistic view of HNLCs and acknowledge this to be a grand question of our time.

R: Yes, there is no doubt to us that the biological utilization of labile particulates is something under-studied and we can add a few sentences to raise the complexity of biological Fe uptake here in general terms. We note that budgeting exercises show that only a few percent of the 'sedimentary' Fe added downstream of such island plumes has to be solubilized to explain observed primary production. Exactly what this fraction is not particularly clear yet. Also there are other confounding factors such as light limitation, 'However, even in these HNLC waters there are also other concurrent factors that mitigate the effect of glacially derived Fe in nearshore waters where light limitation from near-surface particle plumes may offset the positive effect of Fe-fertilization'

L584-596: This is a great discussion on the uncertainties which remain largely in marine iron biogeochemistry.

L672: ": : :additional subsidies of labile carbon: : :" R: Ammended.

L731: Our data for Antarctica is spares, and biased towards summer growth periods. We have little information about the community dynamics throughout the ice-free growth season. L742: We see the same in Antarctic fjords, but lack an early Spring diatom bloom. Instead, flagellates dominate the fjords. A pronounced diatom bloom and sedimentation event spans 2 weeks, and overall production falls dramatically early-Fall.

R: Agreed, we have noted now the data deficiency when discussing differences between Arctic and Antarctic systems (562): 'However, we note a general lack of seasonal and interannual data for Antarctic glacier fjord systems precluding a comprehensive intercomparison of these different systems.'

L758: "of Patagonia" R: Corrected.

L792: It is becoming more apparent that fjords in the Antarctic are highly productive relative to their Arctic counterparts. Primary production in the fjords rivals that of the Fe-limited shelf regions during the summer. Indeed, we find that organic carbon export is greatest in the inner-fjord environment (unpublished). This is more evidence of the differences in behavior between the Arctic and Antarctic.

R: Yes, as above, we have added some very simple comments aiming to clearly distinguish our case studies from the Antarctic and avoid any possible inference that Arctic and Antarctic glacier fjords can be considered as similar with respect to marine primary producers.

L819-822: How then do we reconcile the expansion of the icesheets and the decreased availability of sediments eroded by wind? R: With difficulty on regional scales, but on global scales this is not implausible. High latitude dust sources may be significant locally, but globally are minor compared to dust from the world's low latitude dessert regions. Changes in 'global' dust signals may therefore be highly sensitive to what

happens at low latitudes and relatively insensitive to what happens around the world's IceSheets/glaciers.

L862-863, 865: Autonomous gliders with optical backscatter and seawater sampling capabilities would we a great way to begin to address this. I agree!
* * *

---

## Author Comment (AC3) · 22 Oct 2019

Yes thanks, we are aware of this new article and have updated the revised text with several relevant 2019 papers published in the past few months.

With respect to the region discussed by Hendry et al., we have already presented full depth Si profiles for this fjord and associated literature which covers the Si cycle and diatom activity in this fjord extensively.

The offshore work reported by Hendry et al., is more directly linked to benthic processes occurring in shelf sediments than it is to direct meltwater discharge into the

marine environment. We specifically focused this review on meltwater discharge because a review extending to icebergs/glacial particles/shelf sediments & benthic reprocessing etc would be incredibly extensive.

However, whilst we did mention very briefly the role of benthic processes in a few contexts and pick up briefly on benthic-pelagic coupling in Kongsfjorden (without labelling this as such), we neglected to specifically link glaciers, to shelf sediments and benthic processes, back to water column biogeochemistry. Such processes are well investigated in benthic fjord scale studies in at least some parts of Greenland and Svalbard (e.g. in Kongsfjorden and Young Sound). We will therefore add a few sentences to outline the basic concept of benthic processes and shelf sediments both in the introduction, in the corresponding field site descriptions and in section 11.

---

## Author Response (AR1)

**Replies to reviewers comments.**

**Two reviewers are thanked for detailed comments on the text. Please find below replies to reviewers, alongside an annotated, revised text (any new references discussed, but not referred to in the manuscript, are added at the end).**

This review was comprehensive in its scope of biogeochemical impacts of freshwater discharge in the cryosphere. Using multiple case studies of Arctic fjords, Hopwood et al. capture the range of biogeochemical settings, and in doing so, identify and summarize multiple drivers for diverse phytoplankton response. The review was written with a broad audience in mind, with detailed discussions and patient explanations. The figures aided the discussion and were generally appropriate to the text. I support the publication of this much-needed review pending the appropriate revisions are made. I feel the authors were diligent in their discussion of state-of-knowledge and take a conservative stance when estimating fluxes of dissolved nutrients. I support this approach. However, I am cautious about language which aims to describe ecosystem function as similar in both the Arctic and Antarctic. Few studies exist which focus on the ice-ocean interface (within 1km of marine-terminating glaciers) in the Antarctic. The geochemical gradients are intense here and logistically more challenging to study. It is apparent to me that the Antarctic lacks a robust assessment of the fjords, and so the authors should acknowledge that comparatively less is known about the Antarctic.

R: We thank the reviewer for detailed comments on the text. It was not our intention to apply similarities between Arctic and Antarctic systems, or to draw extensive comparisons between the two in general as there are obviously important biogeochemical differences in the marine context between the two. We add some brief discussion of this at appropriate points in the text. E.g. new lines 729-733.

I think the authors should include in their discussion mention of katabatic wind events and the efficiency at which they mix the upper water column, and the result this would have on export of the surface layer and upwell subsurface sources (see Lundesgaard et al. 2019). Katabatic wind events are important interactions between the atmosphere and ice sheets.

**R: We have added further details around this. See new sections: 127-138 and 619-629**

Lastly, I am pleased with the discussion about new approaches being required to address these highly dynamic ecosystems. Namely, higher resolution (in space and time) studies are needed to understand how this system function and will respond to climate forcing. Specific comments:

L281-282: I do not think we have a well-constrained estimate for the Antarctic. Subglacial discharge is one of the critical fluxes discussed in this review. Recent attention has been given to the subglacial environment and I think it is worth mentioning the uncertainty which surrounds this source. There are biotic and abiotic factors which influence the quality and quantity of iron released to the ocean. Weathering rates are controlled in part by regional geology, but also the microbial communities (namely, chemolithoautotrophs) and exposure to oxygen may be important controls. (Wadham et al. 2010; Tranter, Skidmore, and Wadham 2005) Further, it is nearly impossible to differentiate the effects of tidal uplift, sediment resuspension, glacial calving and subsequent scouring of the sediments at the glacier terminus from purely subglacial discharge. Our understanding of these effects would

be greatly increased if measurements were made proximal to cold-based, low velocity marineterminating glaciers. We can then begin to pick apart the contributions of these different processes. R: Agreed, we have corrected this statement 'around the Arctic' (rather than 'globally') and we have noted to difficulty in distinguishing different Fe sources close to glaciers, especially in Southern Ocean where observations are sparse (new lines 641-650)

L288: Seasonal variation may be an important theme for future directions, both in the Arctic and Antarctic. The authors make this note. Without the aid of the ocean modeling community, we do not yet know how subglacial discharge responds to climate forcing.

**R: Agreed, we can add a comment to this effect in the text. (new lines 611):** *'Yet determining the extent to which these events affect fjord-scale mixing, biogeochemistry and how these rates change in response to climate forcing...'*

My general feeling is that while the comparisons may be obvious, there are important functional differences between the Antarctic and Arctic. And so supporting information should be appropriate. For example L203-206 has two well-known studies of particulate iron in the Antarctic (Gerringa et al., 2012, and Annett et al., 2017). The authors may choose to mention this is an important question in general for particle-enriched iron sources. (Fitzsimmons et al. 2017)

R: Yes there are certainly differences, whilst we do not wish to extensively discuss Antarctic systems (precisely because they are so different), we flag the differences elsewhere and instead re-phrase this sentence to refer exclusively to Arctic studies.

L192-195: I think it is important here to discuss the potential for dissolved-particle exchange, facilitated by undersaturated organic ligands or by dissolution in the guts of zooplankton. (Gledhill and Buck 2012; Barbeau et al. 1996)

**R: We can add a few lines discussing this in addition to our already present notes to the complexity of the Fe cycle. New lines 302-306**

L452-455: This discussion is accurate, however nutrient stoichiometry (both supply and demand) is what drives primary production and selects for specific phytoplankton taxa, especially in enriched environments. In the instance of diatoms, the N:Fe ratio is a good predictor of iron limitation, where a threshold describes the point at which diatoms begin to grow sub-optimally. The application of geochemical proxies (N:Fe, Siex) for nutrient stress should be applied where such data exists (see King and Barbeau 2007; Hogle et al. 2018).

R: We can add some general comments on nutrient stoichiometry here, it is perhaps a little detailed to specifically address the subtleties of Fe-diatom limitation, but in general terms the issue can be discussed as an influence on taxonomic groups (new lines 695-698)

L511-512: This is indicative of Fe-limitation of the phytoplankton community, which is dominated by diatoms during the sampled summer growth season. Please indicate this is a log-transformed ratio.

**R: Explicitly stated.**

L496: "islands occur within" The phytoplankton community must meet several requirements for a pronounced increase in growth to occur. They must be physiologically adapted to use glacially derived iron sources. It is unknown the degree to which phytoplankton in the Antarctic use colloidal iron, which would require biotic and abiotic processes to transform it in to a bioavailable form (ie organic complexation, dissolution, photoreduction). I challenge the simplistic view of HNLCs and acknowledge this to be a grand question of our time.

R: Yes, there is no doubt to us that the biological utilization of labile particulates is something under-studied and we can add a few sentences to raise the complexity of biological Fe uptake here in general terms. We note that budgeting exercises show that only a few percent of the 'sedimentary' Fe added downstream of such island plumes has to be solubilized to explain **observed primary production. Exactly what this fraction is not particularly clear yet. Also there are other confounding factors such as light limitation,** 'However, even in these HNLC waters there are also other concurrent factors that mitigate the effect of glacially derived Fe in nearshore waters where light limitation from near-surface particle plumes may offset the positive effect of Fe-fertilization'

L584-596: This is a great discussion on the uncertainties which remain largely in marine iron biogeochemistry.

L672: ": : :additional subsidies of labile carbon: : :"

**R: Ammended.**

L731: Our data for Antarctica is spares, and biased towards summer growth periods. We have little information about the community dynamics throughout the ice-free growth season. L742: We see the same in Antarctic fjords, but lack an early Spring diatom bloom. Instead, flagellates dominate the fjords. A pronounced diatom bloom and sedimentation event spans 2 weeks, and overall production falls dramatically early-Fall.

**R: Agreed, we have noted now the data deficiency when discussing differences between Arctic and Antarctic systems (729-723)**: 'However, we note a general lack of seasonal and interannual data for Antarctic glacier fjord systems precluding a comprehensive intercomparison of these different systems.'

L758: "of Patagonia"

**R: Corrected.**

L792: It is becoming more apparent that fjords in the Antarctic are highly productive relative to their Arctic counterparts. Primary production in the fjords rivals that of the Fe-limited shelf regions during the summer. Indeed, we find that organic carbon export is greatest in the inner-fjord environment (unpublished). This is more evidence of the differences in behavior between the Arctic and Antarctic.

R: Yes, as above, we have added some very simple comments aiming to clearly distinguish our case studies from the Antarctic and avoid any possible inference that Arctic and Antarctic glacier fjords can be considered as similar with respect to marine primary producers.

*L819-822:* How then do we reconcile the expansion of the icesheets and the decreased availability of sediments eroded by wind?

R: With difficulty on regional scales, but on global scales this is not implausible. High latitude dust sources may be significant locally, but globally are minor compared to dust from the world's low latitude dessert regions. Changes in 'global' dust signals may therefore be highly sensitive to what happens at low latitudes and relatively insensitive to what happens around the world's locSheets/glaciers.

L862-863, 865: Autonomous gliders with optical backscatter and seawater sampling capabilities would we a great way to begin to address this. I agree!

[end of review]

When and where some of the literature quoted is relatively selective and the way it is contextualised in certain circumstances misses nuance. One major omission is a discussion of particulate fluxes (both as part of nutrient budgets, and importance in ballasting and C burial) and indirect processing of glacial inputs (related to particulate inputs; i.e. benthic recycling and/or burial). Given the context of these environments (dominated by inputs of products of physical weathering), and the existence of literature in other glacially influenced regions (e.g. Laura Wehrmann's and associated groups ongoing work in Svalbard; e.g. Wehrmann et al., 2014), this could have been an opportunity to start a balanced discussion. This is an oversight, especially for a review article, and given recent interest in particulate fluxes (not just in glacial locations), even if the authors do not think these flux terms are important.

Sedimentary processes remain a little beyond the scope of the present title, but nevertheless are directly linked in some aspects so we introduce new short sections to better link to associated topics (5.1 and 5.2) which includes research which was not published when we submitted our original text.

We have now introduced the text with a section on primary production which explains why a link between lithogenic 'particulate nutrients' and primary producers [suggested by the reviewer] is not clear to us (new section 3.0). Multiple lines of evidence show that primary producers and primary production is negatively affected by Arctic glacier discharge on a scale comparable to that over which sedimentation occurs- with the notable exception of marine-terminating glaciers that are discussed later in the text- and thus we focus primarily on phenomena that can explain this pattern in primary production.

Lines 401-435 explain why we explicitly focus on dissolved macronutrients and why it is difficult to reconcile field observations of primary production with a few of the reviewers' comments.

As a previous reviewer indicated, there is also a need to discuss and incorporate more recent publications (i.e. Hendry et al., 2019, but also Seifert et al., 2019, Wadham et al., 2019 amongst some others I suggest below), and some key papers have been omitted or not referenced where they should have been. I have some major reservations about section 8, which feels incredibly speculative, and think it should be toned down and incorporated into section 9 in a much reduced form. Specific comments to be addressed are below.

More recent publications which were not available when we wrote or submitted the original text are now incorporated.

The wording in section 8 (now 9) is altered to make it clear where uncertainties remain. We emphasize that a link between stratification and HAB events is well established, as is a link between changing freshwater runoff and HAB events. This is not unique to glacier systems, but is not speculative. We think it is interesting to raise this as a topic for future research as changing glacier discharge does affect many (if not, all) of the drivers responsible for HAB events. The section is clearly highlighted as a question and we now clearly flag that this subject is not well explored in the Arctic, unlike many of the other topics highlighted.

L65: Calcium carbonate is not an ion. This should be corrected to "inorganic salts. . .".

**Rephrased 'inorganic components'**

*L64-68: These plumes also carry large quantities of reactive particular material, including labile particulate nutrients. Whatever you think of their ultimate fate (which can be discussed) I think this is important to note as it is an important characteristic of glacial meltwaters. In this context I'm sure*

the authors will be aware of the literature (some suggestions for inclusion are Hendry et al., 2019, Seifert et al., 2019, Jeandel and Oelkers, 2015, Grimm et al., 2019, Schoenfelt et al 2017, Morgan et al., 2014, Eiriksdottir et al., 2015).

**Lines 248-271 now more extensively discuss sediment loads in glacial discharge.**

Whilst these 'reactive' fluxes may be of intense interest to glaciologists/geochemists, the key question herein is how are they linked to biology? All of our available field evidence suggests that the main link on this spatial/temporal scale in the Arctic is suppression of marine primary production (see section 3)– presumably via light limitation. Thus it is not clear to us what the relevance of lithogenic elemental fluxes is to primary producers on inter-annual timescales. We now introduce the text with this primary production data to better explain this rationale (new section 3) and in lines 402-435 try to reconcile the reviewers' arguments with the field studies highlighted herein.

*Figure 1: I'd like to see the quality of this figure improved before publication. As the first figure and a key map of study areas it's also a little too basic at present.*

**A revised figure 1 is added, (this was previously a last minute add-on at the request of the editor).**

Section 3: I find the referencing in the first paragraph curious. Although by no means do I think that the authors should be referencing some work ahead of others, the first reference of a particular group's work is page 8, where it's critiqued, despite the number of publications from this group that are suitable for referencing before (in this context).

Section 3. The reviewer was, we assume, referring to extensive datasets on freshwater water composition. In section 11 (see especially Figure 11) we discuss the relative importance of different processes and emphasize that freshwater composition is not a major consideration in terms of the large-scale impact of meltwater from the Arctic in the ocean.

There remains an inherent bias in the text towards studies conducted at the 5 field sites discussed in detail. These were selected at workshops based on the availability of marine data for any glaciated regions across the Arctic as this is essential to cover the manuscript topic. We don't therefore think that the literature selection is illogical. There are several study areas which could have been selected as better alternatives if the text had been focused entirely on terrestrial or freshwater systems, but we explicitly focus herein on the marine environment and thus it is difficult to extensively discuss literature from sites where there is a lack of extensive (or any) marine data. In terms of published literature, the only other potential candidates we identified for focus regions were Ryder Glacier and the '79 North' glacier which are both subject to extensive ongoing research programs, but (at the time of writing) were not extensively discussed in terms of the topics covered herein, or other small catchments which are very close to Godthabsfjord or Kongsfjorden.

The authors discuss the need for seasonal datasets to contextual flux information, yet there are already several studies currently available that contain temporal datasets over several months and several years of monitoring for hydro chemical parameters, macronutrients and Fe. The concentrations used on Table 2 are from some of these studies, and are discharge weighted mean concentrations derived from a seasonal dataset (the only DWM concentrations in Greenlandic meltwaters that I know to exist at present). There is certainly a debate that can be had with regard to the particulate nutrient inputs (which the authors should deal with in a more balanced manner), but I do not fully understand why other aspects of those papers have been overlooked. These might not be datasets that span whole melt seasons (typically early May to early September), but they are the longest available at the moment and should be acknowledged as such. I would like to see the current literature discussed in a more nuanced way in the next version of the manuscript.

We add a new introductory section on primary production in Arctic glaciated regions (section 3) which better explains the rationale for the topics within the text. We think the reviewer here is referring to freshwater nutrient time series. It is not an argument we agree with that time series of freshwater data (without corresponding marine data) can be used to make conclusions about the marine environment (see Figure 10) because this is such a small budget term that even defining it with very high resolution doesn't really enhance our understanding of changes in marine primary production.

It's impossible to make even basic comments on fjord-scale processes without some 2D temperature/salinity data and thus we cannot really draw any conclusions about the marine environment from freshwater(only) timeseries-even if the resolution is particularly good.

L163: Semantics but I think this should be "dissolved macronutrients". Again, the role of particulate macronutrients can be critiqued, but this is an important distinction to make. Glacial meltwaters have high concentrations of particulate nutrients (save N), and low concentrations of dissolved nutrients, and it's important to highlight that whatever you think of the eventual fate.

We add lines 402-435 to better explain why a clear distinction is made between dissolved macronutrient fluxes and lithogenic elemental fluxes. 'Labile' elements (rather than organic C/N/P particulates) in particles are not generally considered as nutrients because they are not widely uptaken by cellular processes. With respect to PO4, NO3 and Si, measurements of these compounds are usually conducted unfiltered in the marine environment so it is more common to refer to 'macronutrients' than 'dissolved macronutrients'. Labile particulates cannot be referred to as 'nutrients' unless it can be demonstrated that they actively are taken up into biological systems.

L164-166: There is a push here to emphasise that the PO4 concentrations in glacial meltwaters are particularly low. I'm not arguing against this (they are compared to some marine waters), but the PO4 concentrations in glacial meltwaters from large catchments (see Leverett Glacier) are similar to the global river mean (0.32  $\mu$ M; Meybeck, 1982), and also similar to (or exceeding) PO4 concentrations in Arctic rivers (0.03-0.76  $\mu$ M). Further, the annual yields (normalised to catchment area) are very high (see Table 4 in Hawkings et al., 2016). Again, not all this information may be needed in the context of the review, but it's important to not single out glacial inputs as being particularly nutrient deplete as is currently done

This section is re-written to include new PO4 data from Arctic rivers for comparison (new lines 238-248). Considering the organic and inorganic P phases in rivers, PO4 in meltwater is low. In absolute terms, both meltwater and rivers contain limited quantities of PO4.

L167: This needs a reference and some contextual information. See point above.

We have added more data to support a general overview of rivers/meltwater as sources of different macronutrients including an expanded figure 3 (originally Figure 2).

L178: I'm not sure if I'd call these measurements "extensive" given they are from two small glaciers in a fjord with many meltwater inputs (the major inputs coming from much larger tidewater glaciers). The references given for studies of Svalbard meltwaters also have listed LoD for PO4 is 5 ppb (0.16

 $\mu$ *M*), and a limit of quantification likely even higher (although not mentioned) making those figures difficult to compare to the fjord measurements when the LoD is typically better.

Re-phrased (the references were given as examples, there are a very large number of references giving freshwater nutrient concentrations for Kongsfjorden). Yes the LOD of PO4 is often problematic in these studies and we suspect if field blanks were properly/consistently reported through the literature the calculated PO4 concentrations in glacial freshwater would change. Whilst we do not particularly want to produce a methodological review herein, for Fe, PO4 and Si there are potential well-known problems to raise and so a brief comment on filtration/method artefacts for those compounds/elements where this may be an issue for data quality (Fe/PO4/Si) is now added alongside the data compilation (Tables 2/3 in the text).

L188-189: As above point, I don't really understand the referencing here. There are other appropriate studies that emphasise the existence of reactive particulate Fe that should be referenced here (Bhatia et al., 2013, Schroth et al., 2012, Schroth et al., 2014, Hawkings et al., 2014, Hawkings et al., 2018).

There is, we thought obviously, a strong bias throughout the text to studies conducted in the marine environment at the key fieldsites mentioned. This is explicit because we want to discuss the effect of meltwater in the marine environment, and it is very difficult to contextualise studied that don't have extensive (or any) marine data. It is also very difficult to contextualise studies that don't have accompanying data concerning salinity and other key parameters available (especially for nutrients like Fe and Si that experience significant modification within estuarine zones). The Hawkings and Bhatia works are freshwater based. The Schroth work is more useful in this context and is extensively discussed extensively concerning estuarine mixing (although the accompanying data is not available online or from the author so we cannot comment in as much depth).

To quantify why it is better to use marine/estuarine studies to study Fe/Si in the ocean, consider the following. Estuarine removal flocculates between 60 and 99% of dissolved Fe, which is highly variable between (and even within) different estuarine gradients (Schroth et al., 2014; Sholkovitz et al., 1978; Zhang et al., 2015). Thus the same dissolved Fe concentration measured at zero salinity could plausibly produce values varying by a factor of 40 in saline waters, which is generally much less than seasonal changes in Fe concentrations of any fraction (Hawkings et al., 2014; Statham et al., 2008). Similarly, total Fe shows no consistent straightforward relationship to salinity or to dissolved Fe. Hence it is very difficult to make conclusions about the fate of Fe from freshwater data alone, and improving accuracy in the freshwater endmember doesn't really improve this much, whereas marine studies unambiguously show the actual enrichment irrespective of what the freshwater endmember was.

L198-199: Low nM concentrations are still fairly significant in a marine context, especially when 100 km from the main inputs at the head of the fjord. Surface open ocean waters and even some coastal systems are typically <0.5 nM and often much lower (Johnson, Gordon, & Coale, 1997; Tagliabue et al., 2017). These concentrations would usually be considered very high for marine systems - an important point worth making I think

Costal Fe values are always high relative to offshore waters, this is not unique to near-glacier systems, and whilst these values are 100 km from the nearest glacier, they are only 1 km from the coastline and much less than this (50-200 m) from the sea-floor making it an interesting assumption that they definitely have a direct meltwater origin. Fe concentrations across a salinity gradient should always be discussed with salinity in mind. Normalised to salinity, dissolved Fe concentrations at these locations are not particularly high. Considering that Arctic concentrations

(offshore) peak within the transpolar drift at 4-5 nM dFe (in saline waters) (Rijkenberg et al., 2018; Slagter et al., 2017), glacier estuary concentrations of 1-3 nM dFe are-perhaps surprisingly- low. Total Fe concentrations are higher, but are more challenging to interpret given that they don't behave conservatively and are less relevant to determining Fe availability to primary producers. A full discussion of the Fe-cycle is beyond the scope of this text given the limited relevance to Arctic primary production, it is of course much more relevant in the context of Fe-limitation immediately adjacent to glaciers in the Southern Ocean, but we have deliberately kept an Arctic focus to avoid getting side-tracked.

**L205-206: Schroth et al. (2014) should be referenced in this context as well.**

Yes the study is relevant, although we have tried to limit general points to 2-3 references (we are already over the suggested limit) and in this context the Crusius data cited covers the same region more extensively.

**L211-212: What about biological uptake?**

**This of course results in some drawdown in almost all environments, hence why we started the sentence 'In the absence of biological processes'**

L222-230: The first assertation in this paragraph (that Si is generally released from the particulate phase over a salinity gradient) is based on one referenced paper (Windom et al., 1991). Other review articles on estuarine environments (e.g. review article of Statham, 2012) and many other estuarine papers (e.g. Edmond et al., 1985, Burton et al., 1970, Cloern et al., 2017, Bell, 1994, Raguenau et al., 2002 to list a few) note that conservative behaviour, or in some circumstances reverse weathering and/or adsorp-tion/other removal processes have been observed, especially in similar high sediment, deltaic environments (Treguer et al., 2013, Kamatani et al., 1984), apart from when strong benthic Si fluxes have been inferred (e.g. Eyre and Balls, 1999). I'm not saying dissolution of particulate material is not important in other systems, but the authors need more than one reference to support this generalisation.

**As per earlier comments on riverine PO4, we expand this section and include data from the 3 best studied Arctic river estuaries for comparison to show the range of behaviours observed in different macronutrients with a brief over-view of the underlying reasons for variation between Arctic estuaries (new lines 238-248).**

L230-234: I welcome balanced debate, however, it's disappointing the review makes no mention of incubation experiments performed in this study, which show release of DSi from particulate material to seawater over a period of 30 days in samples that weren't treated to remove ASi. This doesn't necessarily mean DSi is released in the fjord surface, but it's worth consideration especially given the recent findings of Hendry et al. (2019) and Gruber et al. (2019) among others. The former shows strong evidence bottom water modification for example. The benthic environment is currently ignored and the lack of discussion of this is an oversight.

We are not disputing that some particulate Si is released from particles, this is beyond doubt and evident from the shape of the Si/salinity curves. The incubation experiments in question are extrapolated over several times the time period over which these particles would remain in suspension in a glacier fjord. It is very difficult to reconcile with large datasets elsewhere (Fig. 4).

A sub-section (5.1) has been added to expand on the direct benthic-pelagic linkages highlighted by (Halbach et al., 2019) where we now also mention briefly more general benthic processes affected by glaciers (i.e. high sedimentation), yet benthic cycling is not unique to environments affected by

meltwater so it would be beyond the scope of the review to extensively cover benthic pelagic processes (as per dust, icebergs and sea-ice), especially looking at shelf environments over long timescales (geological rather than seasonal/interannual). We explicitly titled and focused the review on 'meltwater' to keep a tight focus.

L242-254 (and Figure 3): I don't disagree with most of the interpretation here, but given that some of the low salinity end members are not dissimilar to Hawkings et al. (2017) (where there are no high salinity end members) it seems curious that the authors explain this by lack of data and complexity of fjord systems in these instances. Simply drawing linear regressions through points in Figure 3 is also misleading and doesn't tell the whole story that is being shown in each dataset. e.g. if you drew a linear regression through the Bowdoin Fjord plots at the same salinities (<10) then it would look very different. It's generally inappropriate to draw a regression line beyond where the data points lie and I'd like to see this corrected for relevant fjords. It would be better to use a GAM model to fit the surface data in Figure 3 and the authors should consider doing so (and not plotting beyond the dataset). In addition it would advantageous to indicate which samples on this figure are taken at the surface and which are taken at depth to avoid confusion. "Leverett" should be Søndre Strømfjord.

We do not think that use of new modelling approaches is appropriate for a review article on an ancillary topic, and as noted there is very little data for this fjord to force such a model (in writing the text, we reviewed this again, and there certainly isn't enough data even to define the fresh/saline endmembers for this fjord, so it would be meaningless to construct even a 2D model for the fjord), but we agree it would be a more useful exercise to do for catchments with more extensive data (any of the case studies herein). Figure 3 (now 4) is re-drawn as suggested. Yes we agree the (Hawkings et al., 2017) data are similar to other fjords (although (Hatton et al., 2019) suggests they aren't). Hence the problem, the high fluxes in the (Hawkings et al., 2017) paper arise from how they are modelled and the assumptions made in this calculation, not because the mid-salinity datapoints are high compared to other datasets.

L252-253: Worth pointing out this is from a small land terminating glacier. Although there's a lot of debate, larger glaciers seem to export meltwaters with comparatively higher dissolved silica concentrations (Wadham et al., 2010). Pedantic, but I'm also not too keen on the term "surface discharge", as it could indicate any meltwater entering the fjord via surface rivers. Surpaglacial meltwater would be a better term. Most supraglacial meltwater is also routed to the glacier bed (and the subglacial drainage system), so I would think this is unlikely to be a large contributor. By "ice melt" I assume the reference is to iceberg melt?

We had much discussion with respect to how to define meltwater as terms used between the oceanographic and glacial communities differ widely. We use 'supraglacial' when specifically referring to samples which are supraglacial, in a marine context we refer to 'surface' and 'subsurface' to define where freshwater enters the water column. These terms are more vague in a glaciological context, but reflect the reality that not much can be determined about the origin of this water from marine profiles alone. We have added a paragraph to explain the rationale (new lines 376-389).

L261-264: Discussion of Hendry et al. (2019) would be useful here. I think the wording misses nuances given flux estimates of Si for ice sheets did not exist before Meire et al. (2016) and Hawkings et al. (2017), and so were considered zero in biogeochemical models and estimates of the global silica cycle. I would consider no estimate an underestimate.

(Hendry et al., 2019) notes the lack of pronounced Si export out of the fjord in question meaning that these processes are definitively sub-grid for global biogeochemical models which is consistent

with them not being explicitly parametrized. The comment concerning models is not strictly correct. Ocean models are forced with observed macronutrient distributions which are available around most of Greenland (excluding the North coastline) and have informed global biogeochemical models for decades, thus any distant effect of meltwater derived material (i.e. beyond sub-model-grid resolution in fjords around the coast) is inherently included in model descriptions of Atlantic macronutrients. It's just not parametrized explicitly, but this is different from being considered zero, a similar comment could be made about many processes that influence nutrient distributions.

Table 2: I was not aware that Lawson et al. (2014) measured dissolved organic nitrogen (DON).

**Lawson reference corrected (wrong reference order)**

The discharge weighted DON concentrations of Wadham et al. (2016) need to be included here (1.7  $\mu$ M). No mention has been made of NH4 concentrations. They are minor but should be discussed for completeness.

Added, with respect to NH4 we add a comment earlier in the text to clarify NH4 is usually present at very low concentrations in marine waters, hence why the case of Kongsfjorden with respect to benthic NH4 release being detectable at the surface is particularly interesting (Halbach et al., 2019). For this reason NH4 fluxes aren't included here (now expanded in the new 'benthic-pelagic coupling' sub-section 5.1).

I think some discussion of methodology with regard to Fe concentrations would also be appropriate here. As the authors know, it is complicated to simply compare concentrations of Fe where measurements are conducted via different methodologies, for example size fractionation (<0.2  $\mu$ m, <0.45  $\mu$ m), and filter type (e.g. PES, PVDF, PC), without noting as such. Polycarbonate (PC) filters (as used in Statham et al., 2008) are particularly problematic as the effective pore size of them reduces sharply upon filtration of even small amounts of sample, especially in highly turbid waters (see Shiller, 2003, for some discussion of this). Further, it is also worth considering representative glacier sample collection. This should be discussed in terms of future research direction. For example, from what I can ascertain, the glaciers samples in Hopwood et al. (2016) that form the Fe concentration estimate in Table 2 are all 1-2 km2, are not ice sheet catchments, and represent insignificant inputs into the fjord. It's questionable how representative a 1-2 km2 glacial catchment is in the context of an ice sheet.

Filtration issues are raised earlier as suggested, as this is a very specific issue it is raised alongside brief comments on other methodological issues (e.g. low PO4 detection limits, NH4 contamination, Si freezing problems) in the data compilation (Table 3 in the text). The Hopwood 2016 text shows full surface transects of a fjord in addition to a few freshwater samples. The large uncertainty in estuarine removal factors for glacial dFe (which range 60-99%) adds up to a 40-fold uncertainty on to how large Fe export is when determined from freshwater concentrations. As noted, there is no significant differences in the fluxes calculated, and if the freshwater endmember for this fjord were back-calculated (again, with inevitable large uncertainty), these concentrations would be within the estimated endmember range. As summarised, for Fe/Si if substantial non-conservative behaviour is occurring at low salinities, high accuracy in the freshwater endmember is of limited use because it doesn't provide much insight into the net addition to these elements occurring over the estuarine salinity gradient.

L281-286: This is not strictly true, as the authors comment later on L319-323. The flux differences for Fe in particular are due to an arbitrarily applied fjord removal in the papers. The 11-fold flux

difference between Stevenson et al. (2017) and Hawkings et al. (2014) is due entirely to the application of an arbitrary fjord removal factor - the fluxat-gate (i.e. the flux from the river into the fjord) are very similar between the studies (note the 90% removal is also discussed in Hawkings et al., 2014, and an estimate of flux after removal given).

We have swapped the order of this section to make it clear that there is no meaningful difference between these fluxes and that apparent differences simply reflect a different flux gate. The flux that matters to primary producers in the marine environment is the flux after any removal processes that occur on short timescales (minutes-days) after/during mixing in the ocean. Removal factors are not arbitrary as they dictate what Fe is available for marine primary production.

L288-293: As several points above. A point is made that seasonal datasets are needed, yet the only publications with datasets >2 months in length have been omitted in the referencing. This needs to be rectified.

L288 This is an interesting way of reading this paragraph and not the meaning that was meant. We note throughout need for discharge estimates (meaning physical data) alongside datasets in the marine environment as we think is clearer from (new lines) 475-491. Given the limited concentrations of macronutrients in freshwater and the strong non-conservative behaviour of those nutrients which are present at high concentrations (Fe/Si), more freshwater data at high resolution doesn't discernibly reduce the uncertainty concerning meltwater effects on primary production. The key point was meant to be that in this context [marine PP] freshwater discharge data is only useful when coupled to marine data for the same region/timescale.

L298-302: I understand why the authors want to make this point. In defence of these studies, the flux calculations are made at the "gate" and therefore represent a first order estimate for inputs into the fjord (which is how elemental fluxes from rivers are almost universally calculated). The elemental estimates for ice sheet fluxes (previously assumed to be inconsequential) are also some of the first, so in that context glacial estimates were underestimated (as they weren't estimated at all before). As is touched upon, the largest flux term in the papers cited is the particulate loading (and the particulate fluxes from the ice sheet are massive – estimated at 8% of global sediment fluxes to the ocean; Overeem et al., 2017), which clearly isn't observed in the surface samples and on the timescales the authors discuss.

L298 The problem with this 'gate' is that it is not appropriate to use it to speculate about changing primary production in the marine environment, especially for N and P, which is widely recognised when dealing with fluxes from freshwater into the marine environment (McClelland et al., 2011b).

It's not clear what the reviewer means by 'clearly isn't observed in surface samples and on the timescales' [we discuss]. We present full depth profiles for all the case studies close to the peak of the meltwater season in studies that were specifically designed to capture the water masses moving in and out of the glacier fjords, therefore any 'flux' that is occurring into the water column on seasonal/annual timescales should be strongly evident.

In a geochemical context sediment loads (lines 248-271) and the associated lithogenic elements fluxes (see lines 400-406) in glacier catchments are high. The question addressed herein is how these loads affect biota. As the net effect on primary producers is negative or negligible (see section 3), it is challenging to consider any fertilization effect of these particles to be a significant feature that should be discussed compared to those effects which can explain the observed patterns of primary production and water column distributions.

It is a shame the fate of these particulates are not discussed in a balanced manner, and only somewhat negatively, if at all.

The effect of particles on scales of 1-100 km from glaciers in the Arctic is apparently negative according to measurements of primary production (see new section 3). We do not think it is unbalanced to focus on effects that could explain this, rather than effects that would have the opposite effect.

It is very hard to reconcile this with the hypothesis the reviewer is referring to [that these particulates have a fertilizing effect]. We attempt to explain why these different perspectives may have arisen in new lines 400-428.

We have reviewed carefully the literature concerning glaciers and primary production in glaciated Arctic regions. We cannot find evidence that glacially derived particles are linked to high primary production in the Arctic despite extensive comments in recent glaciology papers, without supporting citations, that this is the case. The argument in (Wadham et al., 2019) for example cites work that refers to secondary production (Lydersen et al., 2014), not primary production, in Kongsfjorden and this high secondary production is thought to occur precisely because of the negative effects that particle plumes have on primary producers (new lines 508-517). The reference supports the opposite conclusion, that glacier plumes negatively affect primary production.

There is literature discussed herein to show mechanistically that meltwater/particles (on small scales the effects of the two are very hard to distinguish from field observations) in the Arctic have effects on the balance between different taxonomic groups, potentially on bloom timing, mixing (which does facilitate increased primary production) and C export; but we are not aware of any work specifically reporting positive effects of meltwater (in the absence of subglacial discharge inducing mixing) or lithogenic particles in Arctic glacier-fjords (or associated regions) on total marine primary production. There are also several cases where references (as per the Lydersen reference) are miss-cited to imply such a link, for example the Arrigo et al., 2017 manuscript has been cited on multiple occasions to link increasing freshwater with increasing productivity, but the manuscript shows no link between increasing freshwater and increasing productivity, it shows a link between freshwater arrival and bloom timing which is a different concept. Similarly Meire et al., 2017 has been cited to link productivity to meltwater, but the manuscript shows that no such general relationship exists- except for the special case of marine-terminating glaciers. The concept of 'bioavailability' is also been extensively mis-used in recent literature to refer to elemental which are at most 'bioaccessible under some circumstances' which perhaps explains why there is a general miss-perception, as suggested by the reviewer, that these particle plumes are 'highly productive'.

The comments we can find referring to a particle-fertilization effect in the Arctic are circular, they refer back to papers (largely recent Hawkings/Wadham references) which present large lithogenic fluxes and speculate that they may be driving enhanced primary production. But the link is unsubstantiated, and what data we can find for the Arctic shows a negative (not positive) association (see section 3).

L303-307 I think this should come earlier – after line 283.

Changed as noted above.

L319-323 As the authors mention elsewhere, turbidity is important in suppressing surface productivity (via light limitation), which should be mentioned here. Discussion of new work by Seifert et al. (2019) should also be discussed in the context of carbon removal.

The discussion concerning particle plumes and turbidity is now expanded as per earlier suggestions, In the specific sentences here, this (light limitation) is not particularly the case. Whether mid-summer productivity is controlled by only light-limitation or macronutrient-limitation can be assessed by looking at nutrient distribution in near-surface waters. Recent work in this fjord (Holding et al., 2019) suggests primary producers are well adapted to the light conditions in summer and that light-limitation was only a significant proximal-control on primary production at the inner-most fjord station with total primary production instead limited by sparse nitrogen supply. The Seifert work is now discussed in a new section (5.2).

L417-421. This is the first reference to any benthic processes occurring. This is an oversight of the current manuscript, and this deserves discussion. Studies in both polar regions have investigated benthic recycling and diagenetic processes and the authors should discuss this as well (see Wehrmann et al., 2014, Henkel et al., 2018, Buongiorno et al., 2019).

As noted, an extensive discussion of benthic processing is beyond the title of the current manuscript (as per other related themes). We chose our title to be as tightly defined as possible, an extensive review branching out to benthic processes, supra-glacial processes, sea-ice processes, icebergs and dust- would be more comprehensive, but far beyond what we can achieve in a single text. Comments are added however to develop the benthic NH4 story we already alluded to (Halbach et al., 2019) but not flagged as a 'benthic process' in Kongsfjorden and to emphasize the overlapping nature of benthic inputs with meltwater inputs to the ocean (as we already allude to with Fe, new section 5.2).

**449 Can be a few 10s km where turbulent plume is observed and can be spatially variable with time (Tedstone et al., 2012, Hudson et al., 2014).**

We are aware of this, a line is added (re-)emphasizing that these are very broad generalisations. It is not our intention to provide a 'standardised' conceptual model as we note throughout that glacier-fjords across the Arctic are all practically unique and plumes can vary from not being evident at all in surface waters, to surface plumes extending 10s of kilometres along fjords (but are typically more restricted).

L456-458 This is one of several reasons why the limiting nutrients are likely differ. What about riverine inputs, dust inputs etc...?

We have tried to keep the text as focused as possible on the Arctic and an in depth review of differences in Fe sources between the Arctic and Antarctic is detail we do not wish to go into. To a first approximation, the critical difference is the vast difference in remoteness, the increased shelf exposure of the Arctic covers the associated shelf/river influences (we have added a sentence to explain this).

471-471 This is slightly misleading. The coastal regions of Antarctica have low Fe concentrations but there are now several studies highlighting the potential importance of glacial inputs. Further Figure 5 misses out PFe concentrations from Marsay et al. which are consistently >1 nM.

We disagree with this comment, even very close to the glaciers where we can find available data there is evidence of residual nitrate and the potential for dFe limitation. If Fe from glaciers is to have an immediate positive effect on marine primary production, it has to mix into surface high macronutrient, low dFe waters. Under these circumstances, the dFe supplied will then rapidly be drawn down to low levels (unless macronutrients become depleted- which for NO3/PO4 doesn't occur on any large scale around Antarctica). Even in productive areas of coastal polynyas around Antarctica this is the case e.g. new lines 707-718). In any case, low dFe concentrations cannot be used to infer a low total Fe supply as low dFe is typical of post-bloom conditions, even in places not considered to be Fe-limited or HNLC (e.g. Celtic shelf, Birchill et al., 2017). It is not misleading to state that dFe-limitation occurs close to Antarctic glaciers, on the contrary, if it didn't then there wouldn't be such a strong biological response to new dFe input in summer.

Figure 5 does not 'miss' anything essential for the interpretation with respect to Fe limitation or primary production. There are obviously only so many parameters we can show and these 3 are sufficient to see the general contrast between the two cases. Fe limitation in the ocean can be (and is) assessed in marine waters by looking at the ratio in availability of dissolved Fe to NO3 (Moore et al., 2013), this approach quantitatively assess the extent of Fe stress in cells even working across very broad Fe gradients (e.g. (Browning et al., 2017)) where particulate Fe concentrations vary from high to low alongside DFe gradients. This means either than the direct influence of particulate Fe is via the dissolved phase (i.e. the influence of particulates on Fe bioavailability is accounted for in dissolved Fe measurements) or that, if directly available to some organisms, direct particulate Fe uptake is very minor compared to that of dissolved Fe. Neither of these suppositions is surprising considering that most Fe-cellular uptake pathways are specific to organically complexed Fe or free Fe rendering particulate Fe far less accessible to most species (Shaked and Lis, 2012).

Further the authors in this study comment that measurements come following 2 months of intense primary productivity (i.e. these are not traditionally limiting waters, but a productive coastal ecosystem).

We referred to these waters as 'high nitrate, low dFe' which is correct. We did not refer to the levels of productivity. It is not clear what the reviewer means here, productivity and nutrient-limitation are different concepts. There is almost invariably a nutrient proximally limiting phytoplankton growth (except perhaps in extremely productive eastern-boundary upwelling systems). Highly productive regions of the Southern Ocean can still be (and generally are) Fe-(co)-limited and NO3 is not fully depleted during the growth season (e.g. Sedwick et al., 2011).

**L552** This is a rather low estimate of DFe from a grounding line and there is very little information available on concentration estimates. I'm not quite sure how the authors came to this value from Marsayetal. (2017), so it would be useful to provide a sentence to elaborate

The estimate of dFe released beneath an ice shelf is for freshwater ice melt, not for subglacial discharge— as this doesn't exist close to the edge of most ice shelves in the same way as it does for a marine-terminating glacier where subglacial discharge plumes are pronounced at the glacier terminus. It is therefore not derived from Marsay et al., it comes from the freshwater studies cited. We can acknowledge uncertainty in this value (there are no direct measurements to quantify it), but also again note that the vast majority of uncertainty in this calculation comes from the estuarine removal of Fe species during mixing between saline and fresh waters. This dwarfs the uncertainty from any other source (see new lines 804-815).

L584 Completely agree and pertinent point to make given we know almost nothing about ligand binding in glacial fjords (and very little in estuaries more generally). However, I think the perspective here is mainly focused on the idea of bioavailability in "open ocean" waters, which is almost certainly controlled by ligand binding (what this does to the bioavailability I think is still poorly understood given the wide range and complexity of metal stabilising ligands). An increasing number of studies (Kranzler et al. 2011, 2016, Shoenfelt et al. 2017, Grimm et al. 2019) are demonstrating the importance of accessing Fe from particulate pools yet there is very little discussion of this. Surely in coastal areas the particulate pool is likely to be very important given the high concentrations (Schroth et al., 2014) and is almost as poorly understood as the ligand pool? Some balanced discussion of this is important.

Direct accessibility of particulate Fe to pelagic phytoplankton is a bit of a misnomer, there are specific examples of mechanisms individual organisms have developed to capture Fe from particulate sources (Rubin et al., 2011), but cellular uptake processes are overwhelmingly dependent upon dissolved Fe availability. This can be demonstrated in the ocean at large (including high particulate Fe coastal regions) by looking at the extent to which Fe stress corresponds to dFe concentrations; dFe availability explains almost perfectly the extent of Fe-limitation across regimes transiting from high to low particulate Fe, meaning that dFe is to a first approximation the principle factor in determining Fe-limitation (Browning et al., 2017).

The role of particulates is generally understood to be as a buffer of the dissolved pool. Further, as noted, we have specifically focused the text on the Arctic where Fe is not an extensive limiting factor for primary production and thus its biogeochemistry is of much less interest than were we reviewing a similar topic in the Southern Ocean.

Just because particulate concentrations of Fe are high does not mean that organisms will change their biochemistry to access this less labile Fe rather than relying on dissolved Fe. As noted in earlier comments, the major cellular Fe uptake systems all rely on dissolved organic-Fe species. We are not sure that these references support the point the reviewer is making, the Kranzler 2016 work explicitly demonstrates that transformation from the particulate to dissolved phase is required prior to uptake, which is consistent with our comments on the utilization of these particles by biota (308-315).

*L608-609 I don't know what other bedrock types the author's think are likely, but carbonate and silicate bedrock broadly covers them all.*

**Rephrased.**

L620-643 p1: Linked again to my point about lack of discussion on the importance of benthiccycling, Ithinkthereshouldbesomediscussion of the potential role of alkalinity production in sedimentary environments (e.g. via denitrification and sulfate reduction).

**R: This is not directly connected to meltwater and is a generic shelf process which we think is well beyond the scope of the title.**

L620-643 p2: Some contextualisation is needed here. To be my knowledge (and I am definitively not an expert in this) but there tends to be a conservative decline in alkalinity in most estuarine settings (see Cai et al., 2010 and Thomas et al. 2009 for example), so this is not unique. The trend of decreasing alkalinity with increasing freshwater is therefore not particularly surprising in the context of freshwater-saltwater continuum environments as a whole. I agree that monitoring these changes with increasing meltwater discharge will be an important future undertaking.

p3: I think some additional detail in this section would be useful for readers. Could the authors also consider an alternative scenario whereby glacial meltwater have low pCO2 and high pH as glacial meltwater tend to be elevated in pH and correspondinglowpCO2(Tranteretal. 1993,

SharpandTranter, 2017)? Forexample, there is currently no mention of the conclusions of Meire et al. (2015), which shows glacial melt water associated with low pCO2 regions of the fjord. Also see recent studies by Pilcher et al. (2018) and St Pierre et al. (2019).

620 Correct, it is a generally correct statement to state that freshwater generally amplifies ocean acidification, but as noted in the text glacial meltwater has a particularly low TA which means that meltwater is a much more potent acidifier than riverwater. We have added some sentences here to provide more basic detail on the carbonate cycle as it is easy to get confused. The freshwater pH doesn't really matter in terms of to what extent freshwater drives ocean acidification, i.e. it's not possible for freshwater with low TA and high pH to act as a counter-balance to ocean acidification, freshwater with low TA will always acidify because of its buffering capacity (alkalinity), which is invariably low (a few local exceptions to this rule are speculated to occur e.g. see Benetti et al., 2019, but all of the large datasets we can find show meltwater is a low TA freshwater source).

We would rather not raise confusion here by discussing freshwater pH which varies so much in glaciated catchments precisely because of the low TA. In meltwater-affected saline waters, several non-conservative effects come into play, and the carbonate system is further affected by the extent of primary production which lowers pCO2 and thereby pH, and the general under-saturation of pCO2 in meltwater (which increases pCO2 drawdown). The saturation state of meltwater and estuarine waters with respect to pCO2 is also now discussed briefly as a related issue and the non-linear effect of salinity on pCO2 also raised (new lines 889-895).

**L649 L649: What is meant by a DOM concentration? Do you mean DOC concentration**

DOM refers to dissolved organic material, DOC refers explicitly to dissolved organic C, although these two are often used inter-changeably in the literature given that the majority of DOM is DOC. We will make sure to define these at first use.

L689 L689-695: All of the samples in the Holding et al. (2016) bar one are taken from salinities above 34 therefore it's not particularly surprising a clear signature of glacial DOC is observed in bacteria here. Additionally, there is no mention of a glacial DOM in algae, some of which are likely to be mixotrophic as commented in this manuscript (i.e. the interpretation is not straightforward). In this context I really don't think you can consider the Holding et al. estimate of ~11% of bacterial OC in marine waters to be from glacial DOM as minor. It is worth mentioning that other studies (e.g. Fellman et al., 2015, Hagvar et al., 2016) much closer to glacial inputs have found assimilation of glacial DOM into food webs. This is much debated, but one part of the story is that glacial DOM is highly bioavailable (as observed by a number of studies) and is therefore likely consumed very close to the glacier front.

Given the title of the text we are primary concerned herein with the effects of meltwater in the marine environment and are therefore much more interested in the broad-scale response of biogeochemistry across saline areas than in freshwater plumes. It seems obvious that in a freshwater system, all of the DOC will be freshwater-associated, the question we are interested in here is whether any freshwater signals can be detected offshore.

L695 Paulsen et al. (2018) isn't the correct reference to use in this context and is slightly misleading. This study shows that bioavailability is influenced by glacial meltwater inputs not that it is a minor component of bacterial consumption. **The study explicitly demonstrates that glaciers '***are not a major contributor of carbon or of FDOM in the system***' that '***the significant amounts of BDOC in glacial runoff reported byHood et al. (2009), Fellman et al. (2010), and Lawsonet al. (2014), may, in fact, be negligible compared to the degradation potential of the various autochthonous carbon sources that are already present in the fjord***'. This supports the sentence as cited in the revised text.**

Section 8: This whole section feels extremely speculative to me and is not actually correlatedtorealworldobservations, norwithanyobservationsfromtheArctic. Mostof theliteraturecitedprovidestenuouslinkswiththeonlyevidencethatIcanseebasedon the observation that HABs occur in Patagonia and that there are glaciers in Patagonia as well (but not in the same locations at the HABs). The main study cited (Leon-Munoz et al., 2018) was conducted in fjords with very little or no glacial cover, and contains no reference to glaciers, or meltwater inputs. I'm not against the inclusion of some points from this section into the next section (long-term effects of glacier retreat), as it's important to form hypotheses for testing (especially when anticipating future change), but it needs to be significantly toned down, reduced and explicit mentioned that the hypotheses are speculative.

We describe this as a 'hypothesis'. Most of the section is a description of reasonably uncontroversial literature concerning patterns in primary production and stratification. The link between meltwater and stratification is well established, and the link between HABs and stratification is well established. We can of course flag that connecting these two observations with a hypothesis (that changes in glacier-discharge may affect HABs) is not well establishedalthough we have added new references from Alaska and Greenland which mechanistically explain why glacier fjords in these locations may be increasingly affected by HABs in the future.

We have also expanded the rationale behind this potentially being of relevance to the Arctic as there are HAB-forming species present in stratified areas around west-Greenland. The main study site used in *Leon-Munoz et al.*, is not an area where meltwater is the major source of freshwater, but the regional data presented covers areas which do have a majority of local freshwater inputs from glaciers, where changes in glacially derived freshwater inputs are affecting stratification and seasonal patterns of primary production-hence the link to long term changes in glacier fjords. We clarify this more in the revised text.

With respect to studies not referring to 'glacier' or 'meltwater', this is common in oceanographic studies because freshwater sources cannot be easily distinguished at distance from source. Many of the Greenlandic studies referred to similarly refer to 'meteoric water' or 'freshwater' with the meltwater component of this freshwater component being highly variable.

L749-750 I disagree with some of the glaciological interpretation in this paragraph. The study cited (Bliss et al., 2014) is a modelling study to predict future meltwater runoff terms, with no observed data presented (yes future estimates of mass change and runoff are given and are useful, but that is not how this study was cited). This is especially problematic in Patagonia, where there is a relative dearth of data to use for model inputs/validation. There is no evidence to suggest that glacial runoff is in long term decline in this region. The opposite is actually likely to be true with regard to the Patagonian Ice Fields (see recent studies of Forresta et al., 2018, Richter et al., 2019, Li et al., 2019), which are currently the largest contributor to sea level rise per unit area in the world. Glacial meltwater runoff is not intricately linked to precipitation as per non-glacial rivers, but reduced precipitation is likely to amplify mass balance losses. Yes the wording here was incorrectly matched to the reference, we have split this sentence and separated the observations of glacier retreat and reduced runoff, and introduced the concept of peak discharge from future scenarios in model studies which is what we meant to refer to.

L829-830 I don't disagree but references needed here to substantiate point.

We were referring to the studies already cited in the same sentences, but can repeat them for clarity.

Figure 9: Nice looking figure, but I'd really like to see more balance in the interpretation oftheliteraturerepresentedinit. Onemajoromission(againl'mgoingbacktoit)isany lack of benthic feedback. "Sedimentation and Carbon(/nutrient) [burial]" is seen as a one way process here, which is unlikely to be true (see works by Wehrmann amongst many others).

Figure 9. Yes this can be changed.

L945-947: Recommend updating these figures with new data available in Mouginot et al. (2019).

Yes these can be updated, for the purposes of ranking glaciers by discharge there is some small difference depending on the time period chosen.

References referred to (which are not in the main text):

Birchill, A. J., et al. (2017), Seasonal iron depletion in temperate shelf seas, Geophys. Res. Lett., 44, 8987–8996, doi:10.1002/2017GL073881.

Rubin, M., I. Berman-Frank, and Y. Shaked. 2011. Dust and mineral-iron utilization by the marine dinitrogen-fixer Trichodesmium. Nat. Geosci. **4**: 529–534. doi:10.1038/ngeo1181

---

## Referee Report (RR1)

The authors clearly demonstrate their diligence in covering the scope of ice-ocean biogeochemical research. It is apparent that more effort should be devoted to describe the variability of these ecosystems and their responses to regional forcing, with particular emphasis on continued monitoring and automation, under a wider geographical lens (both hemispheres). The developing portrait of the ice-ocean interface may be obscured by shifts in baseline processes — therefore, the case-study approach here is particularly effective at capturing spatial and quasi-time variability, by comparing distinct regions and exchanging "time" for "place." The calculations made in this review, I think, should motivate others to constrain the large uncertainties. Each layer of complexity presented has better informed the modeling community and I think this review will serve the scientific community well to think about multiple controls on primary production in dynamic coastal regions. The authors have whole-heartedly considered initial comments from the reviewers and made the appropriate changes. The review is formatted well and it is easy to extract the summarized findings. It is therefore my recommendation that the review be published in TC.

---

## Referee Report (RR2)

**Review of "How does glacial discharge affect marine biogeochemistry and primary production in the Arctic?" by Hopwood et al.**

I thank Hopwood and co-authors for their detailed responses to my first review. Rather than responding to these comments I've concentrated on addressing the revised manuscript. As per my previous review, I think this manuscript is timely and very well written. The revised version is a clear improvement on the original manuscript, and I thank the authors for taking on board some of my points. I still have a general comment that I would like to be addressed in the next version of the manuscript and have some specific points that I think need to be addressed below. I would like to reiterate that I do not disagree with the authors main points, nor with their more general overarching conclusions. However, I would still like to see some more balance discussion and use of the literature, and I do not think it is particularly helpful to use some references purely to argue when hypotheses appear to be wrong when there are many opportunities for them to be used appropriately elsewhere.

My main general point concerns section 9.0. I understand why the authors want to include it and it is an interesting hypothesis that deserves future attention, but I do not think the current evidence supports its inclusion as a separate chapter over and above section 10.0 (Insights into the long-term effects of glacier-retreat). Many of the "known" factors that help create HABs are somewhat specific to certain regions with most of the literature cited in this case from Patagonia (and do not apply to locations with high glacial coverage even if some HAB species are present). I know of no known location where there is a link between changes in glacial meltwater supply and HABs (yet - I'm sure the authors will correct me here if I'm wrong), and the additional references given do not really help to reinforce the main point of the chapter. I understand that the literature on this is uncontroversial when it comes to Patagonian fjords (with a strong emphasis on non-glacially fed fjords), but I do not fully accept a tentative link above and beyond non-glacial locations where hydrological regimes are changing (i.e. probably many locations) and leading to stratification change. There are other variables at play in Patagonian fjords that might not be applicable to Arctic fjords (e.g. nutrient loading for aquaculture as one example). This review is on the Arctic so a section that relies heavily on Patagonian literature feels like a stretch as a separate section.

Specific comments:
L167-170: I'm curious how this estimate of pan-Arctic primary production compares with more recent estimates (if they exist). As the authors will be aware, estimates from ocean colour are dependent on the algorithm used (Lewis et al., 2016), especially in the Polar regions and when CDOM is sometimes very high (e.g. near Arctic rivers). The annual primary productivity looks very high on the west Greenland shelf in Pabi et al. (2008), which does not necessarily match with some observations (including those referenced). Do the authors have any knowledge of this? Perhaps this is still the best estimate.
L231: This is quite a good example of the point in my first paragraph (using references in a balanced manner). Yde et al. (2014) is a good paper and I wouldn't argue for not including it, but this is the only location in which it is referenced. There are other papers

that are referenced later in the text that describe the chemical composition of glacial meltwaters around Greenland just as well that could be used here but aren't.

L270-272: Generally agree (especially in locations like Patagonia where overdeepening has lead to many proglacial lake forming with glacial retreat) but is dependent on multiple factors are at play including bedrock topography, bedrock composition, hard/soft beds, previous climatological scenarios etc... Suggest "may decline" instead of "declines".

L370: There is probably a simple explanation to this, but where does the value of ~5.9 uM come from in Meire et al. (2016)? Table 1 in that paper suggests the meltwater river values are more like 30 uM.

L399-400: I think this might be missing some nuance that I suspect is mainly a disagreement with what can be considered a meaningful flux (which I don't really want to get into too much as we'll be going around in circles). As the authors know the argument in the e.g. "glaciological literature" that fluxes may be underappreciated is based upon very large fluxes of labile particulates that aren't necessarily reactive on timescales observed (e.g. the demonstration in Hawkings et al. 2017, that a large proportion of ASi dissolves in seawater over the course of several months to a year). I think the "larger picture" is based upon additions to marine nutrient inventories rather than direct fertilisation of marine phytoplankton. For example, the current estimates of silica budgets (e.g. Treguer and De La Rocha, 2013 and Frings et al., 2016) include reactive riverine particles as part of the Si inventory (albeit poorly understood and constrained), not necessarily because they contribute to direct diatom fertilization, but more because they add to the marine Si inventory. The authors touch on this later in the manuscript by reference some of the benthic literature, but it's still a potentially important distinction (again the authors might not agree with me).

L410-415: These articles argue that they may have a positive effect based largely around the reactivity of the particles, which has been demonstrated. The idea that particles are not important when they are sedimented is not strictly accurate either. Benthic reprocessing also includes diffusive fluxes from sediments (e.g. Frings (2017) and Wehrmann et al. (2014) estimates of diffusive Si and Fe/Mn fluxes from sediments are substantial).

L423-435: Nice paragraph!

L436: DOM should be "dissolved organic matter" not "dissolved organic materials". Please correct.

L475-480: Again, there is probably more relevant literature to cite in this context (i.e. seasonal variation in nutrient concentrations from Arctic glaciers), and this is the only location where Brown et al. (1994; a study on an Alpine glacier) is referenced. As I mentioned, it is unfortunate there is not more balance in the referencing here.

L487-491: This is quite disappointing and again emphases my point of lack of balance. I don't really think the point is substantiated here either. A synthesis "of available nutrient distributions in glaciated Arctic catchments, especially for Si and Fe" is a sentence that requires some context. The Fe and Si concentrations in these publications actually agree relatively well with others (in the context of "dissolved" concentrations), so it is perplexing why they are singled out here for criticism when some of these other studies also emphasise large fluxes of e.g. Fe (Bhatia et al. 2013, Stevenson et al., 2017). These studies also show why it is important to look as seasonal datasets due to large

variability in elemental concentrations over a melt season, yet this is not reference at all in the above discussion which is unfortunate. Please also see my point in L410-415.

L643-644: Doesn't need to be solely during the meltwater season. Diffusive release of elements from benthic environments will happen year-round.

L644: References should be Wehrmann et al. (2014).

Section 5.2: I think this section may benefit from some additional discussion on e.g. the role of Fe oxyhydroxides on the export of C to depth (i.e. the "rusty carbon sink"), and on possible adsorption of PO4 to particles (this was briefly mentioned in Cape et al., 2019).

L721-723: Perhaps in non-glacial rivers yes, but the N:P ratio in glacial rivers is much lower than typical Redfield ratio (16:1) if using end members in Table 3 (more like 7:1). Related to the point, but to me this suggests that DON contributes a significant amount of "biolabile" N, contribution from NH4, and/or adsorption of PO4 to particles (and removal from the fjord surface) are all possible. I think this also provides some nice context as to why these environments are likely to be N limited.

L751-758: Citing van der Merwe et al. (2019) would also be appropriate in this section.

L841-861: I think it would also be appropriate to cite van der Merwe et al. (2019) here as well.

L859: How about "how particle bio-lability, ligands and estuarine mixing processes moderate the glacier-to-ocean Fe transfer"

L936: suggest "minor source of total DOM to the fjord".

L943-947: I think there's a bit of repetition here. It is sufficient to say that glacial DOM appears to be more labile than non-glacial DOM because of the high proportion of aliphatic or protein-like compounds (e.g. Barker et al., 2013, Dubnick et al., 2010, Hood et al., 2009, Pautler et al., 2012) commonly associated with microbial activity.

L970-972: I think saying it is diluted and consumed is more appropriate. DOM is relatively unique in that the concentration doesn't really matter (to a degree) – the composition is most important. Even if it dilutes marine waters, it's (glacial meltwater) highly biolabile nature means it is likely to be relatively important for heterotrophs.

L973-976: This is complex and a this is perhaps a generalisation considering the complex mixture of organic compounds present and the relatively high salinity of these sampling locations.

L990-992: Only some may be photodegraded so this is an over generalisation. There is a standing stock of DON in the ocean surface that is pretty recalcitrant are resistant to biological utilisation (it is favourable for nitrogen fixation for example; Letscher et al., 2013).

L995: It is appropriate to reference Wadham et al. (2016) here.

L1002: Appropriate to reference Hawkings et al. (2016) here because it is one of the only papers that has and discusses values for DOP in glacial meltwater rivers (low to negligible concentrations).

L1048-1050: The previous references refer to catchments with barely any glacial meltwater input contribution to discharge and this should be made clear here as this sentence does not make that clear.

L1055: It is slightly misleading to say that freshwater runoff in these scenarios includes glacial meltwater. Please change to "non-glacial freshwater sources". As I noted in my previous review, meltwater discharge in Patagonia is not currently decreasing, as

opposed to non-glacial precipitation driven catchments where precipitation does appear to be undergoing a long-term decline.

L1083-1085: Completely agree that glacial discharge affects stratification, but doesn't it also supply very cold water to the surface. The argument of an increased prevalence of HABs in Patagonian fjords and the Alaskan case study (Vandersea et al., 2018) is that warming of the surface also plays an important role. I therefore don't fully understand the link between increasing glacial meltwater freshwater discharge in the Arctic, a warming fjord surface layer (despite the increasing flux of cold, very fresh water) and HABs.

Table 3: Lawson et al. (2014a) do not produce their own DON data. The concentrations used here are from the Wadham et al. (2016) reference that was "in review" when this article was published. This concentration estimate was likely based off a mean seasonal concentration rather than the discharge weighted concentrations given in Wadham et al. (2016) of 1.7 uM.

New references cited:
Barker, J.D., Dubnick, A., Lyons, W.B., Chin, Y.P. (2013) Changes in Dissolved Organic Matter (DOM) Fluorescence in Proglacial Antarctic Streams. *Arctic, Antarctic, and Alpine Research* 45, 305-317.
Dubnick, A., Barker, J., Sharp, M., Wadham, J., Lis, G., Telling, J., Fitzsimons, S., Jackson, M. (2010) Characterization of dissolved organic matter (DOM) from glacial environments using total fluorescence spectroscopy and parallel factor analysis. *Annals of Glaciology* 51, 111-122.
Frings, P. (2017) Revisiting the dissolution of biogenic Si in marine sediments: a key term in the ocean Si budget. *Acta Geochimica* 36, 429-432.
Frings, P.J., Clymans, W., Fontorbe, G., De La Rocha, C., Conley, D.J. (2016) The continental Si cycle and its impact on the ocean Si isotope budget. *Chemical Geology* 425, 12-36.
Lewis, K.M., Mitchell, B.G., van Dijken, G.L., Arrigo, K.R. (2016) Regional chlorophyll a algorithms in the Arctic Ocean and their effect on satellite-derived primary production estimates. *Deep Sea Research Part II: Topical Studies in Oceanography* 130, 14-27.
Pautler, B.G., Woods, G.C., Dubnick, A., Simpson, A.J., Sharp, M.J., Fitzsimons, S.J., Simpson, M.J. (2012) Molecular Characterization of Dissolved Organic Matter in Glacial Ice: Coupling Natural Abundance 1H NMR and Fluorescence Spectroscopy. *Environmental Science & Technology* 46, 3753-3761.
Treguer, P.J., De La Rocha, C.L. (2013) The World Ocean Silica Cycle. *Annual Review of Marine Science* 5, 477-501.
van der Merwe, P., Wuttig, K., Holmes, T., Trull, T.W., Chase, Z., Townsend, A.T., Goemann, K., Bowie, A.R. (2019) High Lability Fe Particles Sourced From Glacial Erosion Can Meet Previously Unaccounted Biological Demand: Heard Island, Southern Ocean. *Frontiers in Marine Science* 6, 332-332.

---

## Author Response (AR2)

Response to reviewer's comments shown as blue text.

The reviewers and editor are thanked for their further time considering the text, especially over the holiday season. For clarity in responses below, lines refer to R1 as per the reviewer's comments. Revisions are annotated in the updated text (R2).

**Reviewers' comments.**

The authors clearly demonstrate their diligence in covering the scope of ice-ocean biogeochemical research. It is apparent that more effort should be devoted to describe the variability of these ecosystems and their responses to regional forcing, with particular emphasis on continued monitoring and automation, under a wider geographical lens (both hemispheres). The developing portrait of the ice-ocean interface may be obscured by shifts in baseline processes — therefore, the case-study approach here is particularly effective at capturing spatial and quasi-time variability, by comparing distinct regions and exchanging "time" for "place." The calculations made in this review, I think, should motivate others to constrain the large uncertainties. Each layer of complexity presented has better informed the modeling community and I think this review will serve the scientific community well to think about multiple controls on primary production in dynamic coastal regions. The authors have whole-heartedly considered initial comments from the reviewers and made the appropriate changes. The review is formatted well and it is easy to extract the summarized findings. It is therefore my recommendation that the review be published in TC.

**R:** The reviewer is thanked for comments on the text.**

**[end of review]**

I thank Hopwood and co-authors for their detailed responses to my first review. Rather than responding to these comments I've concentrated on addressing the revised manuscript. As per my previous review, I think this manuscript is timely and very well written. The revised version is a clear improvement on the original manuscript, and I thank the authors for taking on board some of my points. I still have a general comment that I would like to be addressed in the next version of the manuscript and have some specific points that I think need to be addressed below. I would like to reiterate that I do not disagree with the authors main points, nor with their more general overarching conclusions. However, I would still like to see some more balance discussion and use of the literature, and I do not think it is particularly helpful to use some references purely to argue when hypotheses appear to be wrong when there are many opportunities for them to be used appropriately elsewhere.

R: We acknowledge that there are areas where we disagree with the lines of argument raised by the reviewer. However, we do not think our newly-revised discussion is unbalanced. There are inevitably going to be a few places in such a broad text where we can always add more references and thank reviewers for suggestions accordingly. In some of the specific examples (see below) where it is claimed that the choice of references is unbalanced, our choices are very similar to that in other similar recent synthesis (e.g. the Wadham 2019 review cited, or the short related comments in Chapter 3 of the IPCC O&C report), thus if we are unbalanced we are no more so than in other comparable texts.

We understand the reviewer is passionate about seasonal shifts in freshwater nutrient chemistry, but these shifts aren't significant enough to be evident in the corresponding marine environment and are not really relevant to the topic of marine biogeochemistry/primary production (PP) (see comment below). The seasonal changes in freshwater nutrient composition are far less significant in terms of potential effects on biota that seasonality in many other variables/phenomena which

**are, at most, lightly covered in the text. Therefore, we feel that it would be unbalanced to single out seasonality in freshwater nutrient content as a particularly important area to cover.**

My main general point concerns section 9.0. I understand why the authors want to include it and it is an interesting hypothesis that deserves future attention, but I do not think the current evidence supports its inclusion as a separate chapter over and above section 10.0 (Insights into the long-term effects of glacier-retreat). Many of the "known" factors that help create HABs are somewhat specific to certain regions with most of the literature cited in this case from Patagonia (and do not apply to locations with high glacial coverage even if some HAB species are present). I know of no known location where there is a link between changes in glacial meltwater supply and HABs (yet - I'm sure the authors will correct me here if I'm wrong), and the additional references given do not really help to reinforce the main point of the chapter. I understand that the literature on this is uncontroversial when it comes to Patagonian fjords (with a strong emphasis on non-glacially fed fjords), but I do not fully accept a tentative link above and beyond non-glacial locations where hydrological regimes are changing (i.e. probably many locations) and leading to stratification change. There are other variables at play in Patagonian fjords that might not be applicable to Arctic fjords (e.g. nutrient loading for aquaculture as one example). This review is on the Arctic so a section that relies heavily on Patagonian literature feels like a stretch as a separate section.

R: The mechanism concerned here, freshwater-driven stratification, is not specific to certain regions, it's generic to many coastal environments [see also the reviewer's comment on temperature/stratification which we feel is incorrect, as it is well established that addition of cold, fresh meltwater into glacier fjords leads to strong stratification and thus very warm surface waters in summer, which is why these locations are considered at risk of HAB increases].

Contrary to the reviewer's comment, the Joli et al., (2018) reference specifically explains the meltwater-driven mechanism that is associated with the distribution of HAB species in N Greenland and may lead to HAB expansion in the Arctic. This is the same mechanism that operates in non-glaciated catchments with freshwater discharge.

To clarify, we have reduced this former section, and we accept that it is the least well-covered in terms of existing Arctic literature. In the revised text, these comments are condensed to three paragraphs to summarise the few papers available and why it is an interesting hypothesis for future work (For clarity the deleted section is removed entirely in the tracked changes text and only shown once where added as 10.1).

**Specific comments:**

L167-170: I'm curious how this estimate of pan-Arctic primary production compares with more recent estimates (if they exist). As the authors will be aware, estimates from ocean colour are dependent on the algorithm used (Lewis et al., 2016), especially in the Polar regions and when CDOM is sometimes very high (e.g. near Arctic rivers). The annual primary productivity looks very high on the west Greenland shelf in Pabi et al. (2008), which does not necessarily match with some observations (including those referenced). Do the authors have any knowledge of this? Perhaps this is still the best estimate.

R: There are several updates to this trend, which generally show that PP (primary production) has been increasing over the past decade across most (almost all) Arctic basins (now added in text). These are broad regional surveys, they thus integrate areas of high PP and low PP within specific geographical regions. There is not necessarily a miss-match with any shelf region, but any "average" will invariably smooth hotspots of PP into a lower regional average, especially where any sort of high PP region has sharp fronts with a lower PP region (e.g. at a shelf break) or a short bloom duration (most areas). We used the 2008 reference (some later work is available by the same group) because it is one of the few Pan-Arctic means available to quote and corresponds better to the dates of most PP measurements shown in section 3. This doesn't affect our subsequent discussion as it is a pan-Arctic phenomenon largely related to warming/increasing open water days. (Any more recent value would slightly increase the pan-Arctic mean and thus PP of each of the glacier fjords relative to an Arctic average would be lower).

Concerning the W Greenland shelf, the values discussed herein are in glaciated catchments and almost invariably inshore of the high productivity observed at the shelf edge, thus there is no contradiction with satellite records which generally cut off most glacier-fjord and inshore systems. Qualitatively, satellite data does match the high productivity measured (cited herein) in some areas of Disko Bay (among the furthest PP points away from the coastline in the data compilation), but is of limited use in most other glaciated 'catchments' because the scale is below that at which satellite data can resolve chl a.

Concerning satellite a logarithms, yes there is extensive debate about what this derived measurement actually represents and what fraction of chl a near the ocean surface (i.e. to what depth) satellite derived estimates penetrate. Never-the-less recent satellite work is extensively calibrated against in situ measurements and there are no other methods for looking at such broadscale PP shifts. The CDOM interference is confined to areas of extensive river plumes and attempts are made to calibrate for it, it doesn't affect glaciated catchments where freshwater leads to low CDOM. The main problems in glaciated regions are particle plumes and the difficulty resolving satellite data within fjord regions.

L231: This is quite a good example of the point in my first paragraph (using references in a balanced manner). Yde et al. (2014) is a good paper and I wouldn't argue for not including it, but this is the only location in which it is referenced. There are other papers that are referenced later in the text that describe the chemical composition of glacial meltwaters around Greenland just as well that could be used here but aren't.

R: It is not clear what the reviewer wants us to do here. There are 20+ papers that could be cited in this section, when making general points like this we have selected a range of literature to cover the breadth of the subject discussed with 2–3 given to complement each other here, for example, a site specific study covering a broad range of chemical properties, an Si piece with additional synthesis of all available macro-nutrient N/P/Si data, and a pan-Greenland Fe piece synthesizing all available trace metal data.

L270-272: Generally agree (especially in locations like Patagonia where overdeepening has lead to many proglacial lake forming with glacial retreat) but is dependent on multiple factors are at play including bedrock topography, bedrock composition, hard/soft beds, previous climatological scenarios etc... Suggest "may decline" instead of "declines".

**Ammended.**

L370: There is probably a simple explanation to this, but where does the value of  $\sim$ 5.9 uM come from in Meire et al. (2016)? Table 1 in that paper suggests the meltwater river values are more like 30 uM.

R: As stated in the text, this value refers to Kongsfjorden and is not from the Meire work which concerns Godthabsfjord.

L399-400: I think this might be missing some nuance that I suspect is mainly a disagreement with what can be considered a meaningful flux (which I don't really want to get into too much as we'll be going around in circles). As the authors know the argument in the e.g. "glaciological literature" that fluxes may be underappreciated is based upon very large fluxes of labile particulates that aren't necessarily reactive on timescales observed (e.g. the demonstration in Hawkings et al. 2017, that a large proportion of ASi dissolves in seawater over the course of several months to a year). I think the "larger picture" is based upon additions to marine nutrient inventories rather than direct fertilisation of marine phytoplankton.

R: We agree there is a clear distinction between long-term nutrient inventory change, and shortterm nutrient availability change (i.e. a compound present at low concentrations in meltwater is adding to the ocean inventory on geological timescales, but meltwater decreases the short-term availability in regional surface waters due to dilution and/or stratification). Yet the papers concerned discuss fluxes in the context of inter-annual changes, not in terms of the geological ocean inventory.

If fluxes are not evident on annual timescales, they can't be discussed in the context of annual changes in the marine environment, and especially not in the context of fertilization, which is how the fluxes referred to are presented, discussed, and interpreted (as summarised in Wadham et al., 2019, in the literature in general and in the IPCC O&C). The larger picture of geological timescale shifts in ocean inventories is indeed interesting, but that is not discussed at all in the references cited and thus it is difficult to infer that this is the message the authors intended to convey.

We find that the comment 'dissolves in seawater over the source of several months to a year' is not quantitatively consistent with any case study we can find (see specific examples below). If the fluxes were occurring due to dissolution on annual timescales, they would still be evident in water masses transiting through glacier-fjord regions on these timescales.

For example, the current estimates of silica budgets (e.g. Treguer and De La Rocha, 2013 and Frings et al., 2016) include reactive riverine particles as part of the Si inventory (albeit poorly understood and constrained), not necessarily because they contribute to direct diatom fertilization, but more because they add to the marine Si inventory. The authors touch on this later in the manuscript by reference some of the benthic literature, but it's still a potentially important distinction (again the authors might not agree with me).

R: This is an important distinction, but the papers discussed herein are not discussing this effect, they are discussing short term annual fluxes, arguing that dissolution occurs in these short timescales, and the potential effects on short term biogeochemistry/productivity — which is what we remain critical of. If the total fluxes referred to are not meant to be interpreted as annual fluxes associated with runoff (which is how they are presented, cited and discussed in later work by the same/similar authors including the recent IPCC report), but instead as long-term geological fluxes into the ocean nutrient inventory, then they would for example also be inclusive of the benthic flux associated with dissolution of these particles (whereas the fluxes critiqued are presented by the same group/authors as being additional to the near-glacier benthic flux, which is inconsistent with the reviewer's comment and supports our interpretation of the data).

If a labile element is dissolving in seawater and adding to the dissolved nutrient inventory (i.e. PO4/NO3/Silicic acid), then the resulting dissolved nutrient will be present in seawater flowing away from the glacier, especially where this 'outflow' is subsurface and not subject to pronounced drawdown by productivity in surface waters (circulation through these regions occurs over a

similar timescale of a few months to a year). So any dissolved inventory, or calculation concerning the flux derived from the dissolved inventory (e.g. Fig 4) in outflowing glacially modified waters is entirely inclusive of any dissolution occurring over short time periods in the water column (or as a benthic flux into the same water mass). These dissolved nutrient inventories are not 'missing' a dissolution term, they are inclusive of it.

Math to prove the above point is unfortunately challenging to do accurately in most of the catchments discussed in the review paper, because we cannot accurately constrain the time it takes for a parcel of water to enter and leave each glacier-fjord system (the residence time), but there is one large Arctic catchment (unfortunately not one of these 5) this can be done reasonably accurately with. Work by Wilson & Straneo (2015), and others, constrains the residence time of water from the 79 North Glacier, one of the largest discharge sources in N Greenland, as about 4-11 months beneath the floating glacier ice tongue i.e. comparable to the timescale over which the reviewer suggests a large amount of dissolution should occur. This catchment is particularly useful to study in terms of abiotic processes as the vast majority (~85%) of freshwater outflow occurs at depth so any nutrients released from meltwater/particles are not subject to pronounced drawdown in the photic zone.

Full depth profiles of all nutrients are available for this catchment e.g. https://doi.pangaea.de/10.1594/PANGAEA.905347 and https://doi.pangaea.de/10.1594/PANGAEA.879197

Looking at the change in Si concentrations contrasting inflowing Atlantic water, and outflowing glacially modified Atlantic water, from the 79 NG fjord system, the enrichment in Si is within uncertainties, i.e. there is no significant constant Si flux occurring from the 79NG system into the Atlantic after water has been beneath the glacier tongue for a timescale comparable to that over which it is suggested dissolution should be occurring. This is very difficult to reconcile with the argument that ASi is dissolving and releasing large quantities of dissolved Si, if this were the case most of the enrichment that could possibly occur from ASi dissolution and enrichment of Si should be evident. However, it is consistent with global biogeochemical models which do not prescribe, or require in order to match observations, a specific ice sheet Si source in the North Atlantic/Arctic.

This is not an isolated example. There are several well studied water masses in glacier fjords (Kongsfjorden is one example) where water enters in winter and then becomes trapped through summer behind sills in inner fjord environments. Such waters are then exposed to high particle loads through the year before being mixed again in autumn and should thereby accumulate dissolved nutrients from processes such as particle dissolution and benthic release. In Kongsfjorden, we cannot find any Si data for such trapped water masses above around 1-10  $\mu$ M (after 3-6 months of exposure to high particle loads), again suggesting that the proposed net dissolution by Hawkings et al., is unrealistically high, or at least that other processes remove or counter act a net dissolution close to the glacier such that the distant effect of Si availability is limited. Likewise, full depth profiles for Godthabsfjord are available (from GEM, Fig. 4) and show that the net change in Si in deep waters flowing into the fjord is modest (about a +30% 'top up' of the inflowing Si over the time of its residence in the fjord-maybe a year or so- far short of what should be evident if the down-fjord Si export in surface waters/efflux from Greenland occurs primarily as a result of dissolution Hawkings et al.)

*L410-415: These articles argue that they may have a positive effect based largely around the reactivity of the particles, which has been demonstrated. The idea that particles are not important*

when they are sedimented is not strictly accurate either. Benthic reprocessing also includes diffusive fluxes from sediments (e.g. Frings (2017) and Wehrmann et al. (2014) estimates of diffusive Si and Fe/Mn fluxes from sediments are substantial).

R: It is not clear what the reviewer means by 'has been demonstrated', particle plumes are universally (as far as any literature we can find shows) a negative influence on primary production. The reactivity of the particles does not equate to a net positive effect on primary production, for example there is no doubt at all that large quantities of labile Fe are released from glacial outflows, but the primary effect of this material in these environments is to deprive phytoplankton of light and to cause death-by-ingestion to filter feeders, hence the high reactivity is irrelevant in the context of organisms exposed to this material.

Yes benthic fluxes are also important, but if the reviewer's earlier comments are correct, that these fluxes are intended to represent long-term accumulation of nutrients in the ocean inventory, calculations with 100% dissolution of lithogenic material to produce annual fluxes are already inclusive of any fluxes arising from re-working on this material and not additional to it. Thus, we find it challenging to follow the reviewer's argument and to reconcile it with the arguments as published.

The problem is not whether or not these fluxes are important, the problem is that fluxes arising from dissolution (or any other estuarine process) are already included in any calculation that derives the flux out of the 'glacier system' into the ocean based on observed marine concentrations, and that numbers derived from such calculations (see above examples) are far smaller than the annual fluxes presented.

**L423-435: Nice paragraph!**

L436: DOM should be "dissolved organic matter" not "dissolved organic materials". Please correct.

R: 'Material' and 'matter' are used interchangeably in the literature. To clarify, we now use 'dissolved organic matter' throughout.

L475-480: Again, there is probably more relevant literature to cite in this context (i.e. seasonal variation in nutrient concentrations from Arctic glaciers), and this is the only location where Brown et al. (1994; a study on an Alpine glacier) is referenced. As I mentioned, it is unfortunate there is not more balance in the referencing here.

R: These sentences lacked any references which was an oversight, we have now added appropriate literature. We do not understand the comment on Brown et al., here. This paragraph was written primarily by the glaciology co-authors and is a statement of relatively uncontroversial literature, Q-C relationships are investigated extensively in older literature and thus citing some older work here establishing the basis of Q-C relationships seems appropriate.

L487-491: This is quite disappointing and again emphases my point of lack of balance. I don't really think the point is substantiated here either. A synthesis "of available nutrient distributions in glaciated Arctic catchments, especially for Si and Fe" is a sentence that requires some context. The Fe and Si concentrations in these publications actually agree relatively well with others (in the context of "dissolved" concentrations), so it is perplexing why they are singled out here for criticism when some of these other studies also emphasise large fluxes of e.g. Fe (Bhatia et al. 2013, Stevenson et al., 2017). R: The sentence 'of available...... catchments' refers to the several pages of discussion preceding, perhaps the phrasing was odd/unclear. We can rephrase 'a synthesis and analysis of available marine nutrient distributions [above]'. The freshwater work critiqued does not agree with marine literature because the marine work in question is inclusive of any dissolution occurring from particles, whereas the critiqued worked argues that a far larger flux arising from the 'dissolution' of new particles should be added on top of the existing dissolved fluxes. Thus, the critiqued work proposes fluxes significantly higher than those which can be derived from the marine data discussed.

The papers in question (critiqued) specifically claim that dissolved nutrient fluxes from freshwater are generally under-estimated due to a lack of consideration of dissolution from labile particulates and that this may be to linked to primary production on this spatio-temporal scale. Other work we can find does not make such specific claims.

Bhatia et al., do make comments concerning the effect of iron in the Atlantic and attempt to link discharge and bloom properties in parts of the Labrador Sea citing a strong correlation between peak bloom magnitude and annual discharge volume. Yet the bloom referred to occurs before meltwater release in the region/years concerned. Thus, a cause-and-effect relationship seems questionable. The Stevenson et al., reference is largely concerning the isotopic signature of Fe around Greenland and does not contain any discussion of primary production or effects in marine waters other than a comparison of the concentrations measured to other work.

The sentence as written in R1 was however incorrect and shouldn't have referred to Fe following changes made to R0 in response to the reviewer noting that Fe fluxes are subject to overlap when the uncertainty on estuarine mixing is accounted for in the same way consistently. It previously stated in R1 'macronutrient fluxes', but then also 'Fe and Si' at the end. It should have been amended to drop 'Fe' and the Hawkings et al 2014 Fe reference from the sentence. Thus the corrected sentence reads: "the recently emphasized hypothesis that macronutrient fluxes from glaciers inteo the ocean have been significantly underestimated (Hawkings et al., 2016, 2017; Wadham et al., 2016) is difficult to reconcile with a synthesis of available nutrient distributions in glaciated Arctic catchments, especially for Si (Fig. 4)'.

The Wadham/Hawkings papers referred to are the only references that we are aware of that specifically claim glacier-to-ocean fluxes have been generally under-estimated because of a hypothesized under-appreciation of labile particulate dissolution, that link meltwater (and the associated particle dissolution) to fertilization/productivity and imply that increasing meltwater/nutrient fluxes will be increasingly important for marine nutrient supply. Accordingly, they are also the only manuscripts cited as substantiating this point in other reviews/synthesis (including Wadham et al., 2019 and several points in Chapter 3 of the IPCC O&C). Thus, our interpretation, if unbalanced, is at least representative of the communities' and the respective authors.

These studies also show why it is important to look as seasonal datasets due to large variability in elemental concentrations over a melt season, yet this is not reference at all in the above discussion which is unfortunate. Please also see my point in L410-415.

R: In freshwater, but this is not relevant to broad-scale marine processes because this term is very small (Figure 10). Seasonal variation in freshwater endmembers has no discernible broad-scale effect on marine nutrient availability in these locations compared to other factors- particularly in light of the effects induced by stratification. The freshwater nutrient addition/dilution is so small that even increasing freshwater concentrations by an order of magnitude wouldn't significantly change PP within uncertainties (see Fig 10) and thus we disagree that seasonal variations in

meltwater composition should be discussed extensively. There may well be other reasons why such seasonal changes are important in a glaciology context, but not generally with respect to marine primary producers.

For Godthabsfjord for example, if we take a low and a high estimate for the N concentrations in freshwater discharge, (0.5-3  $\mu$ M), and assume the variation is entirely due to seasonal shifts, and that the seasonal shift is maximised by a simple move between the two extreme N concentrations, the importance of freshwater for total N input compared to that required to support PP as per Meire et al 2017, would vary between maximum and minimum contributions of 1-7% to PP N uptake over the season. This can hardly be argued to be a significant feature. And this is oversimplistic as the stratifying effect of freshwater, which changes the water column structure and thus mixing and N supply, makes such a calculation meaningless.

Seasonality in general is of course important, but seasonality in light availability and properties, phytoplankton dynamics, subglacial discharge rate intensity and frequency, water column structure, feeding patterns and excretion related nutrient cycling, and the carbonate system are all much more significant that seasonal shifts in freshwater composition. It would be unbalanced to include a section on seasonal shifts in freshwater composition rather than any of these topics.

*L643-644: Doesn't need to be solely during the meltwater season. Diffusive release of elements from benthic environments will happen year-round.*

R: Thank you, we have re-phrased here to avoid misreading of this sentence.

L644: References should be Wehrmann et al. (2014).

**R: Corrected.**

Section 5.2: I think this section may benefit from some additional discussion on e.g. the role of Fe oxyhydroxides on the export of C to depth (i.e. the "rusty carbon sink"), and on possible adsorption of PO4 to particles (this was briefly mentioned in Cape et al., 2019).

R: We are aware of work in prep (or possibly now in review) on both of these research topics, which we agree are interesting and relevant, but were unable to find much literature specifically addressing these topics in an Arctic glacier context. We have added a sentence raising each question at the end of this section as suggested.

L721-723: Perhaps in non-glacial rivers yes, but the N:P ratio in glacial rivers is much lower than typical Redfield ratio (16:1) if using end members in Table 3 (more like 7:1). Related to the point, but to me this suggests that DON contributes a significant amount of "biolabile" N, contribution from NH4, and/or adsorption of PO4 to particles (and removal from the fjord surface) are all possible. I think this also provides some nice context as to why these environments are likely to be N limited.

R: The Redfield ratio is challenging to apply in glaciated catchments, and yes this is largely because we don't have particularly extensive measurements of species like DON/DOP. Ratios are also subject to large uncertainties because many parameters (PO4/NH4) are often close to detection with small differences in the values themselves or how blanks are handled making a very large difference to any ratios, and similarly the effect of including/not including NH4/DON may matter much more than in waters where the bioavailable N/P pools are more dominated by few er species.

The cellular N:P ratio however (e.g. Ren et al., 2019) is a far less ambiguous indicator of whether either N or P is generally deficient because it reflects the actual status of organisms and doesn't

rely on assumptions about what aqueous labile/dissolved species should/shouldn't be included in any ratio calculation and this supports the idea that P isn't particularly in excess.

It's very difficult to comment further as there are few conclusive studies defining the relative importance of N/P to constraining PP in plume environments, and to some extent it may not simply matter as light limitation prevails to such an extent that neither nutrient supply is exhausted by phytoplankton during the short time period water spends in turbid glacier outflows.

L751-758: Citing van der Merwe et al. (2019) would also be appropriate in this section.

R: Relevant van der Merwe work added (an earlier paper by the same group is more appropriate for this specific point).

L841-861: I think it would also be appropriate to cite van der Merwe et al. (2019) here as well.

**R: Added.**

*L859: How about "how particle bio-lability, ligands and estuarine mixing processes moderate the glacier-to-ocean Fe transfer"*

R: 'Bio-lability' is not a well-defined term in a marine context, but it generally refers to the ease with which biology directly changes the lability of solid phases which we are not referring to here. In an Fe-flux context the most important term by far in terms of changing Fe species with respect to increasing the size of the bioavailabile pool is likely the ligand-binding of Fe phases (Gledhill & Buck 2012) rather than the lability of the solid phases as these environments are saturated with labile Fe irrespective of the definition (e.g. Lippiatt work cited). We prefer the terms 'bioaccessibility' and 'bioavailability', which are better defined.

L936: suggest "minor source of total DOM to the fjord".

R: We don't see the difference here, as they are a minor source irrespective of how DOM/DOC is defined.

L943-947: I think there's a bit of repetition here. It is sufficient to say that glacial DOM appears to be more labile than non-glacial DOM because of the high proportion of aliphatic or protein-like compounds (e.g. Barker et al., 2013, Dubnick et al., 2010, Hood et al., 2009, Pautler et al., 2012) commonly associated with microbial activity.

R: There are 3 separate issues: 1) is how the total flux of DOC may change as glaciers retreat, 2) is how labile this material is, 3) is how the lability may change with retreat. Therefore, we prefer the sentences as written.

L970-972: I think saying it is diluted and consumed is more appropriate. DOM is relatively unique in that the concentration doesn't really matter (to a degree) – the composition is most important. Even if it dilutes marine waters, it's (glacial meltwater) highly biolabile nature means it is likely to be relatively important for heterotrophs.

R: The main process reducing concentration on this scale is dilution as evidence by linearity in salinity plots. Lability and concentration are important for any nutrient, not just DOM/C. Even deriving labile OC rather than total OC, by any measure, glacier derived OC does not constitute a significant (or sometimes even measurable) OC enrichment in the marine environment. This can be seen in both chemical, uptake and bacterial distribution studies. The cited studies consider both the lability and the total flux and still conclude the impact is negligible or minor.

L973-976: This is complex and a this is perhaps a generalisation considering the complex mixture of organic compounds present and the relatively high salinity of these sampling locations.

R: The isotopic work quoted is consistent with OC budgets in both Young Sound and Kongsfjorden which reach similar conclusions (see above comment). Irrespective of how mixed the composition of OC in meltwater is, it does not constitute a large fraction of the OC utilized by bacteria in this location.

L990-992: Only some may be photodegraded so this is an over generalisation. There is a standing stock of DON in the ocean surface that is pretty recalcitrant are resistant to biological utilisation (it is favourable for nitrogen fixation for example; Letscher et al., 2013).

R: 991 'is' rephrased 'can be', but we note in the Arctic/glacier context this and biological uptake are the prime degradation routes. The standing stock in offshore marine waters is not related to glacier discharge.

L995: It is appropriate to reference Wadham et al. (2016) here.

**Added.**

L1002: Appropriate to reference Hawkings et al. (2016) here because it is one of the only papers that has and discusses values for DOP in glacial meltwater rivers (low to negligible concentrations).

**Added.**

L1048-1050: The previous references refer to catchments with barely any glacial meltwater input contribution to discharge and this should be made clear here as this sentence does not make that clear.

L1055: It is slightly misleading to say that freshwater runoff in these scenarios includes glacial meltwater. Please change to "non-glacial freshwater sources". As I noted in my previous review, meltwater discharge in Patagonia is not currently decreasing, as opposed to non-glacial precipitation driven catchments where precipitation does appear to be undergoing a long-term decline.

**R: (Obsolete, this section is slimmed and no longer exists as a major component of the text).**

L1083-1085: Completely agree that glacial discharge affects stratification, but doesn't it also supply very cold water to the surface. The argument of an increased prevalence of HABs in Patagonian fjords and the Alaskan case study (Vandersea et al., 2018) is that warming of the surface also plays an important role. I therefore don't fully understand the link between increasing glacial meltwater freshwater discharge in the Arctic, a warming fjord surface layer (despite the increasing flux of cold, very fresh water) and HABs.

R: Perhaps the reviewer is not aware, but cold freshwater induces density stratification, which can result in warming of surface waters. Therefore, even if freshwater enters the Arctic at the ocean surface at temperatures close to zero, it can rapidly heat to warm temperatures in a surface low-salinity layer in a confined fjord (maybe 5-10°C, in a thin layer at the surface in the freshest water in Greenland's glacier fjords). Meltwater in surface layers in glacier fjords is therefore very warm and adding more meltwater further stabilizes surface waters and can lead to higher temperatures. This is well established and is a useful example of how freshwater properties alone cannot easily be used to make extensive comments on marine water column changes.

Hence the freshwater release is driving the temperature change (and thus potentially HAB susceptibility) in glacier fjords.

Table 3: Lawson et al. (2014a) do not produce their own DON data. The concentrations used here are from the Wadham et al. (2016) reference that was "in review" when this article was published. This concentration estimate was likely based off a mean seasonal concentration rather than the discharge weighted concentrations given in Wadham et al. (2016) of 1.7 uM.

R: Thank you. We were not aware of this; references changed accordingly.